# Generalized Oversampling for Learning from Imbalanced datasets and Associated Theory: Application in Regression

**Samuel Stocksieker**  *samuel.stocksieker@univ-amu.fr*
*Laboratoire de Sciences Actuarielle et Financière (UR SAF), Université Claude Bernard Lyon 1*
*Institut de Mathématique de Marseille, Aix-Marseille Université*

**Denys Pommeret**  *denys.pommeret@univ-amu.fr*
*CNRS*
*Centrale Marseille*
*Institut de Mathématique de Marseille, Aix-Marseille Université*

**Arthur Charpentier**  *charpentier.arthur@uqam.ca*
*Département de Mathématique, Université du Québec à Montréal*

**Reviewed on OpenReview:** *https: // openreview. net/ forum? id=DLqPhQxgYu*

## Abstract

In supervised learning, it is quite frequent to be confronted with real imbalanced datasets. This situation leads to a learning difficulty for standard algorithms. Research and solutions in imbalanced learning have mainly focused on classification tasks. Despite its importance, very few solutions exist for imbalanced regression. In this paper, we propose a data augmentation procedure, the GOLIATH algorithm, based on kernel density estimates and especially dedicated to the problem of imbalanced data. This general approach encompasses two large families of synthetic oversampling: those based on perturbations, such as Gaussian Noise, and those based on interpolations, such as SMOTE. It provides an explicit form common to such machine learning algorithms. and new synthetic data generators can be deduced. We apply GOLIATH in imbalanced regression combining such generator procedures with a new wild-bootstrap resampling technique for the target values. We evaluate the performance of the GOLIATH algorithm in imbalanced regression where we compare our approach with state-of-the-art techniques.

## 1 Introduction

Many real-world forecasting problems are based on predictive models in a supervised learning framework and standard algorithms can fail when the target variable is too skewed. Learning from imbalanced data concerns many problems with numerous applications in different fields (Krawczyk (2016), Fernández et al. (2018a)). The major part of such works concerns imbalanced classification (see for instance Buda et al. (2018), Cao et al. (2019), Cui et al. (2019), Huang et al. (2016), Yang & Xu (2020), Branco et al. (2016b) where many solutions propose a pre-processing strategy especially for the generation of new synthetic data. A large part of such existing methods consist in adapting the well know SMOTE algorithm (Fernández et al. (2018b)). Very few works have addressed the problem of imbalanced regression despite its relevance to many important real-world applications in different fields such as economy, meteorology, medicine, or insurance. As for imbalanced classification, some applications focus on predicting rare and extreme values, which can be of great interest. In the literature, the imbalanced regression corresponds to *the correct prediction of rare extreme values of a continuous target variable* Fernández et al. (2018b) but, contrary to the classification tasks, there is no level to quantify the imbalance and the labels are continuous. Unlike in a classification context, learning from imbalanced dataset for regression tasks leads to two additional problems: i) the

definition of the imbalanced phenomenon and ii) the identification of the observations that are considered as minority.

In this paper we propose a very general method, which we shall call GOLIATH (for Generalized Oversampling for Learning from Imbalanced datasets and Associated THeory) to deal with the imbalanced regression problem. The first step of GOLIATH is a synthetic covariates generation based on kernel density estimators. The second step of GOLIATH is concerned with the imbalanced regression: a new method based on a wild-bootstrap procedure is proposed for generating target values given the synthetic covariates. GOLIATH is then a two-step algorithm. In this paper, we concentrate on tabular data rather than images because many applications rely on structured data, and there are still very few solutions available to address them.

Our main contributions can be summarized as follows:

i) From a methodological perspective, **providing a unified and generalized statistical expression encompassing existing data augmentation techniques, notably the popular and widely utilised SMOTE method** (3).
This approach unifies two important families of synthetic data generators: perturbation-based (3.3) and interpolation-based (3.4) methods (see Branco et al. (2016b)). Recent research emphasizes the ongoing importance of explaining the properties of SMOTE, and GOLIATH follows this approach by analyzing the structure of the SMOTE algorithm which can be considered as a mixture distribution.

ii) **Deducing new synthetic data oversampling methods using the large and adaptative expression of GOLIATH** (4): more flexible, more suitable and empirically outperform state-of-the-art approaches (6). More precisely,

 – **Introducing some non-classical kernels density estimates** (4.2), to adapt and control the generation of new synthetic data according to the support of covariates, especially in discrete or bounded cases. Moreover, the proposed kernels use an efficient estimate of the bandwidth parameter yielding the method data-driven.
 – **Extending the interpolation techniques (SMOTE) with a Beta distribution and then with a Gaussian-Beta distribution extending the generation in the data space** rather than onto the segments (4.1).

iii) **Proposing innovative methodologies to address Imbalanced Regression**

 – **Introducing weights for the data generation** (5)
 This is an important issue in this topic, unlike for classification tasks. Furthermore, our approach preserves the continuity of the target variable distribution and avoids making it discrete or binary, which is generally done in other works.
 – **A new technique to generate the target variable** (5): by combining a wild-Bootsrap with a random forest in order to consider its dependencies with the covariates and the new synthetic data.
 – **A new way to generate synthetic data**:
 The "mix" mode, provides a new solution to the main challenge of synthetic data generation, avoiding overfitting and the introduction of bias.

The paper is organized as follows. In Section 3 we give a general form of our data augmentation procedure corresponding to the first step of GOLIATH. We study some standard perturbation and interpolation methods that are included in this approach, such as SMOTE and Gaussian Noise. In Section 4 we develop the theory to obtain new generators. In Section 5 we will look more closely at the imbalanced regression, corresponding to the second step of GOLIATH. Numerical results on several applications are presented in Section 6. Finally, we discuss the method proposed in Section 7.

## 2 Related works

Learning from imbalanced distributions is a topic that concerns several domains and numerous applications Krawczyk (2016). As highlighted in the literature on the subject, especially in the survey by Branco et al. (2016b), the case of regression has been very little addressed and there are very few solutions available to date. Most of the works addressing regression tasks are based on a relevance function Torgo & Ribeiro (2007). The objective of this function, denoted by $\phi$, is to associate a relevance value, ranging from 0 to 1, to each value of the target variable $Y$: $\phi(Y) : \mathbb{R} \longrightarrow [0, 1]$, with 1 representing maximum relevance, for extreme values of $Y$. Using a given threshold $t_r$, it is possible to divide the dataset into two parts: the subset of minority observations, denoted as $D_r$, and the subset of majority observations, denoted as $D_n$, such that: $\mathcal{D}_r = \{(\boldsymbol{x}_i, y_i)_{i=1,\cdots,n} \in \mathcal{D} : \phi(y_i) > t\}$ and $\mathcal{D}_N = \mathcal{D} \setminus \mathcal{D}_r$, $\mathcal{D}$ being the set of all observations.

Once the support is binarized in this way, it becomes possible to adapt solutions from imbalanced classification as SMOTE, Gaussian Noise, etc. (Torgo et al. (2013), Branco et al. (2017), Branco et al. (2019), Ribeiro & Moniz (2020), Song et al. (2022), Camacho et al. (2022), Branco (2018)). This approach presents the disadvantage of dividing the continuous distribution of the target variable into binary classes and therefore involves a loss of information. Indeed, considering an example of a threshold at 0.8, an observation with a relevance value of 0.79 will be considered as majority while an observation with a value of 0.81 will be considered as extreme. Furthermore, this "extreme" observation will have the same importance as an observation with a relevance value of 0.99. Furthermore, providing a utility function and an associated threshold is difficult for the user. However, Ribeiro (2011) suggested constructing a relevance function. This default function is based on a box plot of the distribution, which may not be representative. Additionally, it only considers extremes and not rare values inside the distribution, which are not extreme.

Steininger et al. (2021) proposed a new way of approaching the issue by introducing a cost function. The authors' objective is to individually weight observations based on the rarity of the target value. They therefore propose to deduce a weight inversely proportional to the probability of occurrence of the target variable. The probability of occurrence is estimated by a classical kernel density estimator. This method is similar to the one we propose. The authors use the opposite $(1 - \alpha \widehat{f}(y)$ with $\widehat{f}(y)$ the estimated density function of the target variable $Y$) whereas we use the inverse. The use of the inverse $(1/\widehat{f}(y))$ of the estimated density function is also found in Yang et al. (2021). However, the authors partition the domain of the target variable, which also leads to information loss. Both approaches use this weighting of observations to redefine the loss function, i.e., with a cost-sensitive approach. The main drawback of this approach, while promising, is that the authors modify a neural network. However, as indicated below, this type of approach may not be the most effective on tabular data. Our preprocessing solution has the advantage of being able to be associated with any classical learning algorithm, making GOLIATH universal. Indeed, this is what we do in the experiments where an autoML with 10 different models is combined with GOLIATH.

Huang et al. (2022) built on a similar idea by using the inverse of the kernel density estimator for the target variable to create a utility function. However, similar to approaches based on relevance functions, the authors divide the data into subsets of rare and "normal" observations using a threshold. This approach suffers from the same drawbacks as those based on the utility function, except that the relevance function is more intuitive and allows for the treatment of rare non-extreme data. They then suggested optimizing the undersampling of normal values using the relevance function and assessing the impact on prediction (MSE profit & loss). For oversampling, they propose using a conditional VAE to incorporate the concept of "rare/normal" as useful labels for data generation.

More recently, other methods have emerged by using deep learning approaches, for dealing with images, such as Sen et al. (2023), Ding et al. (2022), Gong et al. (2022) or Wang & Wang (2024). However, the techniques proposed in this context rely on deep learning. While these approaches are highly effective with images, they often require adjustments to be effective with tabular data. Indeed, some authors have shown that standard deep learning approaches can be ineffective for certain tasks with tabular data (e.g Shwartz-Ziv & Armon (2022), Ma et al. (2020), Borisov et al. (2022) or Grinsztajn et al. (2022)). But other works have shown that these techniques can ultimately be effective if adapted specifically to some tabular data (e.g Gorishniy et al. (2021), Katzir et al. (2020) or Shavitt & Segal (2018)).

## 3   A New Kernel-Based Oversampling Formulation

In this section, we present our generic framework and demonstrate that two major families of synthetic data generation can be written in this form.

### 3.1   Notations and Problem Setting

We consider a sequence of observations $\{(\boldsymbol{x}_1, y_1), \cdots, (\boldsymbol{x}_n, y_n)\}$, which are realizations of $n$ iid random variables $(\boldsymbol{X}, Y)$, where the target variable $Y$ is univariate and the covariate $\boldsymbol{X}$ is a $p$-dimensional random vector. The components of $\boldsymbol{X} = (X_1, \cdots, X_p)$ are supposed to be continuous or discrete and $Y$ is supposed to be quantitative. Let $\boldsymbol{x}_{ij}$ the variable $j$ for the observation $i$ and $\boldsymbol{x}_j$ the observed variable $j$. Write $\tilde{\boldsymbol{x}} = \{\boldsymbol{x}_1, \cdots, \boldsymbol{x}_n\}$ the set of all observations and $\boldsymbol{x}^*$ a new synthetic data. Finally, we designate $\boldsymbol{x}_{i.}(\ell)$ as the $\ell$th nearest neighbors of an observation $\boldsymbol{x}_{i.} = \{x_{i1}, \cdots, x_{ip}\}$.

As with classification tasks, learning from an imbalanced distribution can alter the results of standard algorithms in a regression context, i.e., when the target variable is continuous. Indeed, conventional methods presuppose a uniform importance of the target variable values Branco et al. (2016b). Furthermore, the metrics used to measure performance, as well as optimization criteria to calibrate algorithms, measure an average error level across all values in a uniformly weighted manner Torgo & Ribeiro (2009).
**In regression, the continuous and infinite nature of $Y$ introduces two main challenges:**

   i) **It is not easy to define "rare" values and differentiate them from "frequent" values** - unlike in classification where this is directly given by the classes;

   ii) **Therefore, measuring imbalance is not immediate and is complicated to assess** - unlike in classification where it suffices to compare the classes.

If the proposed solution to the imbalanced regression problem relies on synthetic data generation, then it introduces a **third challenge**. Indeed, unlike classification, where the labels of $Y$ remain unchanged when creating synthetic data, in regression, it is necessary to generate new relevant values for the target variable. In this way, GOLIATH combines a two-step procedure: the first generates $X^*$ and the second deduces the generation of $Y^*$ using an innovative procedure based on a wild bootstrap and is based on $X^*$.
Although the problem of imbalanced regression is becoming increasingly recognized, there is no universal formal definition: Ribeiro (2011) and all works associated propose to define it by reducing it to a binary classification problem based on a relevance function (Branco et al. (2016b)) - which we avoid doing. Ren et al. (2022) proposes to define it as follows: "*imbalanced regression assumes that the training set and test set are drawn from different joint distributions, ptrain(x, y) and pbal(x, y) respectively, where the training set's label distribution ptrain(y) is skewed and the balanced test set's label distribution pbal(y) is uniform*. However, we disagree with this idea because, in many real-world applications, the target variable is naturally distributed asymmetrically. Yang et al. (2021) "*formally define the Deep Imbalanced Regression task as learning from imbalanced data with continuous targets, and generalizing to the entire target range*". **We prefer to formally and simply define the problem of imbalanced regression as follows: learning from continuous distributions that include rare values (including extremes) that are precisely relevant to the studied phenomenon.** This definition builds upon the notion of utility of Ribeiro (2011), but we avoid binarizing or discretizing the problem. Since the distribution is continuous, the relevance of values is assumed to be progressive, so binary classification does not seem relevant to us. The scarcity of observations can be due to two factors: either the distribution is inherently asymmetric, and thus the sample will also be theoretically asymmetric, or the distribution is not necessarily asymmetric, but sampling bias generates this phenomenon in the observed sample distribution (as defined in Ren et al. (2022) or more formally in Stocksieker et al. (2023)).

The notion of rarity is therefore crucial in defining the phenomenon of imbalance. We propose to define rarity as follows (inspired by Stocksieker et al. (2023)): Let $Y$ be the target variable taking values in $\mathcal{Y} \subset \mathbb{R}$. We denote by $\widehat{f}_Y$ a kernel density estimator of its probability density function (pdf). We consider that there is an imbalanced situation in one of the following two cases:

Case 1: When $\widehat{f}_Y$ significantly deviates from a given pdf.

Case 2: When $\widehat{f}_Y$ is significantly too small.

In both cases, we write $f_0$ for the given pdf (Case 1) or for the uniform pdf (Case 2). We propose the following definition of imbalance: We define a regression problem as $(\alpha, \beta)$-imbalanced if there exists a subset $\chi \subset \mathcal{Y}$ such that

$$\int_\chi f_0(y)dy \geq \beta \quad \text{and} \quad \frac{\displaystyle\int_\chi \widehat{f}_Y(y)dy}{\displaystyle\int_\chi f_0(y)dy} < \alpha.$$

Note that in Case 2, the quantity $\int_\chi f_0(y)dy$ is the Lebesgue measure of $\chi$, up to a factor. In simpler terms, an imbalanced regression implies that a sample is significantly different from a target distribution (Case 1) or under-represented (Case 2) for at least a significant portion of the support of $Y$. This means that regions with high support in the target distribution would be ignored. The level of imbalance increases notably with larger $(\alpha, \beta)$ values. As mentioned earlier, this issue also relies on the sample size. When $f_Y$ and $f_0$ have the same support, the imbalance diminishes with a growing sample size $n$. Conversely, if certain segments of $Y$ support remain unobserved, the problem may persist even with large $n$.

## 3.2 General Formulation of GOLIATH

We define GOLIATH the generalized oversampling procedure based on the form of the following weighted kernel density estimate:

$$g_{\boldsymbol{x}^*}(\boldsymbol{x}^*|\tilde{\boldsymbol{x}}) = \sum_{i \in \mathcal{I}} \omega_i K_i(\boldsymbol{x}^*, \tilde{\boldsymbol{x}}), \tag{1}$$

where $(K_i)_{i \in \mathcal{I}}$ is a collection of kernels, $(\omega_i)_{i \in \mathcal{I}}$ is a sequence of positive weights with $\sum_{i \in \mathcal{I}} \omega_i = 1$, and $\mathcal{I}$ represents a subset of $\{1, 2, \cdots, n\}$. Here the index $^*$ stands for the synthetic data. In this generalized form of a kernel density estimate, a kernel is associated with each observation $x_i$ and takes into account the set of all points $\tilde{x}$. This is a general form that is specified later depending on the kernels used. We recall that a kernel $K$ is a positive and integrable function such that $K(u, \tilde{x}) \geq 0$ and $\int_{-\infty}^{+\infty} K(u, \tilde{x})du = 1$. In (1) we propose a general form for the conditional density for the synthetic data generators. The objective is to use the flexibility of the kernels to estimate the density of covariates in order to obtain synthetic data that reflects the distribution of the observations.

We can show that (1) generalizes perturbation-based and interpolation-based synthetic data oversampling. We give an illustration with the basic algorithms ROSE, Gaussian Noise and SMOTE in Subsections 3.3 and 3.4 where we demonstrate that these methods are particular cases of the generalized form (1), with corresponding parameters summarized in Appendix A.2. Several existing methods can be rewritten in the form (1) and we give some illustrations in Appendix A.3. In Section 4 we will show that some particularly interesting new methods can be derived from the generic form (1) and we will compare some of them to current competitors in the imbalanced regression context.

REMARK **1.** *The generators in (1) are called smoothed bootstrap methods (see Silverman & Young (1987), Hall et al. (1989), De Angelis & Young (1992)). Smoothed bootstrap consists in drawing samples from kernel density estimators of the distribution. It can be decomposed into two steps: first, a seed is randomly drawn and second, a random noise from the kernel density estimator is added to obtain a new sample. In the form (1), the first step is represented by the drawing weight $\omega_i$ and the second by the kernel $K_i(x)$. Convergence properties of smoothed bootstrap are studied in De Martini & Rapallo (2008) and Falk & Reiss (1989). They proved the consistency of the smoothed bootstrap with classical multivariate kernel estimator and more specifically the convergence in Mallows metric. As described by the authors, the smoothed bootstrap provides better performances than a classical bootstrap when a proper choice of smoothing parameters is used. Other works have focused on the consistency of the multivariate kernel density estimate and proposed a relevant bandwidth matrix, see for instance Silverman (1986), Scott (2015) and Duong & Hazelton (2005). All this work has been carried out outside the context of regression models and only with classical kernels. GOLIATH extends this to unbalanced regression using adapted kernels.*

### 3.3 Rewriting Interpolation Approaches

As presented in Fernández et al. (2018b), the Synthetic Minority Oversampling Technique (SMOTE) Chawla et al. (2002) is considered a "de facto" standard for learning from imbalanced data and has inspired a large number of methods to handle the issue of class imbalance. It is also one of the first techniques adapted to imbalanced target values in regression with Torgo et al. (2013). SMOTE algorithm can be summarized as follows: at each step of the data augmentation procedure[1], an observation is randomly selected, which we shall call *a seed*. Then, one of its $k$ nearest neighbors among $K$, which is a hyperparamer, is randomly drawn. Finally, a new data point is generated by interpolating between these two initial observations.

We will denote by $\boldsymbol{x}_{i.}$ the selected seed and by $\boldsymbol{x}_{i.}(\ell)$ the $\ell$th nearest neighbor of $\boldsymbol{x}_{i.}$ among its $K$ neighbors. The new data is then generated by linear interpolation between $\boldsymbol{x}_{i.}$ and $\boldsymbol{x}_{i.}(\ell)$. Each observation has a probability $1/n$ of being the seed, and each of its $k$ neighbours has a probability $1/k$ of being selected. Finally, writing $\boldsymbol{x}^*$ the synthetic data we have $\boldsymbol{x}^* = \lambda \boldsymbol{x}_{i.} + (1-\lambda)\boldsymbol{x}_{i.}(\ell)$, with $\lambda$ uniformly distributed $\mathcal{U}([0;1])$.

To show that this approach is a particular case of (1) we proceed in three steps:

1. Conditionally to $\boldsymbol{x}_{i.}$ and $\boldsymbol{x}_{i.}(\ell)$, the $j$th component of $\boldsymbol{x}^*$ is generated by a uniform distribution, with $\mathbb{1}$ the indicator function:

$$g_{\boldsymbol{x}^*}^{SMOTE}\left(\boldsymbol{x}_j^*|\tilde{\boldsymbol{x}},\boldsymbol{x}_{i.},\boldsymbol{x}_{i.}(\ell)\right) = \frac{\mathbb{1}_{[\min(\boldsymbol{x}_{ij},\boldsymbol{x}_{ij}(\ell)),\max(\boldsymbol{x}_{ij},\boldsymbol{x}_{ij}(\ell)])}\left(\boldsymbol{x}_j^*\right)}{|\boldsymbol{x}_{ij}(\ell) - \boldsymbol{x}_{ij}|}$$

Each component of $\boldsymbol{x}^*$ is drawn by the same uniform variable, that is $\boldsymbol{x}_j^* = \lambda \boldsymbol{x}_{i.} + (1-\lambda)\boldsymbol{x}_{i.}(\ell)$ for $j = 1,\cdots,p$, and we write the multivariate generating density as follows:

$$g_{\boldsymbol{x}^*}^{SMOTE}(\boldsymbol{x}_j^*|\tilde{\boldsymbol{x}},\boldsymbol{x}_{i.},\boldsymbol{x}_{i.}(\ell)) = \frac{\mathbb{1}_{[\min(\boldsymbol{x}_{ij},\boldsymbol{x}_{ij}(\ell)),\max(\boldsymbol{x}_{ij},\boldsymbol{x}_{ij}(\ell)])}\left(\boldsymbol{x}_j^*\right)}{|\boldsymbol{x}_{ij}(\ell) - \boldsymbol{x}_{ij}|}\mathbb{1}_{v_{i\ell}=v_{j\ell};\forall i,j}.$$

where $v_{i\ell} = (x_i^* - x_{i.}(\ell))/(x_{i.} - x_{i.}(\ell))$

2. Conditionally to $\boldsymbol{x}_{i.}$, $\boldsymbol{x}^*$ is generated according to a uniform mixture model (UMM) on the segments between $\boldsymbol{x}_{i.}$ and its $k$-nn. The same mixture component is used for each component giving:

$$g_{\boldsymbol{x}^*}^{SMOTE}(\boldsymbol{x}^*|\tilde{\boldsymbol{x}},\boldsymbol{x}_{i.}) = \frac{1}{k}\sum_{\ell=1}^{k}\frac{\mathbb{1}_{[\min(\boldsymbol{x}_{ij},\boldsymbol{x}_{ij}(\ell)),\max(\boldsymbol{x}_{ij},\boldsymbol{x}_{ij}(\ell)])}\left(\boldsymbol{x}_j^*\right)}{|\boldsymbol{x}_{ij}(\ell) - \boldsymbol{x}_{ij}|}\mathbb{1}_{v_{i\ell}=v_{j\ell};\forall i,j}.$$

3. More generally, $\boldsymbol{x}^*$ is generated according to a mixture of UMM as follows:

$$g_{\boldsymbol{x}^*}^{SMOTE}(\boldsymbol{x}^*|\tilde{\boldsymbol{x}}) = \frac{1}{n}\sum_{i=1}^{n}g_{\boldsymbol{x}^*}^{SMOTE}(\boldsymbol{x}^*|\tilde{\boldsymbol{x}},\boldsymbol{x}_{i.}) = \frac{1}{n}\sum_{i=1}^{n}\frac{1}{k}\sum_{\ell=1}^{k}\frac{\mathbb{1}_{[\min(\boldsymbol{x}_{ij},\boldsymbol{x}_{ij}(\ell)),\max(\boldsymbol{x}_{ij},\boldsymbol{x}_{ij}(\ell)])}\left(\boldsymbol{x}_j^*\right)}{|\boldsymbol{x}_{ij}(\ell) - \boldsymbol{x}_{ij}|}\mathbb{1}_{v_{i\ell}=v_{j\ell};\forall i,j}$$

$$= \frac{1}{n}\sum_{i=1}^{n}K_i^{SMOTE}(\boldsymbol{x}^*,\tilde{\boldsymbol{x}})$$

We finally obtain the form (1) with $\mathcal{I} = [0,n]$, $\omega_i = 1/n$ and $K_i(\boldsymbol{x}^*,\tilde{\boldsymbol{x}}) = K_i^{SMOTE}(\boldsymbol{x}^*,\tilde{\boldsymbol{x}})$. This new writing represents the conditional SMOTE density given the observation $\tilde{\boldsymbol{x}}$ which can be seen as a mixture of Uniform Mixture Model. We can relate this expression to the work of Elreedy et al. (2023) and Sakho et al. (2024) in which the authors give an expression of the unconditional SMOTE density, that is integrating the distribution over $\tilde{\boldsymbol{x}}$ in a context of class minority. Other methods derived from SMOTE can be recovered by (1) (see Appendix A.3).

---

[1]In the original version of SMOTE, the seed is drawn successively with a loop and not randomly. These two ways are very close when the generated sample size is large

### 3.4 Rewriting Perturbation Approaches

We illustrate (1) by recovering two classical data augmentation procedures, ROSE and Gaussian Noise (GN), as follows:

- At each step of the ROSE algorithm (see Menardi & Torelli (2014)) the seed $x_{i.}$ is selected randomly. Given $x_{i.}$ a synthetic data is generated with a multivariate density

$$g_{\boldsymbol{x}^*}^{ROSE}(\boldsymbol{x}^*|\tilde{\boldsymbol{x}}, \boldsymbol{x}_{i.}) = K_{H_n}^{ROSE}(\boldsymbol{x}^* - \boldsymbol{x}_{i.}) = \frac{1}{|H_n|^{1/2}} K(H_n^{-1/2}(\boldsymbol{x}^* - \boldsymbol{x}_{i.}))),$$

where $K$ denotes the multivariate Gaussian kernel and $H_n = diag(h_1, ..., h_p)$ is the bandwidth matrix proposed by Bowman & Azzalini (1999), with $h_q = (\frac{4}{(p+2)n})^{1/(p+4)} \hat{\sigma}_q, q = 1, ..., p$. Finally, a synthetic random variable $\mathbf{X}^*$ is generated with the density

$$g_{\boldsymbol{x}^*}^{ROSE}(\boldsymbol{x}^*|\tilde{\boldsymbol{x}}) = \frac{1}{n} \sum_{i=1}^n K_{H_n}^{ROSE}(\boldsymbol{x}^* - \boldsymbol{x}_{i.}) = \sum_{i=1}^n \omega_i K_{H_n}^{ROSE}(\boldsymbol{x}^* - \boldsymbol{x}_{i.}).$$

- Similarly to ROSE, at each step of the Gaussian Noise algorithm (see Lee & Sauchi (2000)) a seed is selected and synthetic data is generated. Finally, the generating multivariate density has the form

$$g_{\boldsymbol{x}^*}^{GN}(\boldsymbol{x}^*|\tilde{\boldsymbol{x}}) = \frac{1}{n} \sum_{i=1}^n K_{H_n}^{GN}(\boldsymbol{x}^* - \boldsymbol{x}_{i.}) = \frac{1}{n} \sum_{i=1}^n \frac{1}{|H_n|^{1/2}} K(H_n^{-1/2}(\boldsymbol{x}^* - \boldsymbol{x}_{i.})),$$

where $H_n^{GN} = diag(h_1, ..., h_p)$, $h_q = \sigma_{noise} \hat{\sigma}_q, q = 1, ..., p$.

Both cases are particular cases of (1) with $\omega_i = \frac{1}{n}$ and $K_i(\boldsymbol{x}^*, \tilde{\boldsymbol{x}}) = K_{H_n}(\boldsymbol{x}^* - \boldsymbol{x}_{i.})$, i.e. the same Gaussian kernel for all observations but with a different bandwidth matrix.

### 3.5 Global criticism

Although there are many extensions of SMOTE or ROSE and Gaussian Noise, such techniques suffer from some drawbacks. For the interpolations techniques, the directions in the data space are limited and deterministic because they depend only on the k-nn (nearest neighbors). Moreover, the distance from the seed is also limited because the new sample is on the segment with the drawn nearest neighbor. For the perturbation techniques, the directions in the data space are randomly generated and so they can more explore the space. The distance between the new sample and the seed is also unbounded. However, the directions are randomly chosen and do not respect the correlation between the data and their support and the correlations between variables.

## 4 New Kernel-Based Methods from GOLIATH

Since Goliath is a very generic form, its practical interest lies in the choice of a judicious combination of weights and kernels. We proceed in separating the two families interpolation and pertubation, deducing new generators from the form 1, which allows us to extend the initial approaches.

### 4.1 Generalized Interpolation Approaches

**A general form** We propose a particular form of (1) which generalizes the SMOTE algorithm as follows:

$$g_{\boldsymbol{x}^*}^{int}(\boldsymbol{x}^*|\tilde{\boldsymbol{x}}) = \sum_{i \in \mathcal{I}} \omega_i K_i(\boldsymbol{x}^*, \tilde{\boldsymbol{x}}) = \sum_{i \in \mathcal{I}} \omega_i \sum_{\ell \in \mathcal{J}_i} \pi_{\ell|i} \; g_{i,\ell}^{int}(\boldsymbol{x}^*|\tilde{\boldsymbol{x}})$$

where $g_{i,\ell}^{int}(\boldsymbol{x}^*|\tilde{\boldsymbol{x}})$ is an interpolation function on $[\boldsymbol{x}_{i.}, \boldsymbol{x}_{i.}(\ell)]$, $\mathcal{J}_i$ denoting the set of $k$-nn associated to $\boldsymbol{x}_{i.}$. SMOTE is then a particular case when $\omega_i = \frac{1}{n}$, $\pi_{\ell|i} = \frac{1}{k}$ and $g_{i,\ell}^{int}(\boldsymbol{x}^*|\tilde{\boldsymbol{x}}) = \frac{\mathbb{1}_{[0;\boldsymbol{x}_{i.}(\ell) - \boldsymbol{x}_{i.}]}(\boldsymbol{x}^* - \boldsymbol{x}_{i.})}{|\boldsymbol{x}_{i.}(\ell) - \boldsymbol{x}_{i.}|}$ represents a uniform distribution between the vectors $\boldsymbol{x}_{i.}$ and $\boldsymbol{x}_{i.}(\ell)$.

**Nearest Neighbors Smoothed Bootstrap** Since the uniform distribution coincides with the Beta distribution with parameters $\alpha = \beta = 1$, a natural extension of SMOTE is to consider a general Beta distribution. We find the same idea in Yao et al. (2022) within another context. The very flexibility of the Beta distribution suggests us to propose $\boldsymbol{x}^* = \lambda \boldsymbol{x}_{i.} + (1 - \lambda)\boldsymbol{x}(\ell)_i$ for $i = 1, \cdots, n$, where $\lambda$ follows a generalized Beta distribution. By abuse of notation, we get the following interpolation function:

$$g_{i,\ell}^{int}(\boldsymbol{x}^*|\tilde{\boldsymbol{x}}) = \frac{\Gamma(\alpha + \beta)}{\Gamma(\alpha)\Gamma(\beta)}(\boldsymbol{x}^* - \boldsymbol{x}_{i.})^{\alpha-1}(\boldsymbol{x}_{i.}(\ell) - \boldsymbol{x}^*)^{\beta-1}\mathbb{1}_{[0;\boldsymbol{x}_{i.}(\ell)-\boldsymbol{x}_{i.}]}.$$

**Extended Nearest Neighbors Smoothed Bootstrap** Finally, the previous methods based on interpolation are limited to the "seed - $k$-nn segments" and therefore do not reach all the data space. To avoid generating on a bounded or discrete support we propose to extend these approaches to any part of the support by adding a Gaussian distribution on the segment as follows:

$$g^{e-int}(\boldsymbol{x}^*|\tilde{\boldsymbol{x}}) = \sum_{i\in\mathcal{I}} \omega_i \sum_{\ell\in\mathcal{J}_i} \pi_{\ell|i} \, g_{i,\ell}^{e-int}(\boldsymbol{x}^*|\tilde{\boldsymbol{x}}),$$

where the extended interpolation function is a Beta Gaussian mixture, that is, $g_{i,\ell}^{e-int}(\boldsymbol{x}^*|\tilde{\boldsymbol{x}})$ is the density of a Gaussian distribution $N(\theta, \sigma^2)$ where $\theta$ is generated by $g_{i,\ell}^{int}(\boldsymbol{x}|\tilde{\boldsymbol{x}})$. To rely on the recent literature, we remark that SASYNO algorithm Gu et al. (2020) is a special case of this methodology. This extended version can be viewed as a hybrid method between interpolation and perturbation techniques. It provides a good compromise between the interpolation and perturbation approaches because it can generate in the whole data space as the perturbation approach i.e. constraint-free, but assigns a distribution to the directions towards the segments, that is orienting the perturbation toward the $k$-nn.

REMARK **2.** *We tried to adapt the k-nearest neighbors density estimate (Biau & Devroye (2015)) that is a bandwidth-variable kernel (also called a balloon kernel) as a generator but its computation time is currently too high to be used.*

## 4.2 Generalized Perturbation Approaches

**Classical Smoothed Bootstrap** As the ROSE and GN techniques use a multivariate Gaussian kernel estimate with a diagonal bandwidth matrix, we can rewrite their associated generating density as follow:

$$g_{\boldsymbol{x}^*}(\boldsymbol{x}^*|\tilde{\boldsymbol{x}}) = \sum_{i=1}^{n} \omega_i \prod_{j=1}^{p} K_{h_j}(x_j^* - x_{ij}) \tag{2}$$

with $K_{h_j}(u) = (2\pi)^{-1/2}h_j^{-1}e^{-\frac{1}{2h_j^2}u^2}$ the univariate gaussian kernel density estimator with smoothing parameter $h_j$. Such kernels are clearly not adapted for asymmetric, bounded or discrete variables. This remark is also true for the work of Yang et al. (2021) which uses some symmetric kernels to improve learning of imbalanced datasets.

**Non-Classical Smoothed Bootstrap** To fix the drawback of the classical kernel we extend (2) by adapting (1) to the support of $\boldsymbol{x}$, considering some non-classical kernels (we refer to some works handling the kernel density estimation for specific distributions inspired from Bouezmarni & Rombouts (2010), Someé (2015), Hayfield & Racine (2008), Chen (2000)). We rewrite (1) as

$$g_{\boldsymbol{x}^*}^{per}(\boldsymbol{x}^*|\tilde{\boldsymbol{x}}) = \sum_{i\in\mathcal{I}} \omega_i \prod_{j=1}^{p} K_{h_j}(x_j^*, x_{ij})$$

where $K_{h_j}(u, x)$ is a univariate kernel adapted to the nature of the $j$th variable and specifically defined on $x$ as follows:

- Gaussian kernel for a variable defined on $\mathbb{R}$ (classical kernel):

$$K_h(u, x) = \frac{1}{h\sqrt{2\pi}}e^{-\frac{1}{2}(\frac{u-x}{h})^2}.$$

- Binomial kernel for a discrete variable defined on $\mathbb{N}$:

$$K_h(u, x) = \frac{(x+1)!}{u!(x+1-u)!} \left(\frac{x+h}{x+1}\right)^u \left(\frac{1-h}{x+1}\right)^{x+1-u}.$$

- Gamma kernel for a positive asymmetric distribution defined on $[a, +\infty]$:

$$K_h(u, x) = \frac{u^{(x-a)/h}}{\Gamma(1+(x-a)/h)h^{1+(x-a)/h}} \exp\left(\frac{-u}{h}\right) \mathbb{1}_{[a,+\infty]}(u).$$

- Negative Gamma kernel for a negative asymmetric distribution defined on $[-\infty, b]$:

$$K_h(u, x) = \frac{u^{-(x-b)/h}}{\Gamma(1-(x-b)/h)h^{1-(x-b)/h}} \exp\left(\frac{-u}{h}\right) \mathbb{1}_{[-\infty,b]}(u).$$

- Beta kernel for a variable defined on $[0, 1]$:

$$K_h(u, x) = \frac{u^{x/h}(1-u)^{(1-x)/h}}{\mathcal{B}(\frac{x}{h}+1, \frac{1-x}{h}+1)} \mathbb{1}_{[0,1]}(u).$$

- Truncated Gaussian kernel for a variable defined on $[a, b]$:

$$K_h(u, x) = \frac{\alpha}{h\sqrt{2\pi}} e^{-\frac{1}{2}(\frac{u-x}{h})^2} \mathbb{1}_{[a,b]}(u),$$

$$\alpha := \left(\int_a^b \frac{1}{h\sqrt{2\pi}} e^{-\frac{1}{2}(\frac{u-x}{h})^2}\right)^{-1}$$

Note that if the Dirac kernel ($\mathbb{1}_{\boldsymbol{x}=\boldsymbol{x}_{i.}}$) is used, we get the standard bootstrap: 1 includes also the simple oversampling. It is important to note that the GOLIATH algorithm uses an estimation of the smoothing parameter $h$ provided by some specific R-package dedicated to the density estimation (for instance, it uses the Silverman estimation for the Gaussian kernel). Their estimates are based on properties of univariate consistency. Another technique to deal with skewed or heavy-tailed distributions is to apply a transformation of the data in order to use classical kernel density estimation (Charpentier & Flachaire (2015), Charpentier & Oulidi (2010)) but it necessitates proposing a relevant transformation which exceeds the scope of this paper.

REMARK **3.** *The use of a diagonal bandwidth matrix in (2) does not take into account the correlation between variables. To improve this issue, we could consider a full (symmetric positive definite) smoothing matrix. In that case, we would use a multivariate kernel density estimate considering the correlation between the variables which would be optimal for generating data. However, the estimation of such a matrix is generally based on the covariance matrix which does not adequately capture non-linear correlations. In practice, it can be challenging, even inconsistent, to find a form of a multivariate kernel that adapts to all data and their support. It is interesting to note that kernel functions can be linked to a wide range of statistical work, such as graph-based learning methods (SarcheshmehPour et al. (2023)).*

### 4.3 Goliath Overview

The GOLIATH algorithm is summarized in Figure 4. A cartography of GOLIATH is also given in Figure 1. The algorithm gives the possibility to choose, with the "mode" parameter, the kind of the returned sample: a full synthetic dataset, an augmented one, or a mixed one. The mixed sample is constructed as follows: keep the original observation for the first occurrence of the seed and synthetic data for the next. This mode corresponds to performing an undersampling and an oversampling. More precisely, to preserve the maximum of information and avoid potential overfitting, we suggest to: keep the initial observation for its first drawing and generating synthetic data from it for the other drawing which is the "mix" mode in the GOLIATH algorithm. This technique theoretically helps in reducing bias (by keeping the real data) while avoiding overfitting (by not duplicating the same observations). More details on this option are given in Appendix B.

### 4.3.1 GOLIATH Overall Algorithm

The GOLIATH algorithm is summarized in Figure 4 and Figure 1 presents its cartogrophy. It can be seen that the structure of GOLIATH is that of a generalized smoothed bootstrap, that is, it utilizes weighted kernel density estimators. GOLIATH encompasses the family of existing perturbation-based generators (such as ROSE and GN) which can be referred to as the *Classical Smoothed Bootstrap*. This also includes what is known as the *SMOTE Family*, which is very broad and encompasses many methods (see Fernández et al. (2018b)). We have the two generalizations of the *SMOTE family* that we propose: *Nearest Neighbors Smoothed Bootstrap* and *Extended Nearest Neighbors Smoothed Bootstrap*. We also find the *Non-Classical Smoothed Bootstrap*, with specific kernels that we propose in this paper. Finally, we have also added a proposed kernel that seems interesting but appears to be complicated to use in practice: the *Balloon kernel* using the kNN density estimate.

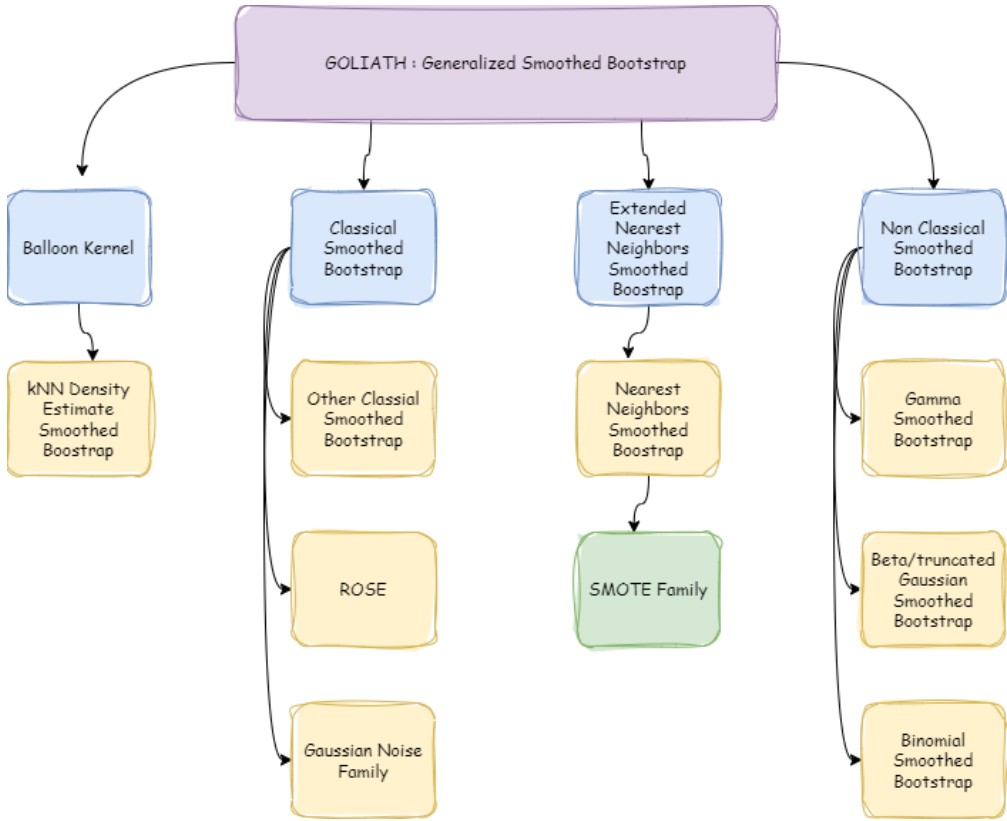

Figure 1: GOLIATH algorithm and cartography

## 5 GOLIATH as a Solution for Imbalanced Regression

In this section, we introduce new methodologies to deal with Imbalanced Regression

Using the first generator step of GOLIATH (methods proposed in 2.3.1 and 2.3.2) we can generate synthetic covariates $X^*$. We then have to generate the target variable $Y$ given such $X^*$.

**A non-parametric method for drawing weights**  We propose here to define the drawing weights $\omega_i$ in (1) as the inverse of the kernel density estimate for the target variable $Y$: the more isolated an observation is, the higher its drawing weight. Our approach is comparable to the Inverse Probability Weighted Estimator (IPWE) in survey sampling Hernán & Robins (2006). The IPWE is a statistical method used to estimate parameters in situations where data come from a population different from the one targeted for inference. IPWE employs weights calculated from the inverse probabilities of sampling events to adjust observations,

often used to correct sampling biases and extrapolate sample results to the target population. To avoid giving disproportionate weights (typically too large a weight for extreme values), we can use a hyperparameter $\alpha$ as below. The weights are normalized to get a sum equal to 1. For $i = 1, \cdots, n$, the normalized drawing weight $\omega_i$ of observation $i$ is thus defined as follows:

$$\omega_i := \frac{q_i}{\sum_j q_j}, \text{ with } q_i := \frac{1}{\widehat{f}_y(y_i)^\alpha}, \tag{3}$$

where $\widehat{f}_y$ denotes a convergent kernel density estimator.

In the following, we assume $\alpha = 1$.

REMARK **4.** *With $\alpha = 1$, the generated random variable $Y^*$ satisfying $\mathbb{P}(Y^* = y_i) = \omega_i$ is close to a uniform distribution. More precisely Stocksieker et al. (2023) generalized a result of Smith & Gelfand (1992) by showing that choosing $q_i = \frac{f_0(y_i)}{\widehat{f}_y(y_i)}$, with $f_0$ a target probability density function (pdf), then the pdf of $Y^*$ converges to $f_0$. In our case $f_0$ is a uniform distribution and we can apply this result as soon as the support of $Y$ is bounded. In this way, we obtain a test sample with a distribution of the target variable $Y$ approximately uniform i.e balanced. We find the same idea in Ren et al. (2022): "Therefore, imbalanced regression assumes that the training set and test set are drawn from different joint distributions, ptrain(x, y) and pbal(x, y) respectively, where the training set's label distribution ptrain(y) is skewed and the balanced test set's label distribution pbal(y) is uniform."*

In Yang et al. (2021) the authors propose also using the inverse of the estimated label density to weight the loss function but apply a discretization of the target variable. The strength of our approach is to preserve the continuity of the distribution whereas most other works propose splitting it into bins that involve a loss of information. The kernel density is automatically adapted to the distribution of $Y$, as proposed in Subsection 4.2.

**A new approach to generate the target variable**   The target variable is not generated in the same way as the covariate. Once the covariates are generated from our generator models, we propose to adapt a wild-bootstrap technique with synthetic features (Wu (1986))[2]. The classical *Wild-Bootstrap* involves uniformly drawing a prediction error $\epsilon_k$ to generate a new $y_i^* := \widehat{y}_i + \epsilon_k v_i$, where $v_i$ is a random variable.

The generation of the target variable is performed as follows:

i) Train a Random Forest on the initial sample;

ii) For each seed $x_s \in \boldsymbol{x}_S$: predict target variable $\widehat{y}_s$ associated to $x_s$ with the Random Forest; $\boldsymbol{x}_S$ being the original observations drawn at step 1.

iii) Obtain the distribution of the absolute residuals on the prediction of the seed: $y_s - \widehat{y}_s$ and draw a randomly residual $\epsilon_k$;

iv) Generate the noise in the *Wild-Bootstrap* $v_s \sim \mathcal{N}(0, \sigma)$, $\sigma$ being a parameter; $v_s$ can also be set to 1 to avoid adding noise to the residual.

v) Generate a new $y_s^*$ associated to the new synthetic $x_s^*$ as follows:

$$y_s^* := y_s + |\epsilon_k| v_s \times sign(\widehat{y}_s - \widehat{y}_s^*)$$

With $v_s = 1$, this form is close to the Wild Bootstrap version with the Rademacher distribution. The idea behind this proposition is to consider taking into account i) the prediction error and ii) the impact of the synthetic covariate on the target variable.

The choice of using a Random Forest is justified by its good predictive performance, its non-parametric nature, and the possibility of getting an error distribution for a given target variable value. It represents the second step of GOLIATH. Other interesting methods to generate $Y$ are proposed in Appendix B.2 (but giving lower numerical performances on our illustration, as presented in Figure 5a). The algorithm is detailed in Appendix B.2.

---

[2]The kernel regression (Nadaraya-Watson estimator) was also tested but not selected because its high computation time and poor performance

# 6 Application in Imbalanced Regression

To evaluate the performance of GOLIATH, we focus on the imbalanced regression context because of the natural capacity of the form (1) to handle continuous variables.

For each dataset, we construct a test sample as 10-30% of the initial dataset and use the weighting previously defined: the inverse of the kernel density estimate for the target variable $Y$. We construct an artificial imbalanced dataset from the remaining sample. More details on the protocol are given in Appendix C.2. One of the objectives of our approach is to obtain better performance than the imbalanced train dataset. We compare our results to existing methods to deal with imbalanced regression from the *UBL* R-package, Branco et al. (2016a), classical oversampling, SMOTE, Gaussian Noise, SMOGN, WERCS and ADASYN from the python-package *ImbalancedLearningRegression* (Wu et al. (2023)). These techniques are used with their automatic relevance function and the same parameters as GOLIATH if any, in particular $k$ for SMOTE and *pert* for the Gaussian Noise.

To avoid sampling effects and obtain a distribution of prediction errors we ran 10 train-test datasets. In the same way, to avoid getting results dependent on some learning algorithms we use 10 models from the *autoML of the H2O R-package* LeDell & Poirier (2020) among the following algorithms: Distributed Random Forest, Extremely Randomized Trees, Generalized Linear Model with regularization, Gradient Boosting Model, Extreme Gradient Boosting and a Fully-connected multi-layer artificial neural network. We present here the aggregated results of the models, a more detailed analysis is available in Appendix C and D.

We then compute the following metrics: RMSE and MAE and weighted-RMSE with our weighting function giving more importance to the rare values. Since the test sample is balanced on the target variable, we considered here the RMSE and MAE metrics as relevant to provide an overview of the average error across the whole target variable.

REMARK **5.** *The basic methods (Gaussian Noise or SMOTE) are different from the UBL version because i) the generation of the target variable $Y$ is realized with wild bootstrap and considers the new synthetic attributes and ii) the weights $\omega$ is defined for all samples while UBL uses a relevant function that divides the dataset into rare and frequent sets. Like the ROSE and Gaussian Noise algorithms, GOLIATH takes into account a parameter tuning the level noise for perturbation approaches (description in Appendix A). It is also possible to use a clustering (Gaussian Mixture Model) in GOLIATH in order to apply a generation by cluster. All datasets provided by GOLIATH in the applications were provided with the "mix" mode. Note that the ROSE algorithm did not exist for imbalanced regression.*

## 6.1 Illustrative Application

In order to get a reference for predictive performance, we chose a balanced dataset from which we build an imbalanced train dataset. The dataset, named *SML2010 Data Set* is available on the Machine Learning repository UCI[3]. It is composed of 24 numeric attributes and 4137 instances. The target variable is the indoor temperature (we construct a unique target variable as the mean temperature of dinning-room and the temperature of the room). We train, with the autoML, the following train dataset:

- Reference values: Full sample (*FTrain*), Imbalanced (*Imb*)
- Benchmark: UBL-Oversampling (*UBL-OS*), UBL-SMOTE for regression (*UBL-SMOTE*), UBL-Gaussian Noise for regression (*UBL-GN*), UBL-SMOGN for regression (*UBL-SMOGN*), UBL-WERCS (*UBL-WERCS*), IRL-ADASYN (*IRL-OS*)
- GOLIATH (step 1): Oversampling (*G-OS*), Gaussian Noise (*G-GN*), Gaussian Noise with GMM-clustering (*G-GNwCl*), ROSE (*G-ROSE*), ROSE with a GMM-clustering (*G-ROSEwCl*), SMOTE (*G-SMOTE*), Non classical Smoothed Bootsrap with contraints on the distributions (*G-NCSB*), Classical Smoothed Bootsrap (*G-CSB*), Nearest Neighbors Smoothed Bootstrap, with Beta distribution, (*G-NNSB*), Nearest Neighbors Smoothed Bootstrap with k-NN weights proportionates to the distance from the seed (*G-NNSBw*), Extended Nearest Neighbors Smoothed Bootstrap (*G-eNNSB*).

---

[3]https://archive.ics.uci.edu/ml/datasets/SML2010

Figure 2a shows the results for RMSE on the test sample. The weighted-RMSE and MAE metrics are shown in Appendix 8 and present similar results. We can observe that the GOLIATH algorithm presents an RMSE smaller than the imbalanced sample and than the benchmark techniques, whatever the generators. The GOLIATH-oversampling is comparable to the UBL-oversampling which confirms, on this dataset, the relevance of the weighting. The Non-Classical Smoothed Bootstrap (NCSB) is as efficient as the Classical Smoothed Bootstrap (CSB, ROSE, GN). However, it provides realistic values for the variables. We can see some examples of inconsistency in Appendix 5b. The results show that the clustering seems to improve the performance. The Nearest Neighbors Smoothed Bootstrap (NNSB) and Extended Nearest Neighbors Smoothed Bootstrap(eNNSB) outperform the original SMOTE. It is important to note that the different parameters (the number of nearest neighbors for interpolation techniques and the level of noise for perturbation techniques) are arbitrary and not optimized here. The heatmap in Figure 2b shows the robustness of the methods with their rank by run with respect to the RMSE. We can observe, based on the mean and standard deviation of the rank, that the Classic Smoothed Bootstrap, the Non-Classical Smoothed Bootstrap, and the Nearest Neighbors Smoothed Bootstrap are the best approaches here.

| Run | 1 | 2 | 3 | 4 | 5 | 6 | 7 | 8 | 9 | 10 | mean | sd |
|---|---|---|---|---|---|---|---|---|---|---|---|---|
| RMSE | BASELINE | | | | | | | | | | | |
| Ftrain | 0,8 | 0,7 | 0,7 | 0,9 | 0,8 | 0,7 | 0,8 | 0,8 | 0,9 | 0,9 | 0,8 | 0,1 |
| Imb | 2,19 | 1,79 | 1,48 | 1,44 | 1,85 | 1,85 | 2,08 | 2,02 | 1,89 | 1,59 | 1,82 | 0,25 |
| RMSE | BENCHMARK | | | | | | | | | | | |
| UBL-OS | 2,21 | 1,96 | 1,69 | 1,82 | 2,12 | 1,83 | 2,38 | 2,22 | 2,07 | 1,85 | 2,02 | 0,22 |
| UBL-SMOTE | 1,87 | 1,85 | 1,60 | 1,53 | 1,68 | 1,97 | 2,12 | 2,13 | 1,85 | 1,77 | 1,84 | 0,20 |
| UBL-GN | 2,24 | 1,98 | 1,67 | 1,60 | 1,92 | 1,61 | 2,10 | 2,01 | 1,81 | 1,67 | 1,86 | 0,22 |
| UBL-SMOGN | 2,17 | 1,81 | 1,74 | 1,67 | 1,92 | 1,96 | 2,20 | 2,17 | 1,94 | 1,52 | 1,91 | 0,23 |
| UBL-WERCS | 2,43 | 1,87 | 1,80 | 1,59 | 2,08 | 1,64 | 2,24 | 2,09 | 2,13 | 1,86 | 1,97 | 0,27 |
| IRL-ADASYN | 2,63 | 2,26 | 2,17 | 2,03 | 2,58 | 2,31 | 2,50 | 2,57 | 2,44 | 2,33 | 2,38 | 0,20 |
| RMSE | GOLIATH | | | | | | | | | | | |
| G-OS | 2,11 | 1,83 | 1,67 | 1,71 | 1,86 | 2,10 | 2,16 | 2,21 | 2,13 | 1,93 | 1,97 | 0,20 |
| G-GN | 2,00 | 1,82 | 1,22 | 1,48 | 1,76 | 1,49 | 1,57 | 1,92 | 1,61 | 1,66 | 1,65 | 0,23 |
| G-GNwCl | 1,65 | 1,61 | 1,56 | 1,37 | 1,54 | 1,99 | 1,91 | 1,78 | 1,65 | 1,48 | 1,65 | 0,19 |
| G-ROSE | 1,60 | 1,73 | 1,69 | 1,35 | 1,57 | 1,47 | 1,68 | 1,93 | 1,71 | 1,60 | 1,63 | 0,16 |
| G-ROSEwCl | 1,84 | 1,60 | 1,29 | 1,48 | 1,43 | 1,79 | 1,77 | 2,01 | 1,77 | 1,49 | 1,65 | 0,22 |
| G-SMOTE | 1,95 | 1,80 | 1,61 | 1,54 | 1,63 | 1,73 | 2,25 | 2,67 | 1,30 | 1,93 | 1,84 | 0,39 |
| G-NCSB | 2,19 | 1,77 | 1,54 | 1,33 | 1,44 | 1,63 | 1,70 | 2,05 | 1,78 | 1,66 | 1,71 | 0,26 |
| G-CSB | 1,69 | 1,68 | 1,55 | 1,30 | 1,50 | 1,82 | 1,78 | 2,00 | 1,72 | 1,52 | 1,66 | 0,20 |
| G-NNSB | 1,67 | 1,56 | 1,57 | 1,30 | 1,77 | 1,54 | 1,79 | 1,29 | 1,45 | 1,60 | 1,55 | 0,17 |
| G-NNSBw | 1,62 | 1,97 | 1,46 | 1,46 | 1,64 | 1,83 | 1,54 | 1,90 | 1,64 | 1,74 | 1,68 | 0,18 |
| G-eNNSB | 2,20 | 1,79 | 1,45 | 1,15 | 1,29 | 1,63 | 1,73 | 1,86 | 1,46 | 1,57 | 1,61 | 0,30 |

| Run | 1 | 2 | 3 | 4 | 5 | 6 | 7 | 8 | 9 | 10 | mean | sd |
|---|---|---|---|---|---|---|---|---|---|---|---|---|
| RMSE Rank | BASELINE | | | | | | | | | | | |
| Imb | 13 | 8 | 5 | 7 | 12 | 13 | 10 | 10 | 13 | 6 | 10 | 3 |
| RMSE Rank | BENCHMARK | | | | | | | | | | | |
| UBL-OS | 15 | 15 | 15 | 17 | 17 | 12 | 17 | 16 | 15 | 14 | 15 | 2 |
| UBL-SMOTE | 7 | 13 | 10 | 11 | 9 | 15 | 12 | 13 | 12 | 13 | 12 | 2 |
| UBL-GN | 16 | 17 | 13 | 14 | 15 | 4 | 11 | 9 | 11 | 11 | 12 | 4 |
| UBL-SMOGN | 11 | 10 | 16 | 15 | 15 | 14 | 14 | 14 | 14 | 4 | 13 | 4 |
| UBL-WERCS | 17 | 14 | 17 | 13 | 16 | 7 | 15 | 12 | 17 | 15 | 14 | 3 |
| IRL-ADASYN | 18 | 18 | 18 | 18 | 18 | 18 | 18 | 17 | 18 | 18 | 18 | 0 |
| RMSE Rank | GOLIATH | | | | | | | | | | | |
| G-OS | 10 | 12 | 13 | 16 | 13 | 17 | 13 | 15 | 17 | 17 | 14 | 2 |
| G-GN | 9 | 11 | 1 | 10 | 10 | 2 | 2 | 5 | 4 | 10 | 6 | 4 |
| G-GNwCl | 3 | 3 | 8 | 6 | 5 | 16 | 9 | 2 | 6 | 1 | 6 | 4 |
| G-ROSE | 1 | 5 | 15 | 5 | 6 | 1 | 3 | 6 | 7 | 8 | 6 | 4 |
| G-ROSEwCl | 6 | 2 | 2 | 10 | 2 | 9 | 6 | 9 | 9 | 2 | 6 | 3 |
| G-SMOTE | 8 | 9 | 11 | 12 | 7 | 8 | 16 | 18 | 1 | 17 | 11 | 5 |
| G-NCSB | 13 | 6 | 6 | 4 | 3 | 6 | 4 | 11 | 10 | 10 | 7 | 3 |
| G-CSB | 5 | 4 | 7 | 3 | 4 | 10 | 7 | 7 | 8 | 4 | 6 | 2 |
| G-NNSB | 4 | 1 | 9 | 3 | 11 | 3 | 8 | 1 | 2 | 8 | 5 | 4 |
| G-NNSBw | 2 | 16 | 4 | 8 | 8 | 12 | 1 | 4 | 5 | 12 | 7 | 5 |
| G-eNNSB | 14 | 8 | 3 | 1 | 1 | 6 | 5 | 3 | 3 | 5 | 5 | 4 |

(a) RMSE Heatmap        (b) RMSE Ranking

Figure 2: Numerical simulation. RMSE values and ranking for the full sample, the imbalanced sample, 6 competitors, and GOLIATH associated with 11 different generating methods

The RMSE-rank represents the ranking of approaches according to the RMSE for a run: rank 1 corresponds to the training dataset that offers the smallest RMSE on the test sample. We also compared the results using the R-package *IRon: Solving Imbalanced Regression Tasks* [4], a useful and relevant package specific to Imbalanced Regression based on Ribeiro & Moniz (2020). The results in Appendix C.6 demonstrate that GOLIATH outperforms UBL approaches, even when considering their performance metrics (weighted MSE, weighted MAE, and SERA).

## 6.2 Imbalanced Regression Applications

We test our approach on several real data set from a repository provided as a benchmark for imbalanced regression problems[5] and presented in Branco et al. (2019) (descriptions in Appendix D). Figures 3a and 3b present RMSE gain (wrt the imbalanced dataset) and the median of the RMSE ranking. We can observe on these datasets that the GOLIATH algorithm empirically outperforms the state-of-the-art techniques, especially the Non-Classical Smoothed Bootstrap and the Extended Nearest Neighbors Smoothed Bootstrap.

We can see on these several applications, with several runs, several learning algorithms, and several performance metrics that the GOLIATH approach seems relevant to deal with imbalanced regression. In general GOLIATH gives better results, especially when it is combined with and extended nearest neighbors smoothed bootstrap in its first step of covariates generation.

---

[4] https://cran.r-project.org/web/packages/IRon/IRon.pdf
[5] https://paobranco.github.io/DataSets-IR/

| Dataset | NO2 | cpuSm | Boston | Bank8FM | Abalone |
|---|---|---|---|---|---|
| **RMSE gain** | BENCHMARK | | | | |
| UBL-OS | -3% | 2% | -7% | 63% | 0% |
| UBL-SMOTE | -10% | -2% | -12% | 27% | -4% |
| UBL-GN | -8% | -5% | -7% | 57% | -3% |
| UBL-SMOGN | -10% | -3% | -10% | 29% | -3% |
| UBL-WERCS | -4% | 3% | -1% | 57% | -1% |
| IRL-ADASYN | 10% | 22% | -1% | 69% | NA |
| **RMSE gain** | GOLIATH | | | | |
| G-OS | -1% | -2% | -3% | 57% | -3% |
| G-GN | -11% | -18% | -21% | 0% | -9% |
| G-GNwCl | -10% | -18% | -14% | 13% | -6% |
| G-ROSE | -9% | -17% | -17% | -6% | -9% |
| G-ROSEwCl | -9% | -19% | -21% | 0% | -8% |
| G-SMOTE | -6% | -2% | -6% | 54% | -11% |
| G-NCSB | -9% | -23% | -23% | -6% | -9% |
| G-CSB | -12% | -20% | -17% | 0% | -9% |
| G-NNSB | -13% | -11% | -12% | 31% | -14% |
| G-NNSBw | -15% | -11% | -14% | 35% | -13% |
| G-eNNSB | -15% | -7% | -19% | -29% | -21% |

(a) RMSE-gain

| Dataset | NO2 | cpuSm | Boston | Bank8FM | Abalone |
|---|---|---|---|---|---|
| **RMSE rank** | BENCHMARK | | | | |
| UBL-OS | 14,0 | 14,5 | 13,0 | 17,0 | 16,5 |
| UBL-SMOTE | 10,0 | 11,5 | 11,0 | 11,0 | 12,0 |
| UBL-GN | 10,5 | 9,5 | 11,5 | 15,5 | 13,0 |
| UBL-SMOGN | 8,0 | 12,0 | 11,5 | 12,0 | 12,0 |
| UBL-WERCS | 14,0 | 16,0 | 15,5 | 16,5 | 15,0 |
| IRL-ADASYN | 18,0 | 18,0 | 16,0 | 18,0 | NA |
| **RMSE rank** | GOLIATH | | | | |
| G-OS | 16,0 | 14,0 | 14,5 | 15,5 | 13,5 |
| G-GN | 6,5 | 5,5 | 2,5 | 6,0 | 6,5 |
| G-GNwCl | 8,5 | 5,0 | 6,5 | 8,0 | 9,5 |
| G-ROSE | 8,5 | 4,0 | 4,0 | 4,5 | 7,0 |
| G-ROSEwCl | 10,0 | 4,0 | 3,0 | 8,0 | 8,0 |
| G-SMOTE | 13,5 | 11,5 | 12,5 | 15,5 | 4,5 |
| G-NCSB | 8,0 | 2,0 | 4,0 | 5,0 | 5,5 |
| G-CSB | 6,5 | 3,5 | 6,0 | 5,0 | 6,0 |
| G-NNSB | 6,0 | 8,5 | 8,0 | 12,0 | 2,5 |
| G-NNSBw | 3,0 | 7,0 | 6,5 | 12,0 | 3,5 |
| G-eNNSB | 2,0 | 10,0 | 5,5 | 1,5 | 1,0 |

(b) Median of the RMSE-rank

Figure 3: Datasets. RMSE values and ranking for 6 competitors, and GOLIATH associated with 11 different generating methods

## 6.3 Discussion

Based on all our numerical evidences, we strongly recommend the use of GOLIATH when step 1 combines a smoothed Bootsrap with nearest neighbors. The five versions of GOLIATH involving bootstrap procedures appear to be the best performing and most stable in terms of numerical results (both RMSE gain and rank).

# 7 Discussion and Perspectives

GOLIATH is an algorithm gathering two large families of synthetic data oversampling. Many methods can be rewritten as particular cases of it. This approach gets the advantage to obtain a general form for the generator which is based both on the theoretical foundations of kernel estimators and classical smoothed bootstrap techniques. It provides a general expression for the conditional density of the generator. The use of well-chosen kernels makes it possible to take into account the nature of the covariates: continuous, discrete, totally or partially bounded. Our approach generalizes the SMOTE algorithm by providing weights and flexible densities for interpolation. We also extend this technique to wider support than that of the observations by combining interpolation and perturbation approaches. Numerical applications in imbalanced regression models demonstrate that GOLIATH and its variants are very competitive, especially when the generator used in step 1 is the extended nearest neighbors smoothed bootstrap.

The weights $\omega_i$ (and $\pi_{j|i}$ in the interpolation case) offer a large flexibility. For instance, it is possible to handle classification tasks by conditioning with the minority class. We could deal with multi-class classification too. It is also possible to combine some extensions of SMOTE that propose to focus on specific samples in the synthetic data generation (as ADASYN) with a kernel approach in order to perform the methodology.

As a perspective, a natural extension of this work is to automate the choice of the kernel estimators, the weights, as well as some parameters according to the data. For example by defining a weights function for the nearest neighbor instead of defining the parameter $k$. Indeed, the parameter $k$ is sometimes unsuitable and we could suggest a dynamic weighting depending on the neighborhood. It is also possible to define a kernel according to the neighborhood into the same dataset. For instance, an interpolation approach is favored within clusters when neighboring points are considered close to the observation. On the other hand, a perturbation approach is preferred when the observation appears isolated. Finally, non-standard kernels enable handling specific distributions such as bounded or discrete ones.

We also could define $\omega_i$ in order to generate a target distribution as done in Stocksieker et al. (2023). Finally, the perturbation-based approaches, based on kernel density estimators, may find it challenging to accurately capture dependencies between variables. The interpolation approaches consider it but the generation is limited to the segments. The extended-SMOTE proposes a first solution. GOLIATH proposes also an innovative method to generate $Y$ based on the generated $X$, regardless of the generator used. It would also be interesting and potentially effective to use multiple generators and capitalize on the strengths of each. The generators could be applied locally based on the data characteristics.

Another research direction would be to better consider i) correlations between variables while respecting their definition domain and ii) mixed data. Finally, it could be interesting to test GOLIATH on image datasets, by combining it with a deep-learning model (Deep Imbalanced Regression framework).

## Acknowledgment

The authors thank the reviewers for their useful and relevant comments that significantly improved the manuscript. S. Stocksieker would like to acknowledge the support of the Research Chair DIALog under the aegis of the Risk Foundation, a joint initiative by *CNP Assurance*. D. Pommeret would like to acknowledge the support received from the Research Chair ACTIONS under the aegis of the Risk Foundation, an initiative by BNP Paribas Cardif and the Institute of Actuaries of France.

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
