# A    Summary of Existing Methods

## A.1    Summary of the Original Methods

**Randomly Over Sampling Examples (ROSE):**    The idea of ROSE algorithm (Menardi & Torelli (2014)) for classification can be summarized as follows:

1. Select $y^* = \mathcal{Y}_j, j = 0, 1$ with probability $1/\pi_j$, $\pi_j = 1/2$ for binary classification.

2. Draw uniformly a seed $(\boldsymbol{x_i}, y_i)$ such that $y_i = y^*$, with probability $p_i = 1/n_j$, $n_j < n$ being the size of $\mathcal{Y}_j$

3. Generate a sample $\boldsymbol{x}^*$ from $K_{H_j}(\cdot, \boldsymbol{x_i})$, where $K_{H_j}$ is a probability distribution centered at $\boldsymbol{x_i}$ and with covariance matrix $H_j$.

The authors consider Gaussian Kernels for $K$ with diagonal smoothing matrices $H_j = diag(h_1^j, \cdots, h_p^j)$. Thus, the generations of new samples from the class $\mathcal{Y}_j$ correspond to data from the kernel density estimate $f(\mathbf{x}|\mathcal{Y_j})$. Under the assumption that the true conditional density underlying the data follows a Gaussian distribution, they suggest using the proposition of Bowman & Azzalini (1999):
$h_q^{(j)} = (4/((d + 2)n)^{1/(d+4)} \widehat{\sigma_q^{(j)}}$ $(q = 1, \cdots, p; j = 0, 1)$, where $\widehat{\sigma_q^{(j)}}$ is the sample estimate of the standard deviation of the q-th dimension of the observations belonging to the class $\mathcal{Y}_j$.

**Gaussian Noise (GN):**    The idea of GN algorithm for classification (Lee & Sauchi (2000)), can be summarized as follows:

1. Choose $m$ being the number of replicates of the training data from the imbalanced class that is when $y = y_{min}$.

2. For each $\boldsymbol{x_i}$ from the imbalanced class, generate $m$ noisy replicates $\boldsymbol{x}^*$ of the form $(\boldsymbol{x} + \boldsymbol{\epsilon}, y_{min}), \boldsymbol{\epsilon} = (\epsilon_1, \cdots, \epsilon_p)$ where $\epsilon_j, j = 1, \cdots, p$ is a Gaussian noise, i.e. $x_q^* = x_q + \mathcal{N}(0, \widehat{\sigma}_q \times \sigma_{noise})$, where $x_q$ is the feature value $q$ from the original observation, and $\sigma_{noise}$ is defined by the user.

**Synthetic Minority Oversampling Technique (SMOTE):**    The idea of the SMOTE algorithm (Chawla et al. (2002)) can be summarized as follows:

1. Compute the $k-$ nearest neighbors for each minority sample

2. Do a loop on the minority class samples: generate $N$ synthetic samples for each seed $x_s$ as follows:

    (a) Choose one of the $k-$ nearest neighbors of $x_s$, say $x_j(s)$.
    (b) Compute $\lambda$ as a random number between 0 and 1.
    (c) Create a new synthetic sample $x^*$ defined as: $x^* := \lambda x_s + (1 - \lambda)(x_j(s) - x_s)$.

## A.2    Original Methods within the GOLIATH Form

Table 1 summarizes the original perturbation approaches ROSE and GN and the original interpolation approaches SMOTE as the form 1. $n$ represents here the number of samples in the minority class.

Table 1: Summary of the parametrization of the original methods within the GOLIATH form

| Generator | $\mathcal{I}$ | $\omega_i$ | $K_i(x)$ | Precision |
|---|---|---|---|---|
| ROSE | $[1, n]$ | $1/n$ | $\frac{1}{\|H_n\|^{1/2}} K(H_n^{-1/2}(\boldsymbol{x} - \boldsymbol{x}_{i.})))$ | $K(\cdot)$ the multivariate Gaussian kernel, $H_n$ the bandwidth matrix proposed by Bowman & Azzalini (1999) |
| GN | $[1, n]$ | $1/n$ | $\frac{1}{\|H_n\|^{1/2}} K(H_n^{-1/2}(\boldsymbol{x} - \boldsymbol{x}_{i.})))$ | $K(\cdot)$ the multivariate Gaussian kernel, $H_n$ a diagonal matrix with a fraction of the empirical standard deviations |
| SMOTE | $[1, n]$ | $1/n$ | $\frac{1}{k} \sum_{\ell=1}^{k} \frac{\mathbb{1}_{[0,\boldsymbol{x}_{i.}(\ell)-\boldsymbol{x}_{i.}]}(\boldsymbol{x}^* - \boldsymbol{x}_{i.})}{\|\boldsymbol{x}_{i.}(\ell) - \boldsymbol{x}_{i.}\|}$ | $K(\cdot)$ a Uniform Mixture Model with $k$ components having the same weight, depending of the $k$-nn of $\boldsymbol{x}_{i.}$ |

### A.3 Other Existing Methods within the GOLIATH Form

Here we present some extensions of SMOTE that can be written within a simplified GOLIATH form 1. These methods are applied in Imbalanced classification. $n$ represents the number of samples in the minority class. We note by $K_i^{SMOTE}(x)$ the SMOTE kernel defined on Table 1 $\frac{1}{k} \sum_{\ell=1}^{k} \frac{\mathbb{1}_{[0,\boldsymbol{x}_{i.}(\ell)-\boldsymbol{x}_{i.}]}(\boldsymbol{x}^* - \boldsymbol{x}_{i.})}{\|\boldsymbol{x}_{i.}(\ell) - \boldsymbol{x}_{i.}\|}$. We note by $n_{\mathcal{I}}$ the number f instances which respects the condition of $\mathcal{I}$. Note that it is possible to write within the GOLIATH form 1 the under-sampling methods as well as over-sampling methods and hybridization.

## B Differences with *Utility-Based Learning* Approach

Here we present some differences with the *Utility-Based Learning* approach proposed in Branco et al. (2016a) and associated works, for example Torgo et al. (2013), Branco et al. (2017), Branco et al. (2019), Ribeiro & Moniz (2020). This is the first and main solution in the Imbalanced Regression. These works are considered references for Imbalanced Regression Learning.

Table 3: Differences between the UBL approach and the GOLIATH approach

| **Characteristic** | **UBL approach** | **GOLIATH approach** |
|---|---|---|
| Rebalancing | Using a binarization of $y$ | Using the continuous distribution of $y$ |
| Flexibility | Limited to some adaptations of imbalanced classification methods | Adaptation of imbalanced classification methods (SMOTE family) and kernel-based methods, for continuous distributions |
| Parametrization of the weights | Based on a relevance function that binarizes the imbalance problem: automatic or defined by the user | Naturally based on the inverse of the kernel density estimate of $y$ or defined by the user |

**Remark on the definition of the weights:** note that the automatic relevance function with the UBL package does not work for every dataset (error message). We are often asked to define the relevance function which can be difficult for the user. The GOLIATH is pretty reliable because of the simplicity of the approach. However, it is important to define relevant safeguards for the definition of the weight, especially for the very

Table 2: Summary of the parametrization of some extensions SMOTE methods within the GOLIATH formulation

| Generator | $\mathcal{I}$ | $\omega_i$ | $K_i(x)$ | Precision |
|---|---|---|---|---|
| Borderline-SMOTE (Han et al. (2005)) | $x_i$ such as $\frac{k^{mj}(x_i)}{k} \geq 50\%$ | $1/n_\mathcal{I}$ | $K_i^{SMOTE}(x)$ | $k^{mj}(x_i)$ represents the number of majority examples among the $k$ nearest neighbors of $x_i$, $n_\mathcal{I} := \sum_i \mathbb{1}_{k^{mj}(x_i)/k \geq 50\%}$ |
| Safelevel-SMOTE (Bunkhumporn-pat et al. (2009)) | $x_i$ according to $SLR_{ij}$ and $SF_i$: cf below | $1/n_\mathcal{I}$ | adapted kernel: cf below | $SFR_{ij} := \frac{SL_i}{SL_{ij}} = \frac{SL(x_i)}{SL(x_j(i))}$ with $SL(x)$ the number of minority instances in k nearest neighbours for x |
| Safelevel-SMOTE | $x_i$ such as $SFR_{ij} = \infty \, \& \, SL_i \neq 0$ | $1/n_\mathcal{I}$ | $x$ (oversampling) | oversampling if $x_j(i)$ is considered as noise |
| Safelevel-SMOTE | $x_i$ such as $SFR_{ij} = 1 \, \& \, SL_i = SL_{ij}$ | $1/n_\mathcal{I}$ | $K_i^{SMOTE}(x)$ | SMOTE if $x_j(i)$ is considered as safe |
| Safelevel-SMOTE | $x_i$ such as $SFR_{ij} > 1$ | $1/n_\mathcal{I}$ | $K_i^{SMOTE}(x)$ with $\lambda \sim \mathcal{U}([0, 0.5])$ | SMOTE closer to $x_i$ if $x_j(i)$ is considered as not safe |
| Safelevel-SMOTE | $x_i$ such as $SFR_{ij} < 1$ | $1/n_\mathcal{I}$ | $K_i^{SMOTE}(x)$ with $\lambda \sim \mathcal{U}([0.5, 1])$ | SMOTE closer to $x_j(i)$ if $x_i$ is considered as not safe |
| ADASYN (He et al. (2008)) | $[1, n]$ | $\widehat{r_i} := \frac{r_i}{\sum_i r_i}$ and $r_i = \frac{\Delta_i}{k}$ | $K_i^{SMOTE}(x)$ | $\Delta_i$ represents the number of examples in the $k$ nearest neighbors of $x_i$ that belong to the majority class |
| Kernel-ADASYN (Tang & He (2015)) | $[1, n]$ | $\widehat{r_i} := kde(r_i)$ and $r_i = \frac{\Delta_i}{k}$ | $K_i^{SMOTE}(x)$ | evolution of the ADASYN algorithm by using a Gaussian kernel density estimate to normalize $r_i$ |
| SMOTE-TomekLink (Batista et al. (2004)) | $[1, n]$ | 1 if $x_i$ and $x^*$ do not form a Tomek link | $K_i^{SMOTE}(x)$ | This a acceptance-rejection method according to the *Tomek-link*(Tomek (1976)) applied after the generation of $x^*$ with SMOTE |
| SMOTE-ENN (Batista et al. (2004)) | $[1, n]$ | 1 if $y_i = y_j(i), j \in [1, k_{ENN}]$ | $K_i^{SMOTE}(x)$ | Similar to SMOTE-TomekLink but by using the rule of ENN (Wilson (1972)), $k_{ENN}$ represents the number of the NN considering for the ENN rule |
| Kmeans-SMOTE (Douzas et al. (2018)) | $x_i \in$ filtered clusters | sampling weight based on its minority density into the filtered cluster | $K_i^{SMOTE}(x)$ | SMOTE applied on the filtered clusters defined as imbalanced cluster |

extreme value which presents a very low probability and so a high value of its inverse. We suggest using a trimming sequence as a hyperparameter, as often proposed in non-parametric statistics inference.

**Remark on the computation time:** Both the UBL package and the GOLIATH algorithm are fast enough to generate a new sample: between 3 and 5 seconds for a dataset with about 500 rows. Note that, with the Non-Classical Smoothed Bootstrap, the estimation of the bandwidth parameter for a non-Gaussian distribution could take several minutes due to the package used, especially for a Binomial one.

**Remarks on the use of GOLIATH:**

- As described in the paper, GOLIATH proposes 3 modes of data generation: "synth" to obtain a complete synthetic dataset, "augment" to obtain the original dataset augmented with synthetic

observations, and "mix" which is a mixed approach: the original sample for the first occurrence of the seed drawn and a synthetic observation for the next.

- The automatic weights used are defined as the inverse of the kernel density estimate of the $y$ distribution with the "mix" mode and "synth" because a new dataset is built which is equivalent to realizing an oversampling and an undersampling. However, with the mod "augment", the weights are defined as the squared inverse $\frac{1}{\hat{f}^2}$ because we realize an oversampling and want to draw more extreme values.

- The parameters of GOLIATH depend on the generator used: the number of the nearest neighbors $k$, the tuning noise for Classical and Non-Classical Smoothed Bootstrap equivalent to $hmult$ for the ROSE algorithm, and $pert$ for the Gaussian Noise algorithm.

- It is possible to define another distance with the Nearest Neighbors Smoothed Bootsrap. GOLIATH computes the $k$-NN using a distance in the R-package $philentropy$ that proposes a large choice of distance.

- It is possible to use GOLIATH as a simple generator of data (non-supervised framework) with a $Y$ defined as null. This corresponds to the first step of GOLIATH. It is also possible to use GOLIATH in a classical (non-imbalanced) supervised framework (with a $Y$ non-null) to perform learning and prediction.

- The clustering is an option. If activated, it is possible to define a clustering based on the initial train dataset. It is also possible to define a clustering on $Y$ distribution in order to define the clusters according to the frequencies of $Y$ values, in the same idea as the UBL approach.

- The $m$ parameter represents the maximum number of components in the Gaussian Mixture Model used for the clustering. Thus, this algorithm seeks to optimize the clustering with a number of clusters between 1 and $m$. Obviously, it is possible to use another clustering algorithm as k-means.

## B.1 Important Required Packages

The GOLIATH algorithm uses the following R-packages:

- *Ake: Associated Kernel Estimations, used for the bandwidth parameter of the Binomial distributions* (`https://cran.r-project.org/web/packages/Ake/Ake.pdf`)

- *ks: Kernel Smoothing, used for the bandwidth parameter of the Gaussian distributions* `https://cran.r-project.org/web/packages/ks/ks.pdf`

- *np: Nonparametric Kernel Smoothing Methods for Mixed Data Types, used for the bandwidth parameter of the Beta, truncated Gaussian and Gamma distributions* `https://cran.r-project.org/web/packages/np/np.pdf`

- *kernelboot: Smoothed Bootstrap and Random Generation from Kernel Densities, used for the classical smoothed bootstrap* `https://cran.r-project.org/web/packages/kernelboot/kernelboot.pdf`

- *randomForest: Breiman and Cutler's Random Forests for Classification and Regression, used for the estimation of y for the new synthetic data* `https://cran.r-project.org/web/packages/randomForest/randomForest.pdf`

- *mclust: Gaussian Mixture Modelling for Model-Based Clustering, Classification, and Density Estimation, used for the GMM clustering* (`https://cran.r-project.org/web/packages/mclust/mclust.pdf`)

- *philentropy: Similarity and Distance Quantification Between Probability Functions, used for the computation of the $k-$ nearest neighbors in the Nearest Neighbors Smoothed Bootsrap* (`https://cran.r-project.org/web/packages/philentropy/philentropy.pdf`)

## B.2 GOLIATH algorithm

The GOLIATH algorithm is presented in Figure 4.

---

**Input** covariate X; target variable Y=null; mod="mix"; type; method-Y=1; clustering=F; seed s=null; components
GMM m=n-row(X); weights w=rep(1,n-row(X)); synthetic data sample size N=n-row(X); parameter p)
**Clustering**   // *Optional application of a clustering on the train*
clust = Cluster(X,Y,clustering)   // *clust=1 for all samples if clustering=F*
**Seed drawing**   // *weighted oversampling*
if s = null then s = draw(X,N,w)
**X generation**   // *Synthetic data generation for the covariates*
if m < n then synth = GMM(X,Y,m,N)
  else for each c in clust:
    if type = "CSB" then synth = G-CSB(X,N,w, s, p)
    else if type = "NCSB" then synth = G-NCSB(X,N,w, s, p)
    else if type = "NNSB" then synth = G-NNSB(X,N,w, s, p)
    else if type = "eNNSB" then synth = G-eNNSB(X,N,w, s, p)
    **Y generation**   // *Optional Synthetic Y generation*
    if Y <> null then synth[,Y] = G-Y(X,Y,synth,s,method-Y, p)
  end For
**Output**
if m<n or mod = "synth" then return synth
else if mod = "augment" then return (X,Y) + synth
else return mix((X,Y),synth,s)

---

Figure 4: GOLIATH algorithm

With the GOLIATH algorithm:

- The user can apply the generation by cluster with the option "clustering"

- The sampling weights are automatically deduced and applied

- The generation of synthetic data, potentially performed by cluster, depends on the mode chosen by the user:

  - CSB: Classical Smoothed Bootstrap
  - NCSB: Non-Classical Smoothed Bootstrap
  - NNSB: Nearest Neighbors Smoothed Bootstrap
  - eNNSB: Extended Nearest Neighbors Smoothed Bootstrap

- The target variable $Y$ is then generated conditionally on the synthetic data $x^*$.

- Applies a new method for constructing the final sample: the "mix" mode.

**Remark on the computation time:** Both the UBL package and the GOLIATH algorithm are fast enough to generate a new sample: between 3 and 5 seconds for a dataset with about 500 rows. Note that, with the Non-Classical Smoothed Bootstrap, the estimation of the bandwidth parameter for a non-Gaussian distribution could take several minutes due to the package used, especially for the binomial one.

We proposed to generate a synthetic target value $y^*$ associated to a new synthetic covariate $x^*$ using the wild bootstrap approach. This technique is well-known in the regression framework, especially in the presence of heteroskedasticity.

---

**Algorithm 1** GOLIATH algorithm for Y

---

**Input** *covariates* X; *target variable* Y; *synthetic X* synth, *drawn samples for the synthetic X* seed; method for the generation of y method_Y=1; *standard deviation for the Gaussian distribution of the noise in the Wild Bootstrap* sigma=0, *proportion of the perturbation for the Gaussian Noise method* pert
**RF prediction**          *// Prediction with the Random Forest algorithm*
model = RF(X,Y)          *// Training the model*
predSynth = predict(model, synth)          *// Prediction on the synthetic data*
predSeed = predict(model, X[seed])          *// Prediction on the seed data*
eps = Y[seed] - predSeed          *// Distribution of the prediction error*
**Y Generation**          *// generation of synthetic Y using the prediction error*
For i in synth
  if method_Y = 1 then
    eps = Y[seed][i] - DistribPredSeed[i]          *// Distribution of the absolute residuals on the prediction of the seed*
    v = Gaussian(0,sigma)          *// Noise in the wild bootstrap*
    synth[i,Y]=Y[seed][i]   +   abs(Random(eps,1))   *   v   *   sign(AveragePredSeed[i]-AveragePredSynth[i])

  else if method_Y = 0 then
    synth[i,Y]=Y[seed][i]
  else if method_Y = 2 then
    eps = AveragePredSynth[i] - AveragePredSeed[i]          *// difference between the average prediction*
    synth[i,Y]=Y[seed][i] + eps
  else if method_Y = 3 then
    eps = Y[seed][i] - DistribPredSeed[i]          *// Distribution of the absolute residuals on the prediction of the seed*
    v = Gaussian(0,sigma)          *// Noise in the wild bootstrap*
    h = kde(eps)          *// bandwidth parameter with a gaussian kernel*
    synth[i,Y]=Y[seed][i] + abs(Gaussian(0,h)) * v * sign(AveragePredSeed[i]-AveragePredSynth[i])
  else if method_Y = 4 then
    sig = standardDeviation(Y,w_Y)          *// weighted standard deviation of y*
    synth[i,Y]=Y[seed][i] + abs(Gaussian(0,sig * pert)) * sign(AveragePredSeed[i]-AveragePredSynth[i])
  else if method_Y = 5 then
    h = kde(Y,w_Y)          *// bandwidth parameter with a gaussian kernel for the weighted kernel density estimate of y*
    synth[i,Y]=Y[seed][i] + abs(Gaussian(0,h)) * sign(AveragePredSeed[i]-AveragePredSynth[i])
end For
return Synth[,Y]

---

**Details of the methods:**

- **method 0:** the synthetic $y^*$ is equal to the $y$ seed i.e. the value of $y$ associated to the $X$ used for generating the new synthetic $x^*$

- **method 1 (by default):** the synthetic $y^*$ is generated by using an adapted wild bootstrap technique, as described in the paper. Note that with a sigma parameter equal to 0, we obtain the residual resampling technique: this is the technique used by default.

- **method 2:** the synthetic $y^*$ is generated by adding to the $y$ seed the difference between the seed prediction and the synthetic prediction. The idea is to shift the $y$ seed with the difference of the predictions representing the impact of the synthetic $x^*$ in according to the seed $x$

- **method 3:** the synthetic $y^*$ is generated by using a classical smoothed bootstrap on the distribution of the error prediction. This is a "smoothing" version of the method 1.

- **method 4:** the synthetic $y^*$ is generated by using an adapted Gaussian Noise used a weighted standard deviation of $y$ and the sign of the difference between the predictions of $y$ and $y^*$

- **method 5:** the synthetic $y^*$ is generated by using a classical smoothed bootstrap and the sign of the difference between the predictions of $y$ and $y^*$.

As detailed in the paper, method 1 for the generation of $y$ proposes to use an adapted wild bootstrap. The differences with a classical wild bootstrap are:

- GOLIATH draw belongs to the distribution of the error prediction for the same $y$ while the Wild Bootstrap draw belongs to the distribution of the error prediction on the whole training dataset.

- GOLIATH generates a $y$ value for a new $x^*$ sample while the Wild Bootstrap generates a $y$ value for an existing $x$ drawn in the training dataset.

- To consider the previous item, GOLIATH suggests using the sign between the average prediction of $y$ associated with $x^*$ and associated with $x$. This represents the impact of generating new synthetic data $x^*$ from a seed $x$.

Figure 5a shows the difference in the RMSE with the different methods. These results are obtained on the illustrative application with a Classical Smoothed Bootstrap.

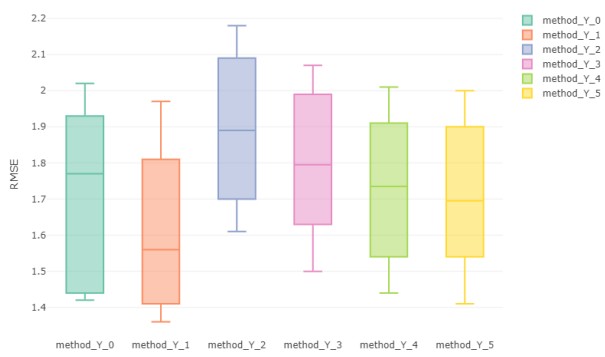

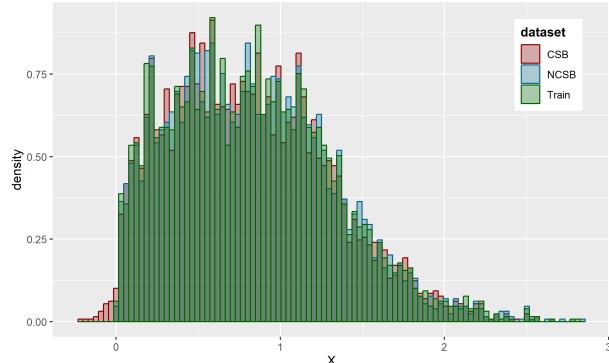

(a) Boxplots of RMSE obtained for different generation methods of $y$, for the same values of synthetic data $x$ obtained with a Classical Smoothed Bootstrap

(b) Illustration of a Classical Smoothed Bootstrap vs Non-Classical Smoothed Bootstrap

Figure 5: Illustrations of GOLIATH

### B.3   Illustration of the generators

Figure 5b represents the distributions of the positive variable *WholeWeight* in the dataset *Abalone*. As we can see, the Classical Smoothed Bootstrap can generate synthetic data beyond the $X$ support. Indeed, a Classical Smoothed Bootstrap is based on a classical kernel which is symmetric distribution. For a given point (0 for example) there is as much chance to generate a smaller value (negative) as a larger one (positive). For a positive asymmetric distribution, this can lead to obtaining outliers: as negative values. We observe that the Non-Classical Smoothed Bootstrap, using a Gamma Kernel, generates proper values of $x$.

## C   Complementary Results for Illustrative Application

**Remark:**   A decomposition by model of the stacked RMSE is not possible because the 10 models are not the same for each run.

### C.1   Dataset details

The dataset used in the illustrative application is *SML2010 Data Set* from the Machine Learning repository UCI (`https://archive.ics.uci.edu/ml/datasets/SML2010`). It is composed of 24 numeric attributes and

4137 instances. The target variable is the indoor temperature (we construct a unique target variable as the mean temperature of dinning-room and the temperature of the room). Figure 6 gives the histograms for all covariates $X$ and the target variable $y$. It can be observed that some variables (lighting and wind for instance) are bounded because of their positivity.

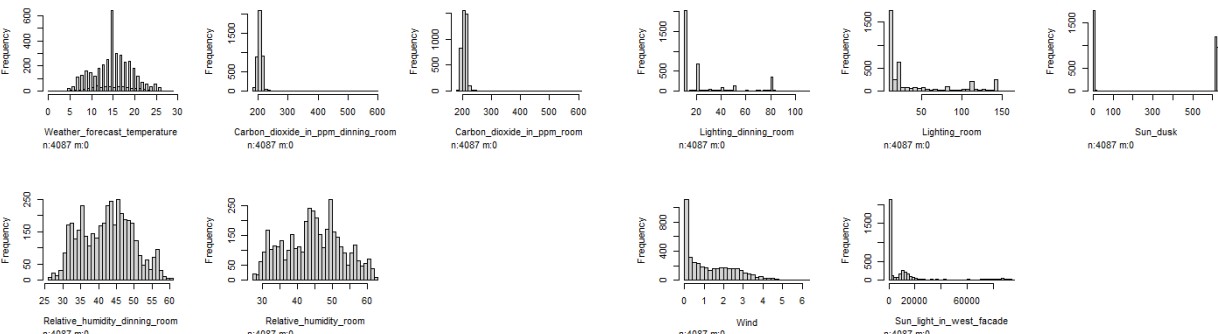

(a) Histograms of the covariates $x$ in the original dataset (b) Histograms of the covariates $x$ in the original dataset

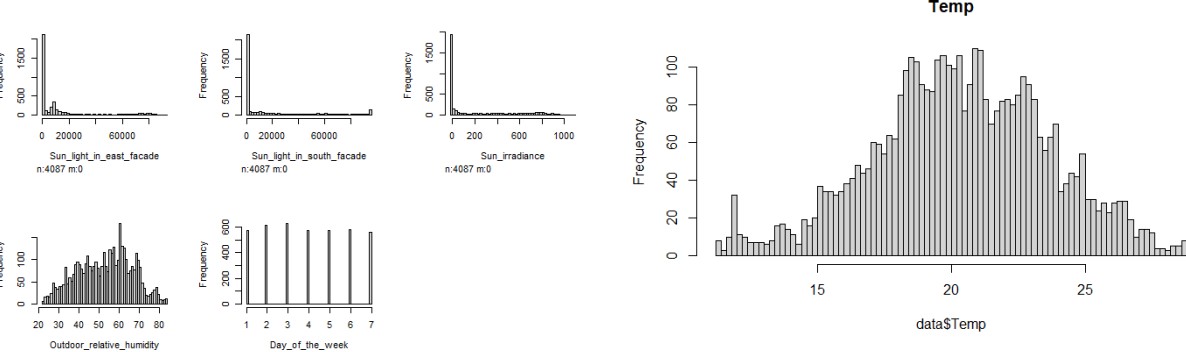

(c) Histograms of the covariates $x$ in the original dataset

(d) Histograms of the target variable $Y$ in the original dataset

Figure 6: Histograms of $X$ and $y$ in the illustrative application dataset

Table 4: Descriptive statistics of the dataset

| Variable | Min. | 1st Qu. | Median | Mean | 3rd Qu. | Max. |
|---|---|---|---|---|---|---|
| Weather_forecast_temperature | 0 | 12 | 15 | 15.1 | 18 | 29 |
| Carbon_dioxide_in_ppm_dinning_room | 187.34 | 200.33 | 205.22 | 206.72 | 210.07 | 594.39 |
| Carbon_dioxide_in_ppm_room | 188.91 | 201.83 | 208.97 | 209.74 | 212.37 | 609.24 |
| Relative_humidity_dinning_room | 26.17 | 36.02 | 42.73 | 42.31 | 47.5 | 60.96 |
| Relative_humidity_room | 27.26 | 38.33 | 44.71 | 44.46 | 50.2 | 62.59 |
| Lighting_dinning_room | 10.74 | 11.56 | 14.33 | 29.11 | 40.75 | 111.8 |
| Lighting_room | 11.33 | 13.51 | 22.21 | 42.56 | 55.28 | 162.96 |
| Rain | 0 | 0 | 0 | 0.03 | 0 | 1 |
| Sun_dusk | 0.61 | 0.65 | 612.95 | 335.72 | 619.76 | 625 |
| Wind | 0 | 0.17 | 0.96 | 1.3 | 2.23 | 6.32 |
| Sun_light_in_west_facade | 0 | 0 | 831.49 | 14876.48 | 14691.85 | 95278.4 |
| Sun_light_in_east_facade | 0 | 0 | 1125.38 | 13680.87 | 13108.25 | 92367.5 |
| Sun_light_in_south_facade | 0 | 0 | 716.8 | 20028.33 | 34069.8 | 95704.4 |
| Sun_irradiance | -4.16 | -3.25 | 12.22 | 234.14 | 488.37 | 1094.66 |
| Outdoor_relative_humidity | 22.25 | 42.46 | 54.38 | 53.07 | 62.89 | 83.81 |
| Day_of_the_week | 1 | 2 | 4 | 3.96 | 6 | 7 |
| Temp | 11.21 | 18.26 | 20.31 | 20.33 | 22.66 | 28.73 |

## C.2 Protocol

The protocol for the experiments on the illustrative application can be summarized as follows:

1. Define *test_prop* the desired proportion of the test dataset: 10% here

2. Define $N\_sample$ the number of the runs i.e. the desired train-test set: 10 here

3. Define the proportion of the imbalanced dataset $imb\_prop$: 10% here.

4. Construct $N\_sample$ train-test set: repeat the following instructions:

   - draw a test sample with a size $size(data) \times test\_prop$,

   - draw $size(data - test) \times imb\_prop$ from the remaining dataset with weights squared in order to get slightly more rare observations: on the side here. An illustration is given on Figure 7a.

5. Generate the new train datasets with the different methods. The generation is based on a weighting function. In Figure 7b, we compare

   - the histogram of the target variable $Y$

   - the weights obtained with the inverse of the kernel density estimate of the $y$ distributions: giving a distribution $y$ approximately uniform: "1/f"

   - the weights obtained with the squared inverse of the kernel density estimate of the $y$ distributions: giving a distribution $y$ inverse to that of $y$ in the initial sample: "1/f²"

   - the weights obtained with the UBL approach: "UBL"

   - the weights obtained with the method proposed in Steininger et al. (2021): "1-a.f", with $a = 8$

6. Predict the test value according to the new train datasets

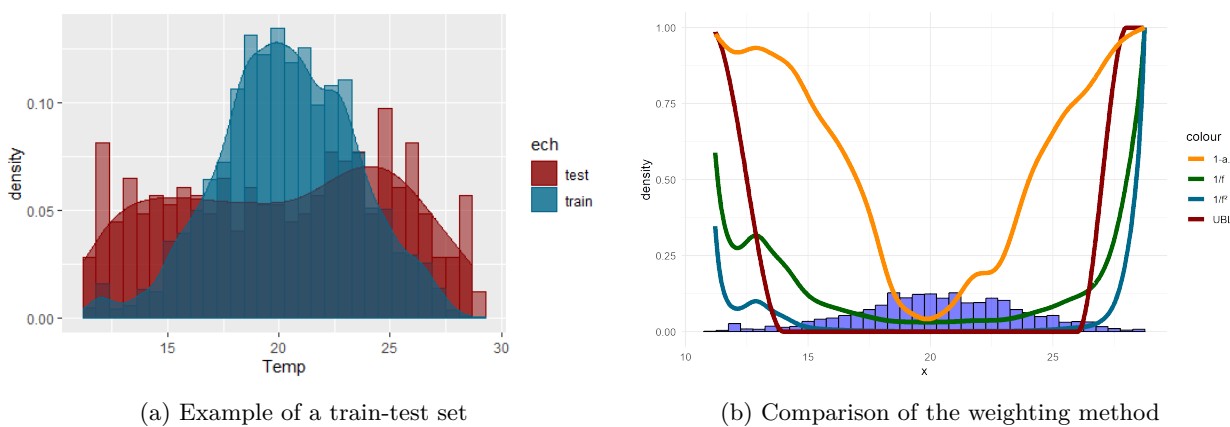

(a) Example of a train-test set          (b) Comparison of the weighting method

Figure 7: Illustration of the protocol

## C.3   Predictive performance metrics

The results for the MAE are similar to the RMSE results. The weighted-RMSE is still better with GOLIATH algorithm than UBL approach but GOLIATH-SMOTE presents an important value. The weighted-RMSE is clearly better with the (extended-) Nearest Neighbors Smoothed Bootstrap.

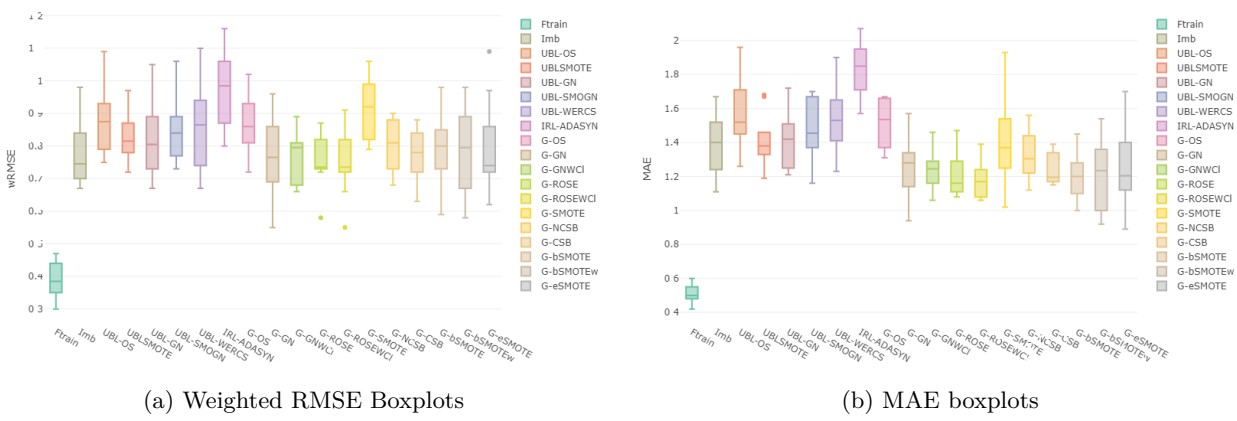

(a) Weighted RMSE Boxplots

(b) MAE boxplots

Figure 8: Boxplots of weighted RMSE and MAE for 10 runs

## C.4    Results for 20 models

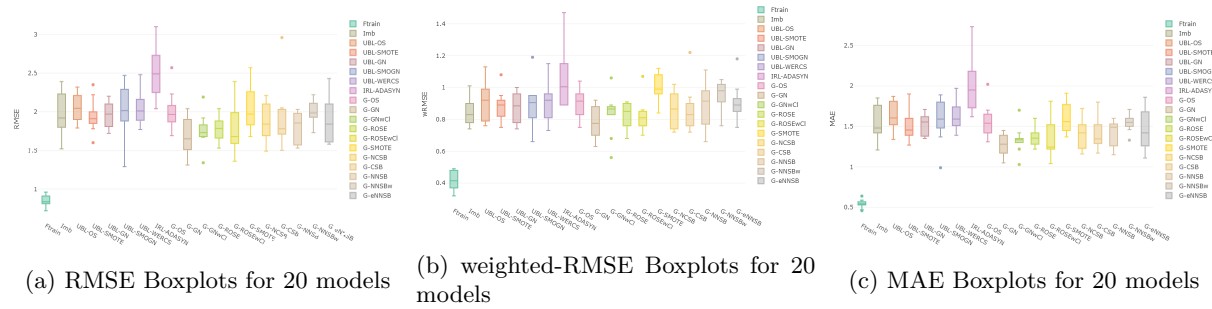

(a) RMSE Boxplots for 20 models

(b) weighted-RMSE Boxplots for 20 models

(c) MAE Boxplots for 20 models

Figure 9: Boxplots of predictive performance metrics for 20 runs

The results for the 20 models generally confirm those obtained with 10 runs. However, GOLIATH-SMOTE seems quite worse than the other GOLIATH techniques. It is important to see that the Nearest Neighbors Smoothed Bootstrap improves these results.

## C.5    Results for 20 runs

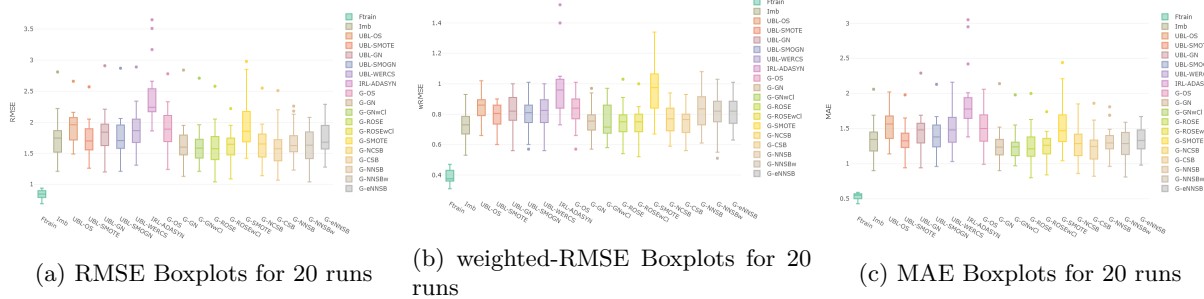

(a) RMSE Boxplots for 20 runs

(b) weighted-RMSE Boxplots for 20 runs

(c) MAE Boxplots for 20 runs

Figure 10: Boxplots of predictive performance metrics for 20 runs

The results for the 20 runs confirm those obtained with 10 runs.

### C.6    IRon specific metrics for Imbalanced Regression

The R-package *IRon: Solving Imbalanced Regression Tasks* ([https://cran.r-project.org/web/packages/IRon/IRon.pdf](https://cran.r-project.org/web/packages/IRon/IRon.pdf)), is a useful and relevant package specific to Imbalanced Regression. It is based on Ribeiro & Moniz (2020) and offers several adapted metrics. Below, we propose an analysis of these predictive performance metrics in order to compare the approaches.

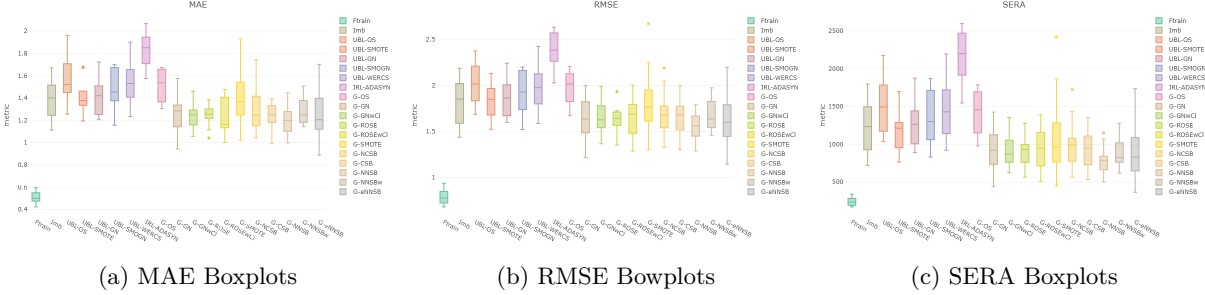

(a) MAE Boxplots      (b) RMSE Bowplots      (c) SERA Boxplots

Figure 11: Boxplots of predictive performance metrics with IRon package

The results obtained with the IRon package confirm those obtained from our metrics: The RMSE, MAE, and SERA present the same look. The Biases are lower for GOLIATH but the variances are higher.

### C.7    Impact analysis of parameters

### C.7.1    Imbalance Ratio

In this part, we analyze the results obtained according to the imbalance ratio. As a reminder, in the illustrative application we construct the imbalanced sample with a draw depending of weights as defined in 3. Here, we analyse different values of parameter $\alpha$: 0 (uniform drawing : low imbalance), 0.5, 1, 1.5 and 2 (high imbalance). To be more precise, a test sample is obtained with these draw weights and then a learning sample is drawn from the remaining sample. So if the parameter

$$\alpha$$

is high then the probability of drawing rare values will be stronger and therefore the rare values will have more chances of being in the test sample. For

$$\alpha = 0$$

the draw is uniform so the values have as much chance of being in the test sample as the learning sample.

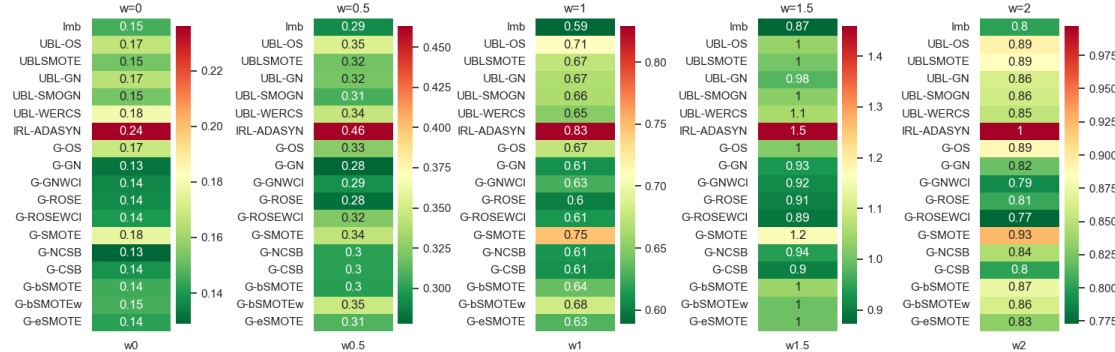

Figure 12: Heatmap of weighted RMSE mean by train samples and for different imbalanced ratios

We observe that regardless of the level of imbalance (w=0: low imbalance to w=2: high imbalance), GO-LIATH gives better results than the benchmark.

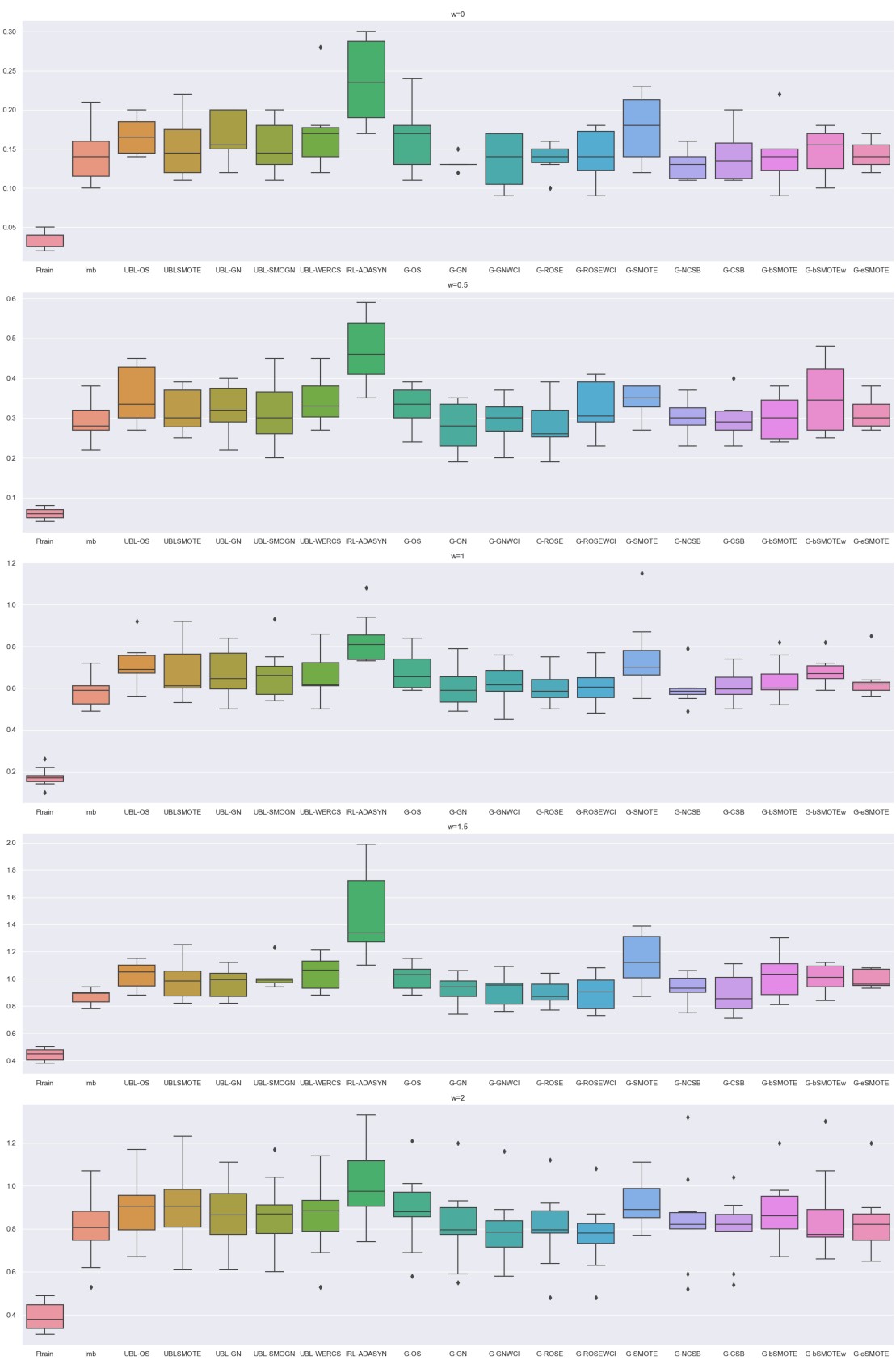

Figure 13: Boxplots of weighted RMSE by train samples and for different imbalanced ratios

### C.7.2 Sample Size

In this part, we analyze the results obtained according to the sample size. As a reminder, in the illustrative application we construct the imbalanced sample with a draw from the reminded population.

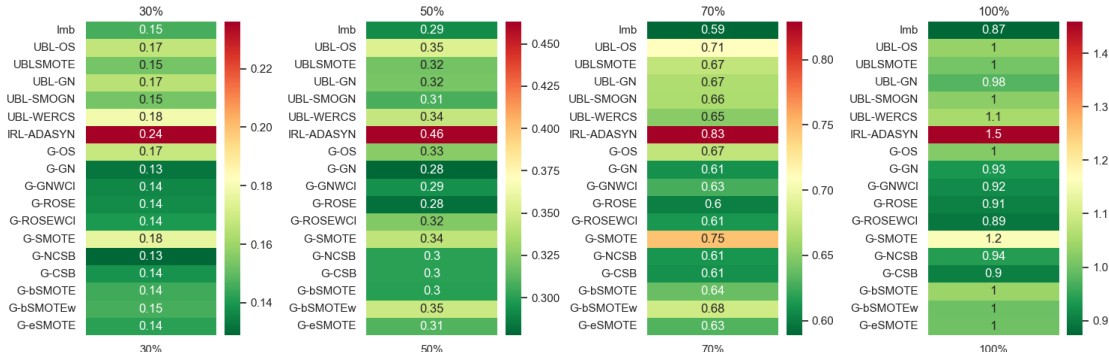

Figure 14: Heatmap of weighted RMSE mean by train samples and for different sample size

We observe that regardless of the sample size (30%: small sample to 100%: big sample), GOLIATH gives better results than the benchmark.

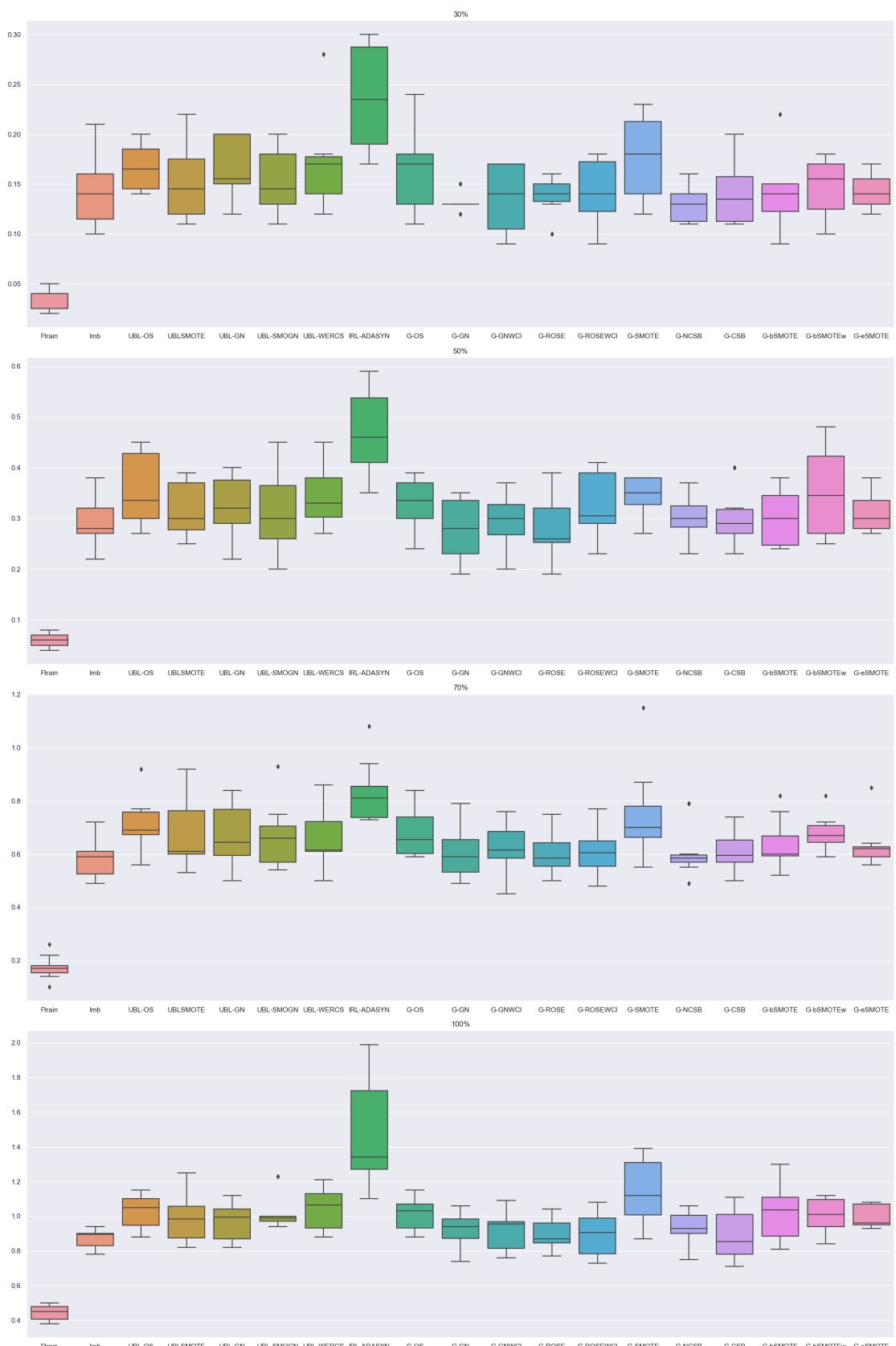

Figure 15: Boxplots of weighted RMSE by train samples and for different sample size

### C.7.3 Noise

In this part, we analyze the results obtained according to the noise : parameter $k$ which is the number of neighboors for the interpolation approaches and parameter $pert$ which is the noise for perturbation approaches.

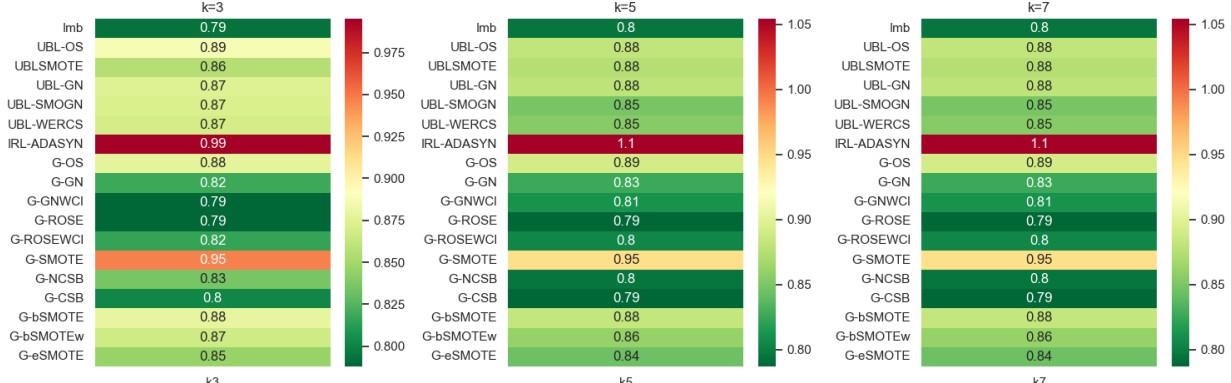

(a) Heatmap of weighted RMSE mean by train samples and for different noise parameter $k$

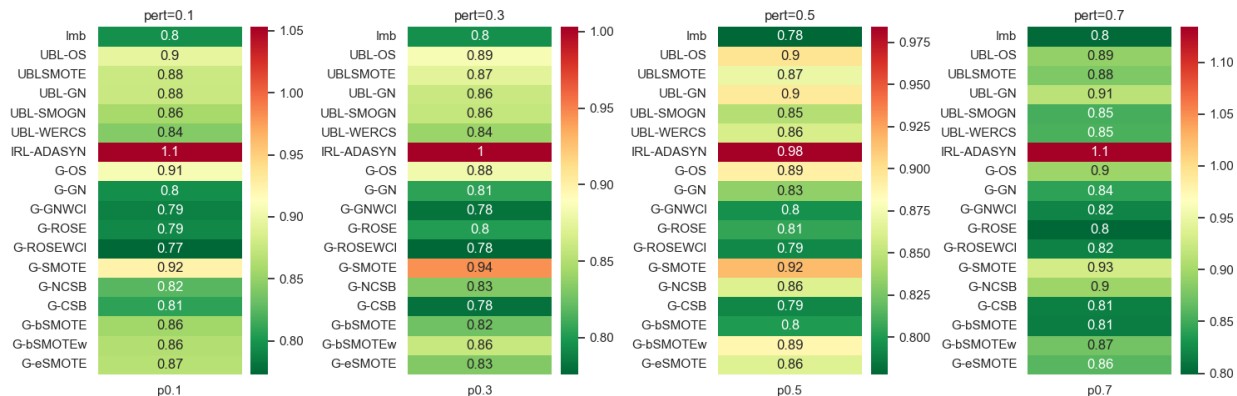

(b) Heatmap of weighted RMSE mean by train samples and for different noise parameter $k$

We observe that regardless of the level of noise, GOLIATH gives better results than the benchmark.

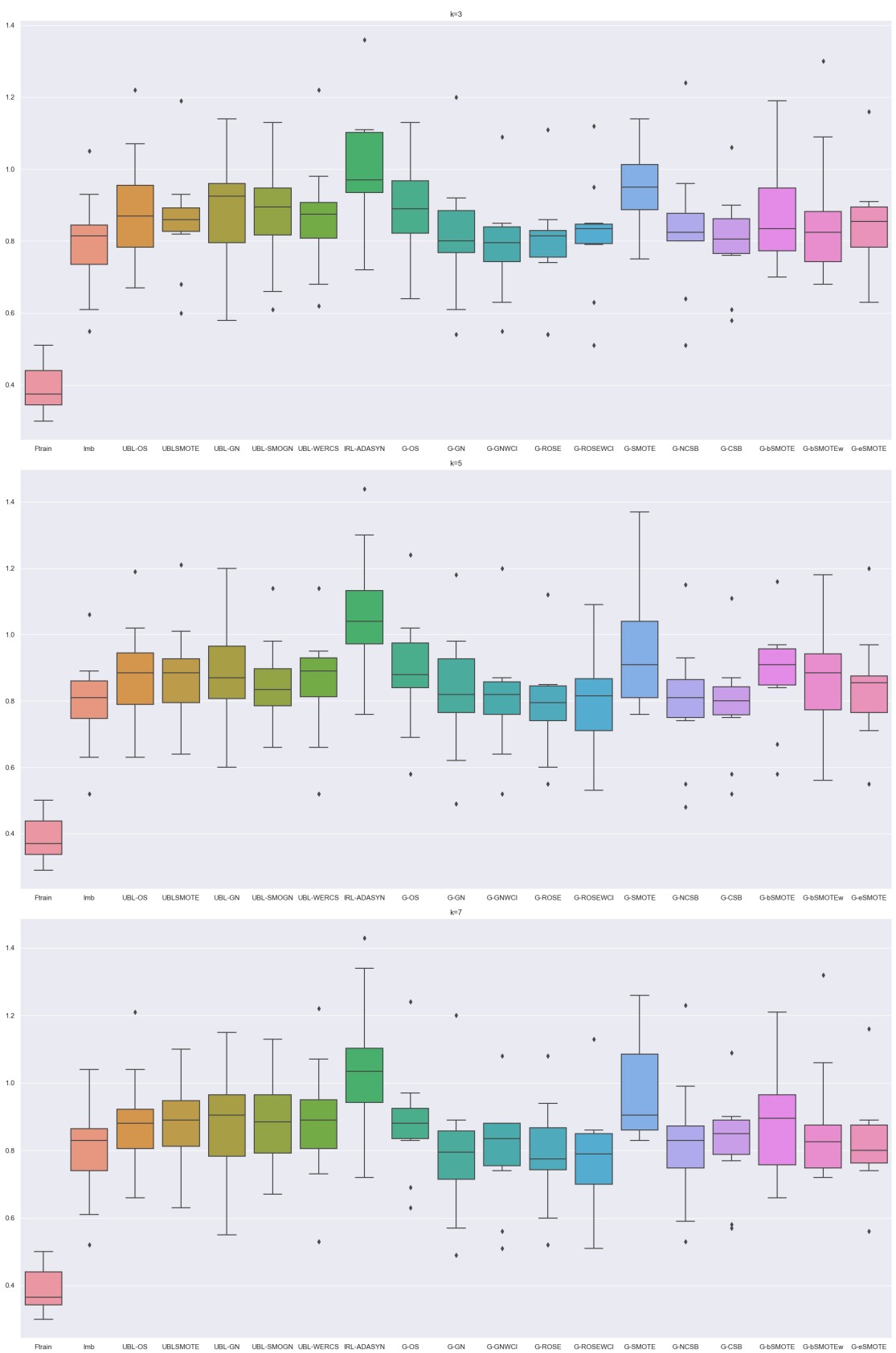

Figure 17: Boxplots of weighted RMSE by train samples and for different $k$ parameter

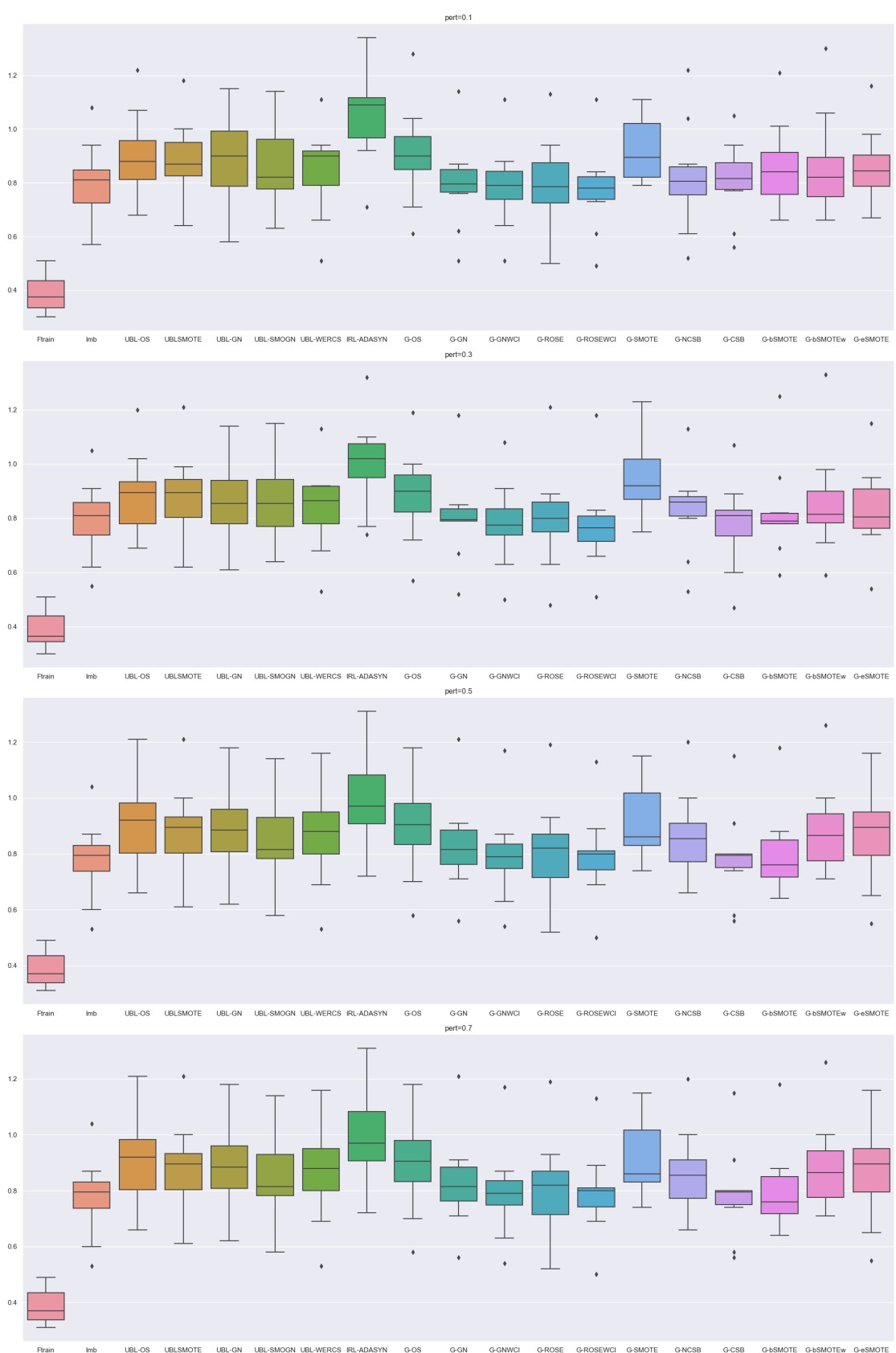

Figure 18: Boxplots of weighted RMSE by train samples and for different *pert* parameter

# D    Complementary Results for Imbalanced Regression Applications

## D.1    Protocol

The protocol for the applications is quite similar to the previous one for the illustrative application. It can be summarized as follow:

1. Data preprocessing: removal of eventual categorical covariates, eventual conversion of some covariates, removal of missing data.

2. Define *test_prop* the desired proportion of the test dataset: cf below.

3. Define *N_sample* the number of the runs i.e. the desired train-test set: 10 here.

4. Define an eventual proportion of the imbalanced dataset *imb_prop* to obtain an extremely imbalanced dataset: cf below.

5. Construct *N_sample* train-test set: repeat the following instructions:
   - draw a test sample with a size $size(data) \times test\_prop$,
   - draw $size(data - test) \times imb\_prop$ from the remaining dataset with weights squared.

6. Generate the new train datasets with the different methods. The generation is based on a weighting function.

7. Predict the test value according to the new train datasets.

*test_prop*: Abalone: 30%, Bank8FM: 50%, Boston: 30%, CpuSm: 5%, NO2: 10%
*imb_prop*: Abalone: 10%, Bank8FM: 10%, Boston: 100%, CpuSm: 5%, NO2: 100%

## D.2    Details for the Abalone dataset

The Abalone dataset is composed of 4177 observations and 8 numerical variables including 0 discrete. More details on the covariates and the target variable are given below. We can observe especially on the histograms the distribution and the eventual boundaries of the variables.

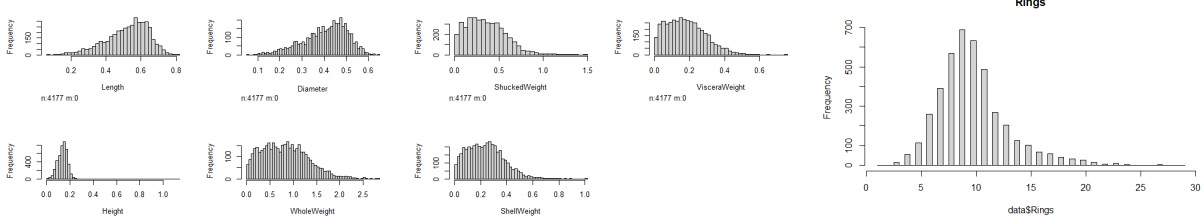

(a) Histograms of the covariates $x$ in the original dataset    (b) Histograms of the covariates $x$ in the original dataset    (c) Histogram of the target variable $Y$ in the original dataset

Figure 19: Histograms of $X$ and $y$ for the Abalone dataset

Table 5: Descriptive statistics of the dataset

| Variable | Min. | 1st Qu. | Median | Mean | 3rd Qu. | Max. |
|---|---|---|---|---|---|---|
| Rings | 1 | 8 | 9 | 9.93 | 11 | 29 |
| Length | 0.07 | 0.45 | 0.54 | 0.52 | 0.62 | 0.81 |
| Diameter | 0.06 | 0.35 | 0.42 | 0.41 | 0.48 | 0.65 |
| Height | 0 | 0.12 | 0.14 | 0.14 | 0.16 | 1.13 |
| WholeWeight | 0 | 0.44 | 0.8 | 0.83 | 1.15 | 2.83 |
| ShuckedWeight | 0 | 0.19 | 0.34 | 0.36 | 0.5 | 1.49 |
| VisceraWeight | 0 | 0.09 | 0.17 | 0.18 | 0.25 | 0.76 |
| ShellWeight | 0 | 0.13 | 0.23 | 0.24 | 0.33 | 1 |

### D.3 Details for the Bank8FM dataset

The Bank8FM dataset is composed of 4499 observations and 9 numerical variables including 1 discrete. More details on the covariates and the target variable are given below. We can observe especially on the histograms the distribution and the eventual boundaries of the variables.

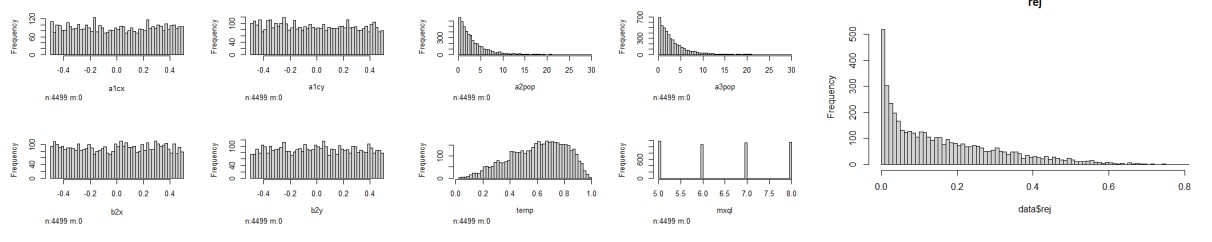

(a) Histograms of the covariates $x$ in the original dataset

(b) Histograms of the covariates $x$ in the original dataset

(c) Histogram of the target variable $Y$ in the original dataset

Figure 20: Histograms of $X$ and $y$ for the Bank8FM dataset

Table 6: Descriptive statistics of the dataset

| Variable | Min. | 1st Qu. | Median | Mean | 3rd Qu. | Max. |
|---|---|---|---|---|---|---|
| rej | 0 | 0.03 | 0.12 | 0.16 | 0.25 | 0.8 |
| a1cx | -0.5 | -0.26 | 0 | 0 | 0.26 | 0.5 |
| a1cy | -0.5 | -0.27 | -0.02 | -0.01 | 0.24 | 0.5 |
| b2x | -0.5 | -0.25 | 0.01 | 0 | 0.25 | 0.5 |
| b2y | -0.5 | -0.25 | 0 | 0 | 0.25 | 0.5 |
| a2pop | 0 | 0.9 | 2.13 | 3.05 | 4.19 | 29.71 |
| a3pop | 0 | 0.93 | 2.14 | 3.08 | 4.21 | 29.68 |
| temp | 0.04 | 0.46 | 0.63 | 0.6 | 0.77 | 0.98 |
| mxql | 5 | 5 | 7 | 6.49 | 8 | 8 |

### D.4 Details for the Boston dataset

The Boston dataset is composed of 506 observations and 13 numerical variables including 1 discrete. More details on the covariates and the target variable are given below. We can observe especially on the histograms the distribution and the eventual boundaries of the variables.

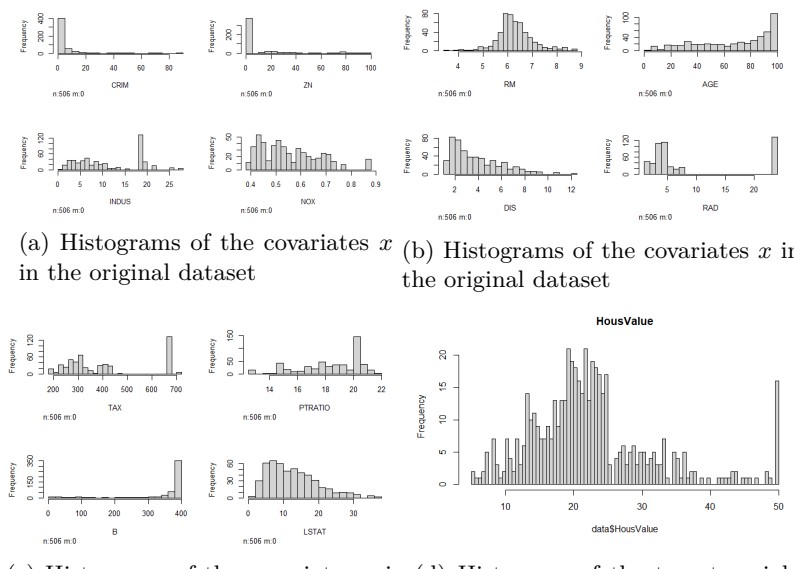

(a) Histograms of the covariates $x$ in the original dataset

(b) Histograms of the covariates $x$ in the original dataset

(c) Histograms of the covariates $x$ in the original dataset

(d) Histogram of the target variable $Y$ in the original dataset

Figure 21: Histograms of $X$ and $y$ for the Boston dataset

Table 7: Descriptive statistics of the dataset

| Variable | Min. | 1st Qu. | Median | Mean | 3rd Qu. | Max. |
|---|---|---|---|---|---|---|
| HousValue | 5 | 17.02 | 21.2 | 22.53 | 25 | 50 |
| CRIM | 0.01 | 0.08 | 0.26 | 3.61 | 3.68 | 88.98 |
| ZN | 0 | 0 | 0 | 11.36 | 12.5 | 100 |
| INDUS | 0.46 | 5.19 | 9.69 | 11.14 | 18.1 | 27.74 |
| NOX | 0.38 | 0.45 | 0.54 | 0.55 | 0.62 | 0.87 |
| RM | 3.56 | 5.89 | 6.21 | 6.28 | 6.62 | 8.78 |
| AGE | 2.9 | 45.02 | 77.5 | 68.57 | 94.07 | 100 |
| DIS | 1.13 | 2.1 | 3.21 | 3.79 | 5.19 | 12.13 |
| RAD | 1 | 4 | 5 | 9.55 | 24 | 24 |
| TAX | 187 | 279 | 330 | 408.24 | 666 | 711 |
| PTRATIO | 12.6 | 17.4 | 19.05 | 18.46 | 20.2 | 22 |
| B | 0.32 | 375.38 | 391.44 | 356.67 | 396.22 | 396.9 |
| LSTAT | 1.73 | 6.95 | 11.36 | 12.65 | 16.96 | 37.97 |

## D.5 Details for the CpuSm dataset

The CpuSm dataset is composed of 8192 observations and 13 numerical variables including 0 discrete. More details on the covariates and the target variable are given below. We can observe especially on the histograms the distribution and the eventual boundaries of the variables.

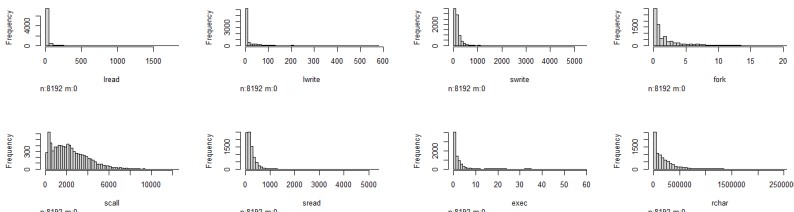

(a) Histograms of the covariates $x$ in the original dataset (b) Histograms of the covariates $x$ in the original dataset

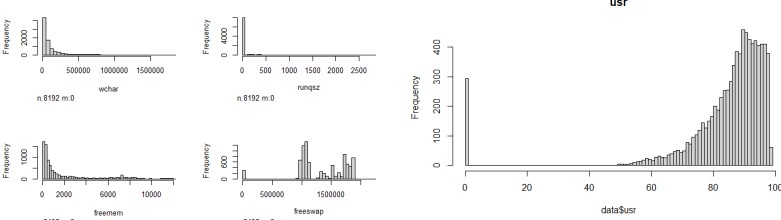

(c) Histograms of the covariates $x$ in the original dataset (d) Histogram of the target variable $Y$ in the original dataset

Figure 22: Histograms of $X$ and $y$ for the CpuSm dataset

Table 8: Descriptive statistics of the dataset

| Variable | Min. | 1st Qu. | Median | Mean | 3rd Qu. | Max. |
|---|---|---|---|---|---|---|
| usr | 0 | 81 | 89 | 83.97 | 94 | 99 |
| lread | 0 | 2 | 7 | 19.56 | 20 | 1845 |
| lwrite | 0 | 0 | 1 | 13.11 | 10 | 575 |
| scall | 109 | 1012 | 2051.5 | 2306.32 | 3317.25 | 12493 |
| sread | 6 | 86 | 166 | 210.48 | 279 | 5318 |
| swrite | 7 | 63 | 117 | 150.06 | 185 | 5456 |
| fork | 0 | 0.4 | 0.8 | 1.88 | 2.2 | 20.12 |
| exec | 0 | 0.2 | 1.2 | 2.79 | 2.8 | 59.56 |
| rchar | 278 | 33864.25 | 124779.5 | 197013.67 | 267669.25 | 2526649 |
| wchar | 1498 | 22935.5 | 46620 | 95898.29 | 106148 | 1801623 |
| runqsz | 1 | 1.2 | 2 | 19.63 | 3 | 2823 |
| freemem | 55 | 231 | 579 | 1763.46 | 2002.25 | 12027 |
| freeswap | 2 | 1042623.5 | 1289289.5 | 1328125.96 | 1730379.5 | 2243187 |

## D.6 Details for the NO2 dataset

The NO2 dataset is composed of 500 observations and 8 numerical variables including 0 discrete. More details on the covariates and the target variable are given below. We can observe especially on the histograms the distribution and the eventual boundaries of the variables.

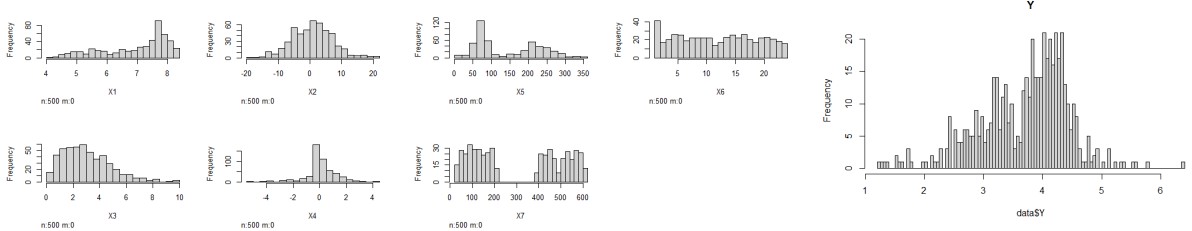

(a) Histograms of the covariates $x$ in the original dataset (b) Histograms of the covariates $x$ in the original dataset (c) Histogram of the target variable $Y$ in the original dataset

Figure 23: Histograms of $X$ and $y$ for the NO2 dataset

Table 9: Descriptive statistics of the dataset

| Variable | Min. | 1st Qu. | Median | Mean | 3rd Qu. | Max. |
|---|---|---|---|---|---|---|
| Y | 1.22 | 3.21 | 3.85 | 3.7 | 4.22 | 6.4 |
| X1 | 4.13 | 6.18 | 7.43 | 6.97 | 7.79 | 8.35 |
| X2 | -18.6 | -3.9 | 1.1 | 0.85 | 4.9 | 21.1 |
| X3 | 0.3 | 1.67 | 2.8 | 3.06 | 4.2 | 9.9 |
| X4 | -5.4 | -0.2 | 0 | 0.15 | 0.6 | 4.3 |
| X5 | 2 | 72 | 97 | 143.37 | 220 | 359 |
| X6 | 1 | 6 | 12.5 | 12.38 | 18 | 24 |
| X7 | 32 | 118.75 | 212 | 310.47 | 513 | 608 |

## D.7 Predictive performances metrics

The following Figures show the predictive performance metrics for the 5 datasets. We can see that the previous results on the illustrative application are confirmed: GOLIATH outperforms the results.

**Abalone dataset** The following Figures show the predictive performance metrics for the Abalone dataset.

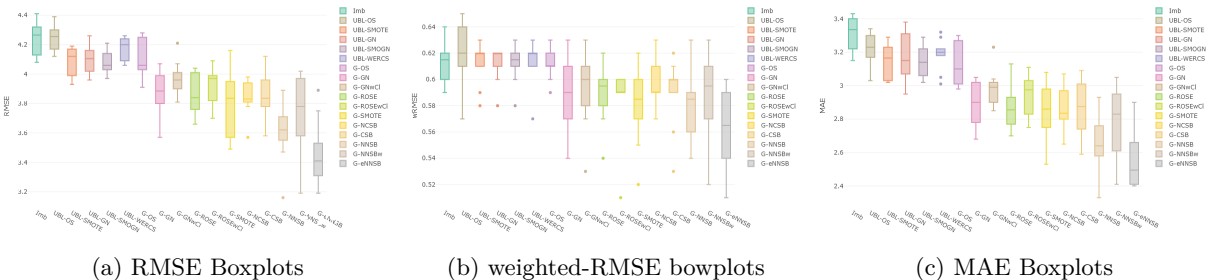

(a) RMSE Boxplots          (b) weighted-RMSE bowplots          (c) MAE Boxplots

Figure 24: Boxplots of predictive performance metrics for Abalone Dataset

**Bank8FM dataset** The following Figures show the predictive performance metrics for the Bank8FM dataset.

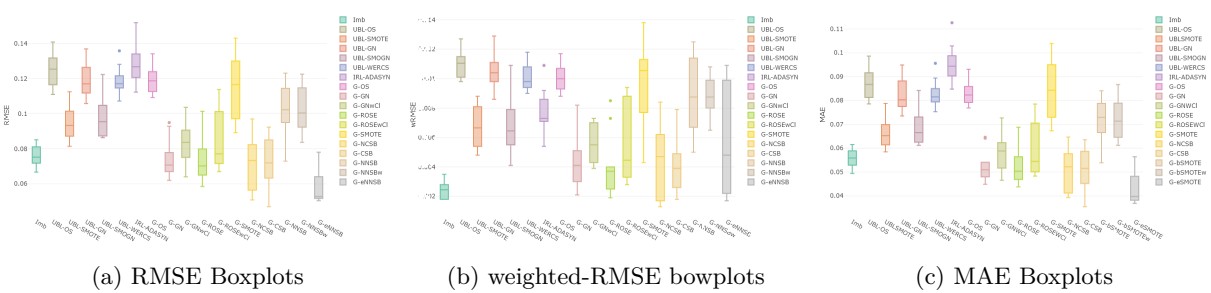

(a) RMSE Boxplots          (b) weighted-RMSE bowplots          (c) MAE Boxplots

Figure 25: Boxplots of predictive performance metrics for Bank8FM Dataset

**Boston dataset** The following Figures show the predictive performance metrics for the Boston dataset.

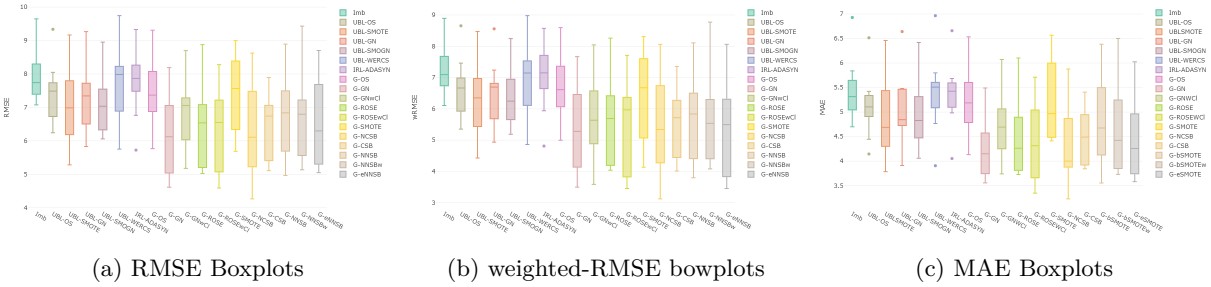

(a) RMSE Boxplots        (b) weighted-RMSE bowplots        (c) MAE Boxplots

Figure 26: Boxplots of predictive performance metrics for Boston Dataset

**CpuSm dataset**   The following Figures show the predictive performance metrics for the CpuSm dataset.

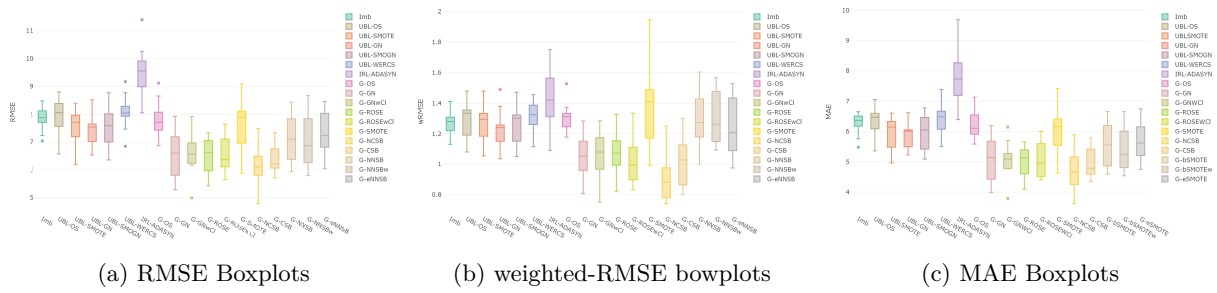

(a) RMSE Boxplots        (b) weighted-RMSE bowplots        (c) MAE Boxplots

Figure 27: Boxplots of predictive performance metrics for CpuSm Dataset

**NO2 dataset**   The following Figures show the predictive performance metrics for the NO2 dataset.

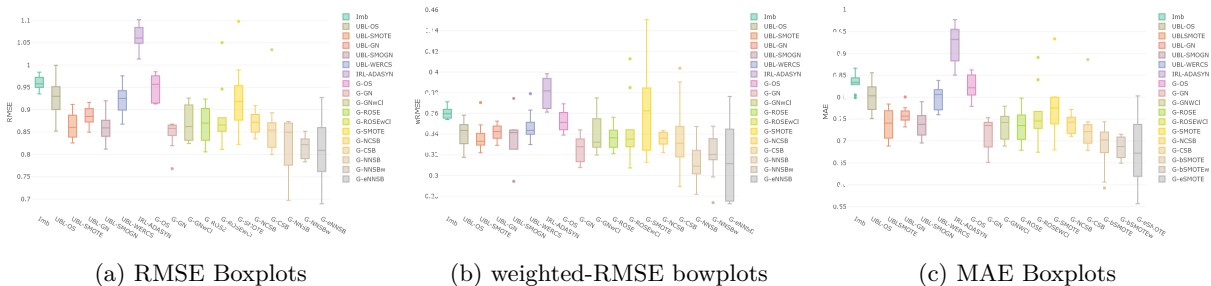

(a) RMSE Boxplots        (b) weighted-RMSE bowplots        (c) MAE Boxplots

Figure 28: Boxplots of predictive performance metrics for NO2 Dataset

### D.8 Impact analysis of parameters

#### D.8.1 Abalone dataset

In this part, we analyze the results obtained according to the noise : parameter $k$ which is the number of neighboors for the interpolation approaches and parameter $pert$ which is the noise for perturbation approaches.

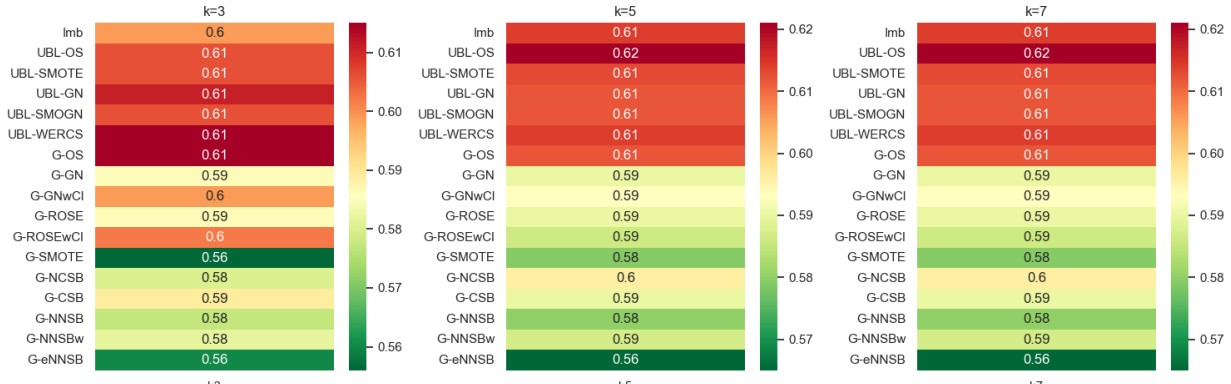

(a) Heatmap of weighted RMSE mean by train samples and for different noise parameter $k$

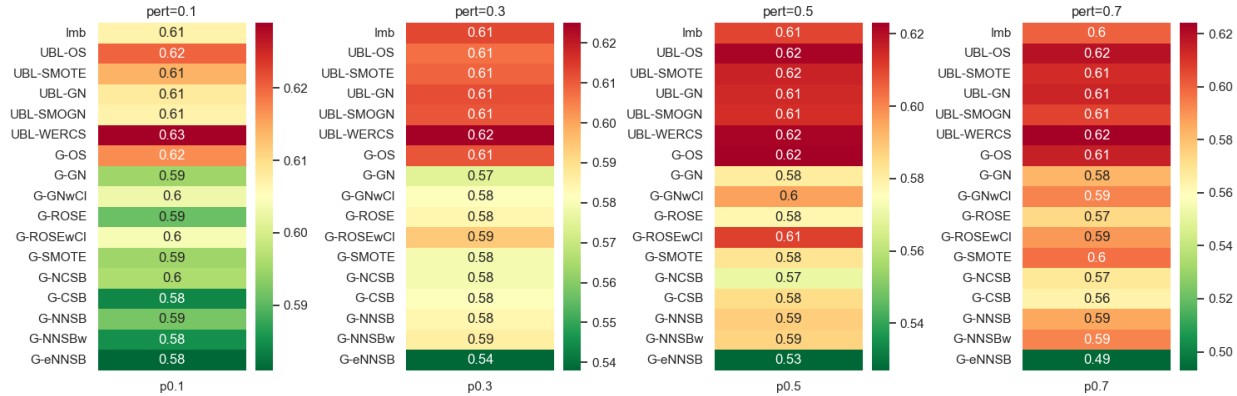

(b) Heatmap of weighted RMSE mean by train samples and for different noise parameter $k$

We observe that regardless of the level of noise, GOLIATH gives better results than the benchmark.

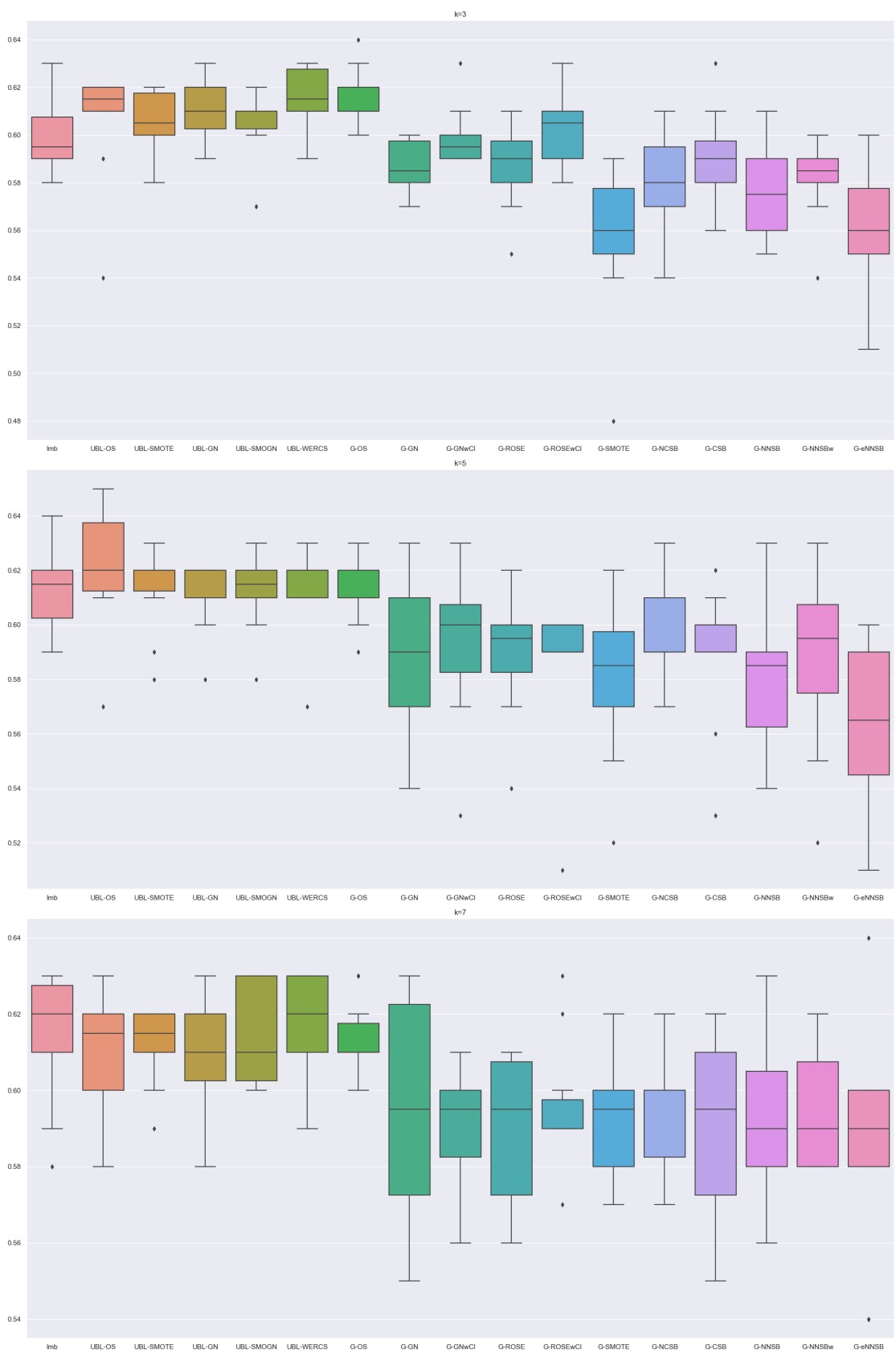

Figure 30: Boxplots of weighted RMSE by train samples and for different $k$ parameter

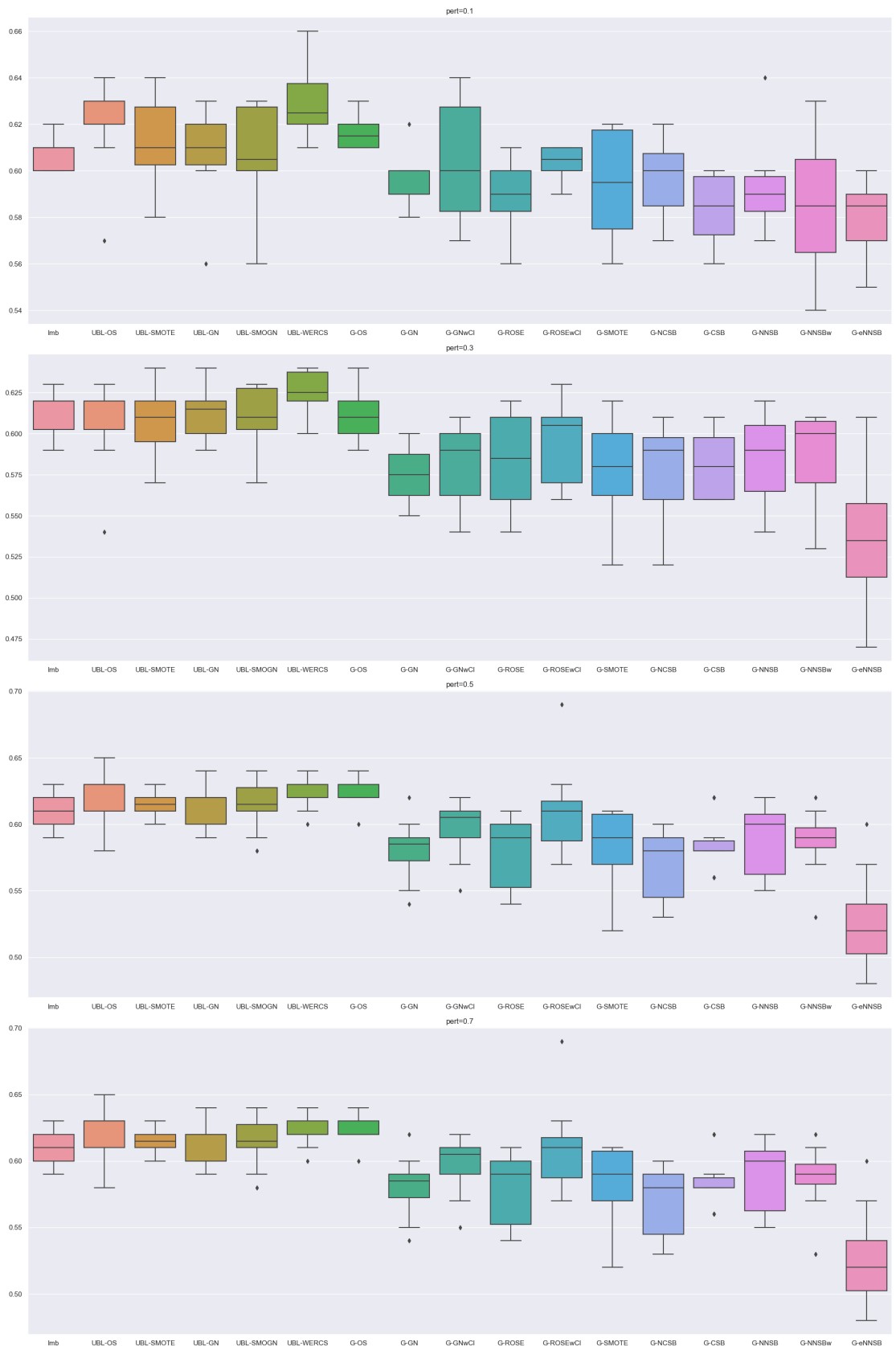

Figure 31: Boxplots of weighted RMSE by train samples and for different *pert* parameter

### D.8.2 Bank8fm dataset

In this part, we analyze the results obtained according to the noise : parameter $k$ which is the number of neighboors for the interpolation approaches and parameter $pert$ which is the noise for perturbation approaches.

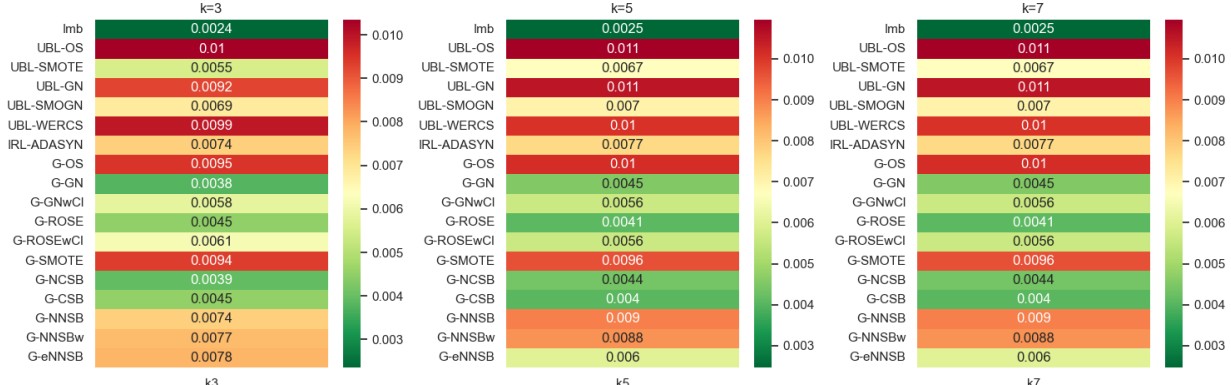

(a) Heatmap of weighted RMSE mean by train samples and for different noise parameter $k$

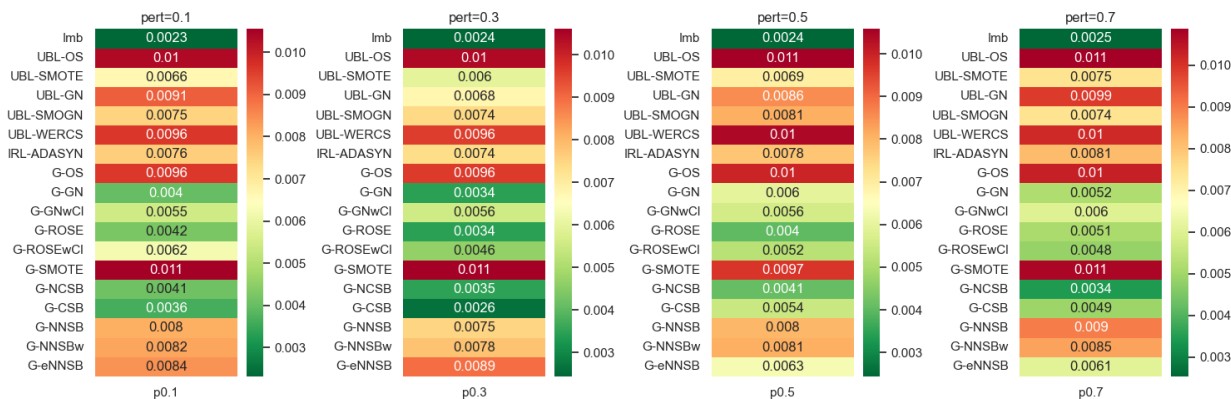

(b) Heatmap of weighted RMSE mean by train samples and for different noise parameter $k$

We observe that regardless of the level of noise, GOLIATH gives better results than the benchmark.

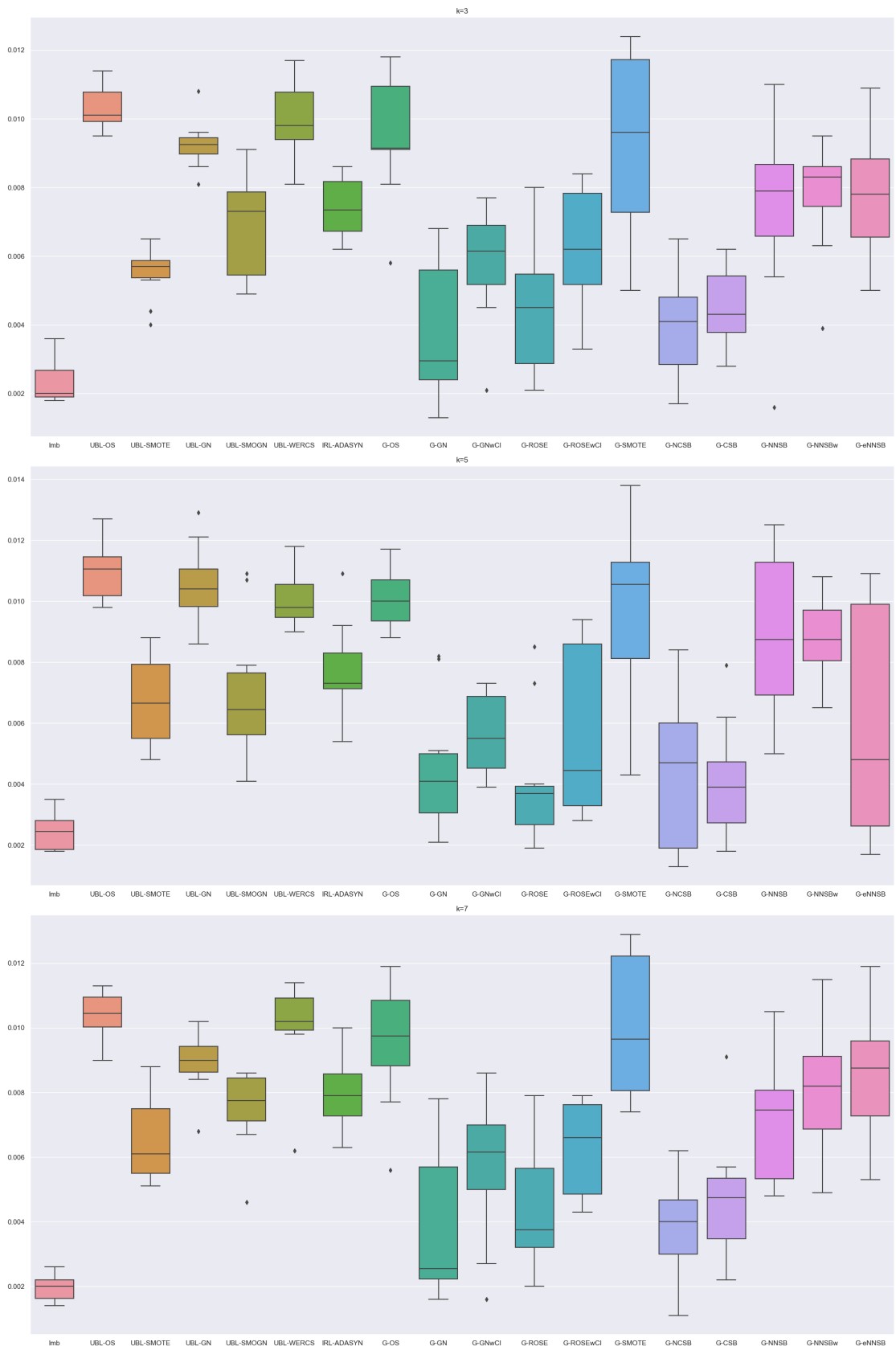

Figure 33: Boxplots of weighted RMSE by train samples and for different $k$ parameter

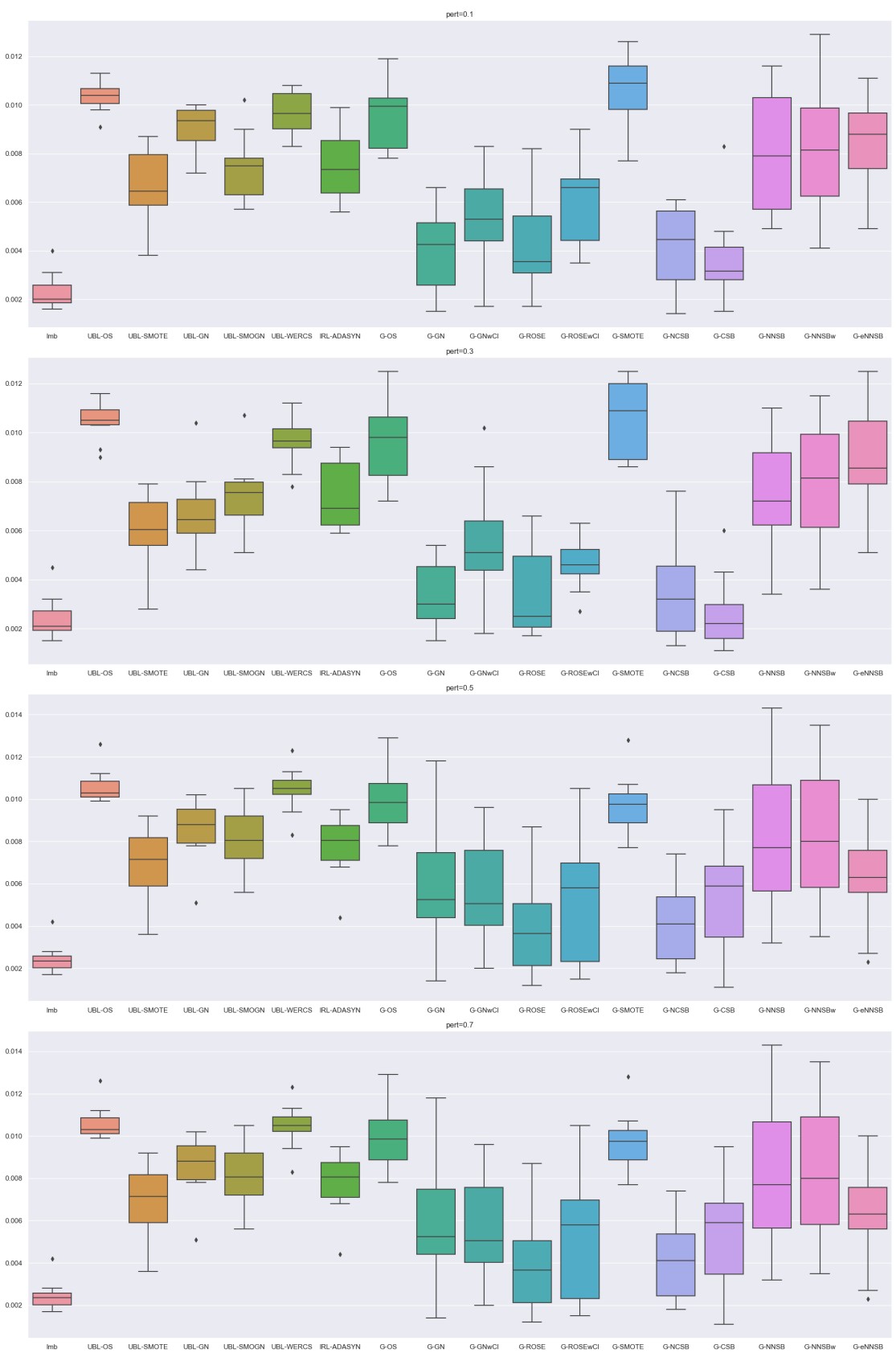

Figure 34: Boxplots of weighted RMSE by train samples and for different *pert* parameter

### D.8.3 Boston dataset

In this part, we analyze the results obtained according to the noise : parameter $k$ which is the number of neighboors for the interpolation approaches and parameter $pert$ which is the noise for perturbation approaches.

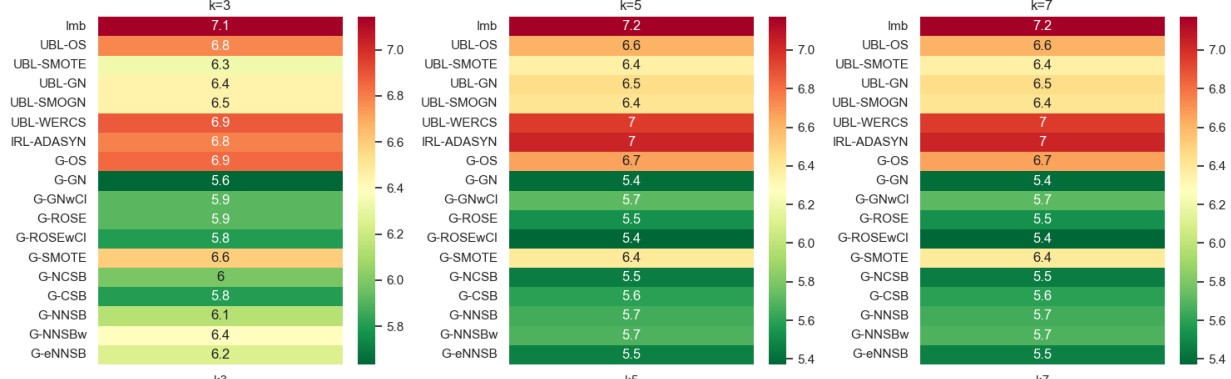

(a) Heatmap of weighted RMSE mean by train samples and for different noise parameter $k$

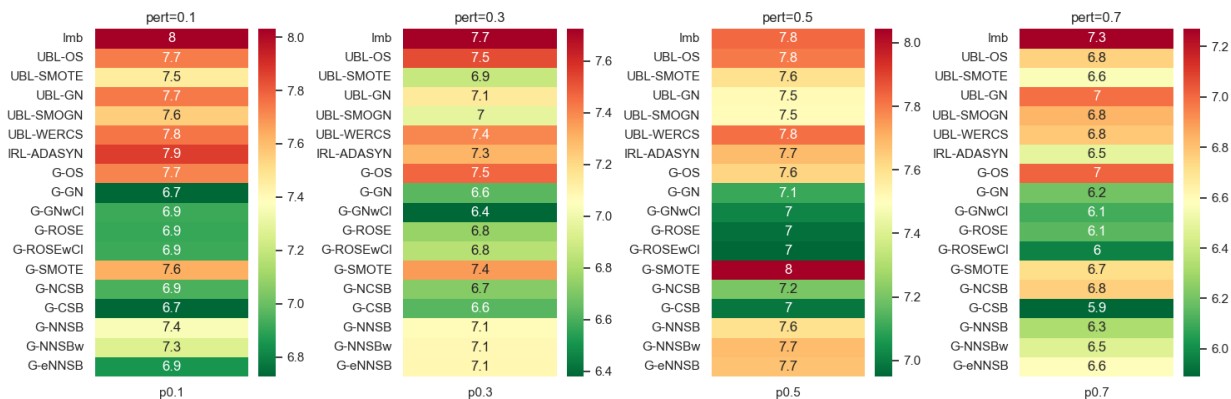

(b) Heatmap of weighted RMSE mean by train samples and for different noise parameter $k$

We observe that regardless of the level of noise, GOLIATH gives better results than the benchmark.

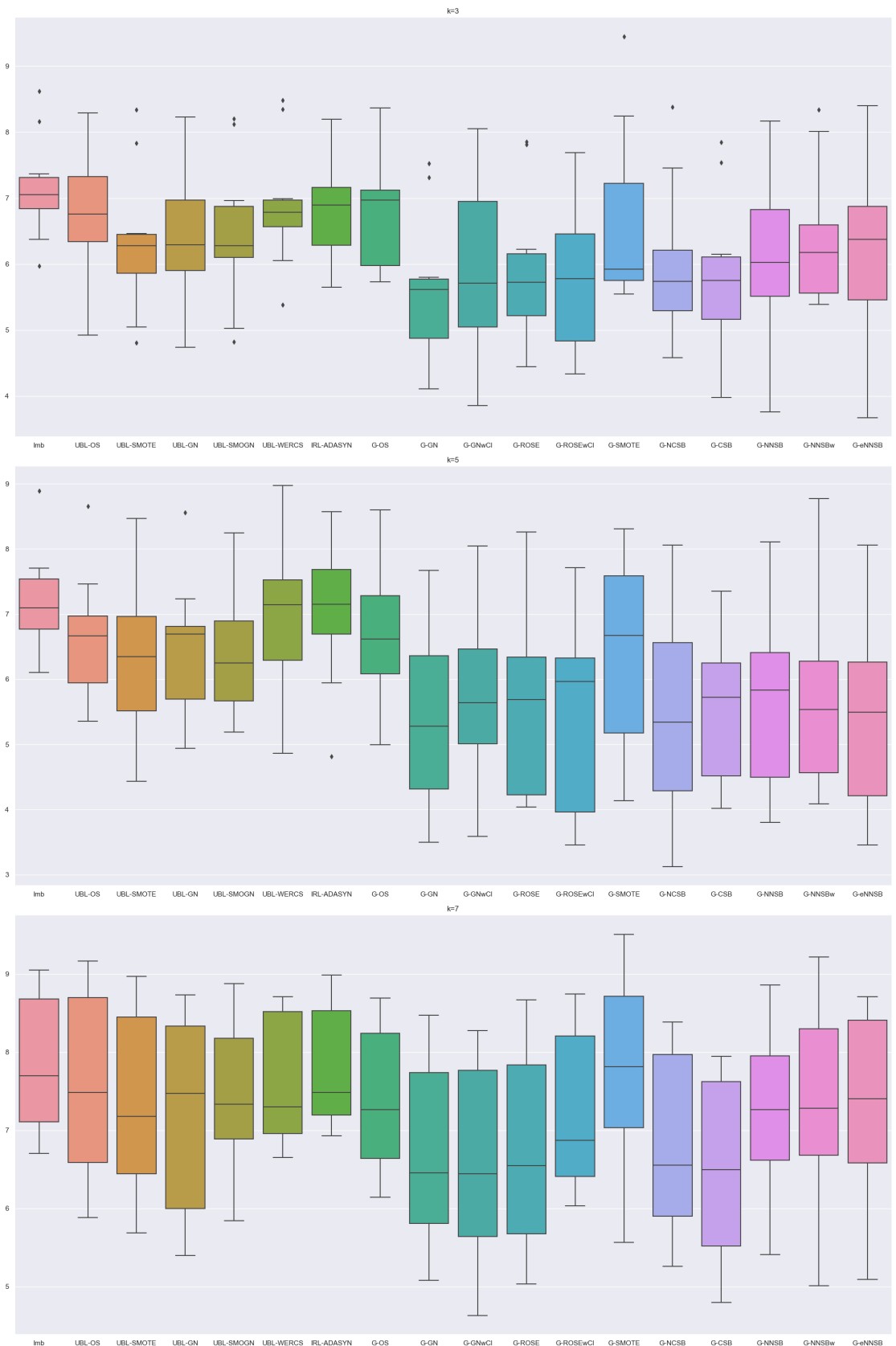

Figure 36: Boxplots of weighted RMSE by train samples and for different $k$ parameter

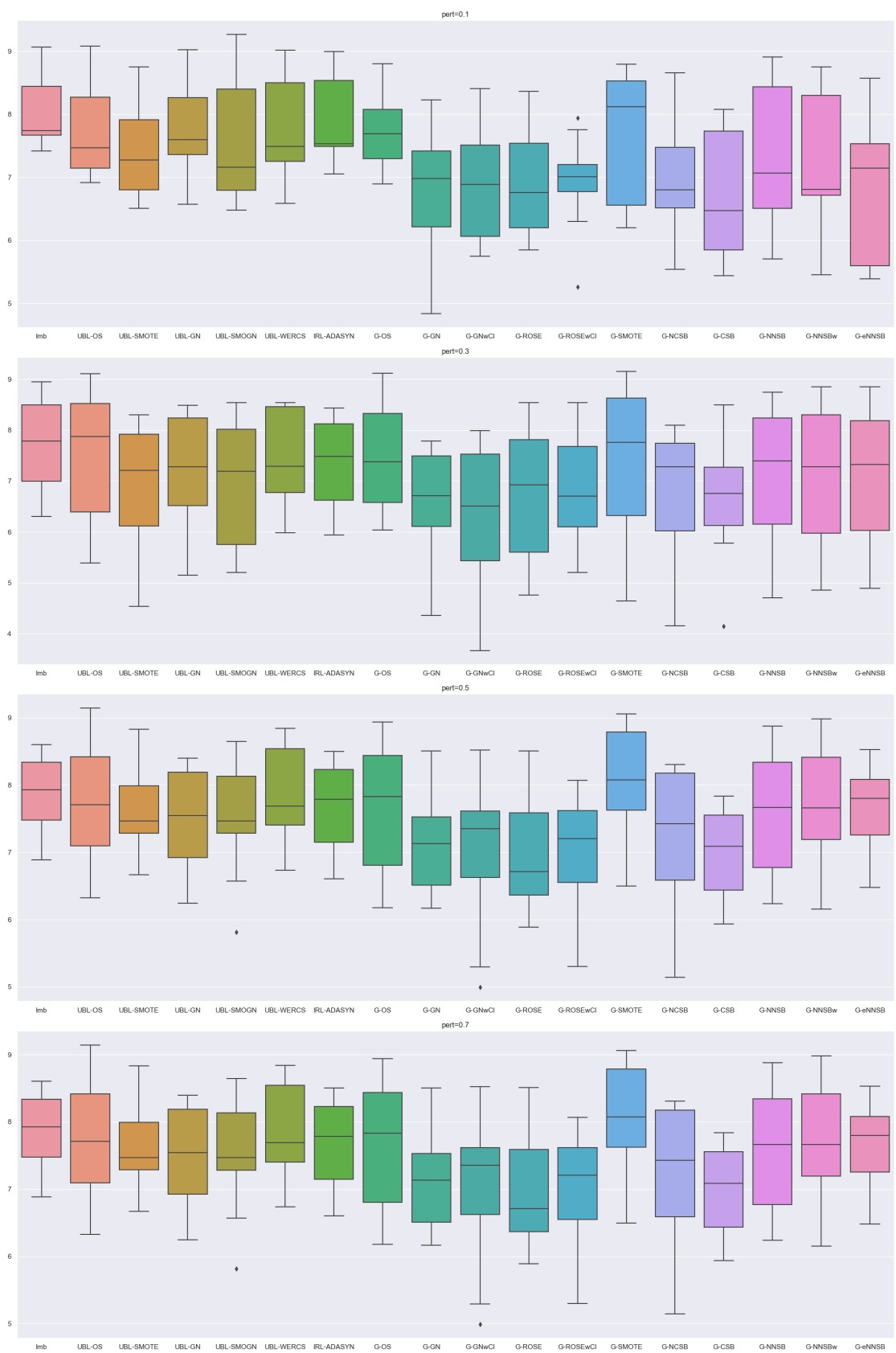

Figure 37: Boxplots of weighted RMSE by train samples and for different *pert* parameter

### D.8.4 CpuSm dataset

In this part, we analyze the results obtained according to the noise : parameter $k$ which is the number of neighboors for the interpolation approaches and parameter $pert$ which is the noise for perturbation approaches.

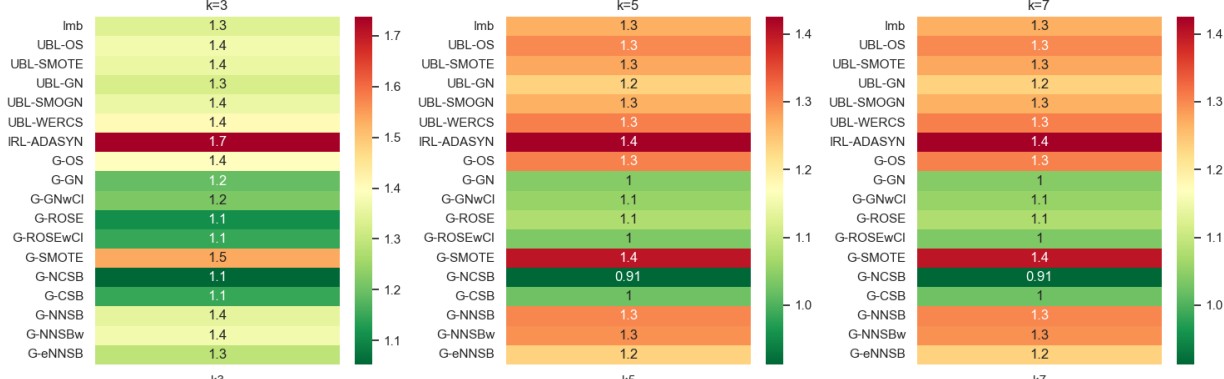

(a) Heatmap of weighted RMSE mean by train samples and for different noise parameter $k$

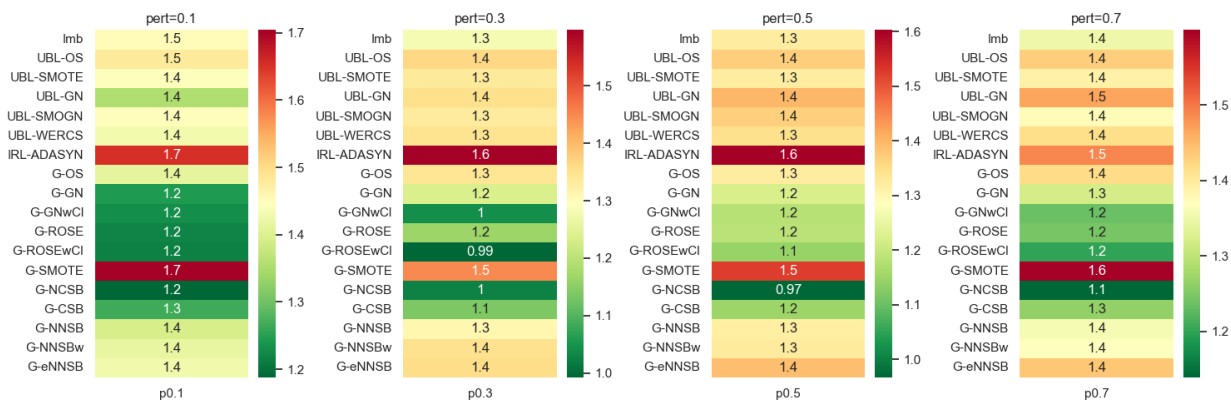

(b) Heatmap of weighted RMSE mean by train samples and for different noise parameter $k$

We observe that regardless of the level of noise, GOLIATH gives better results than the benchmark.

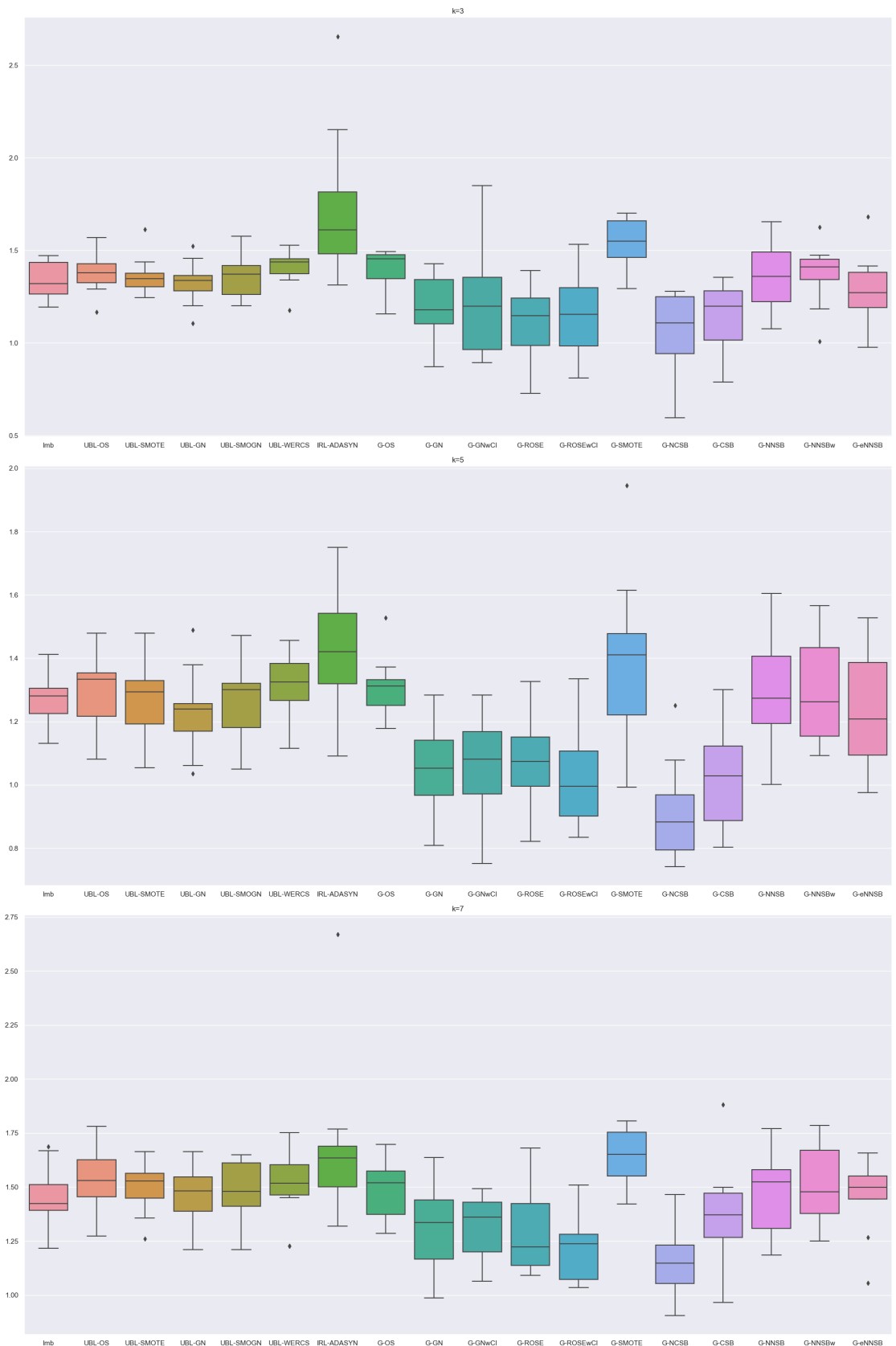

Figure 39: Boxplots of weighted RMSE by train samples and for different $k$ parameter

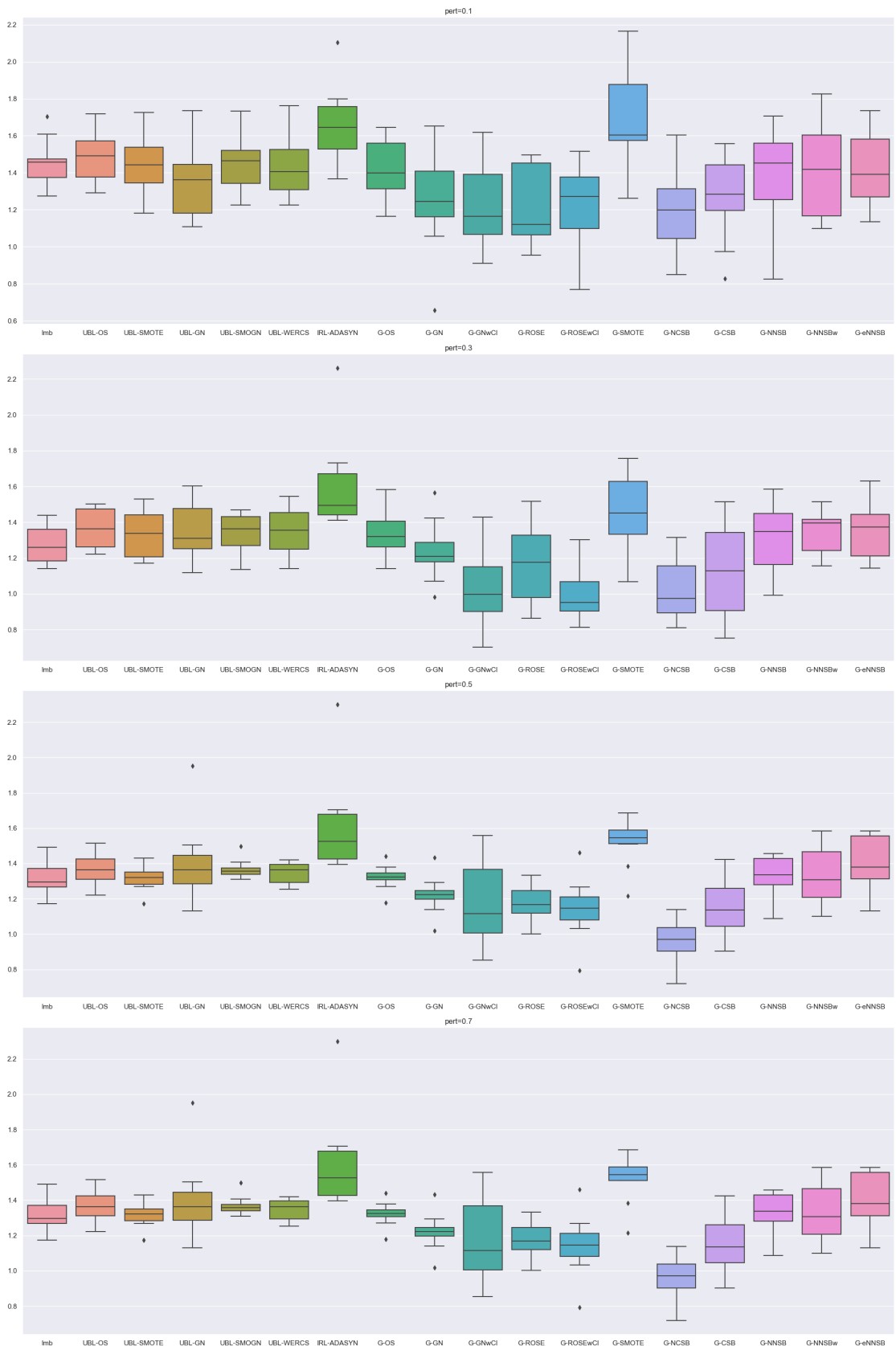

Figure 40: Boxplots of weighted RMSE by train samples and for different *pert* parameter

### D.8.5 NO2 dataset

In this part, we analyze the results obtained according to the noise : parameter $k$ which is the number of neighboors for the interpolation approaches and parameter *pert* which is the noise for perturbation approaches.

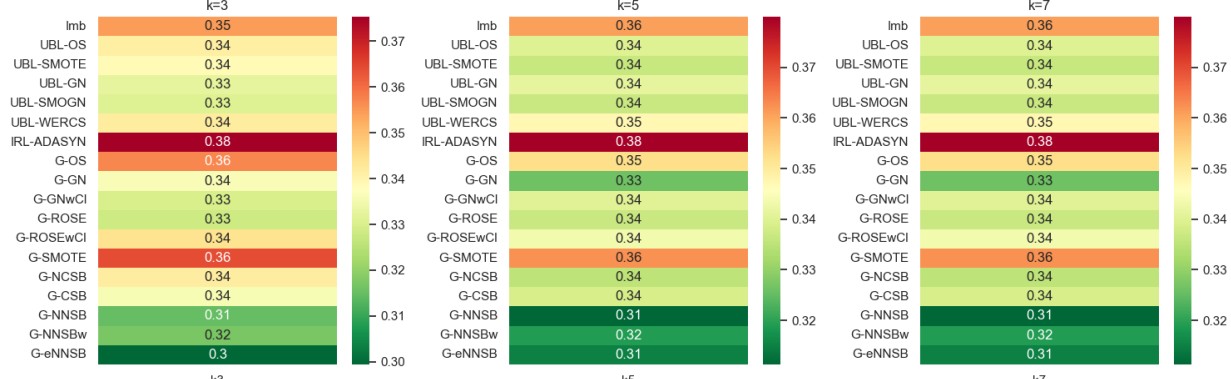

(a) Heatmap of weighted RMSE mean by train samples and for different noise parameter $k$

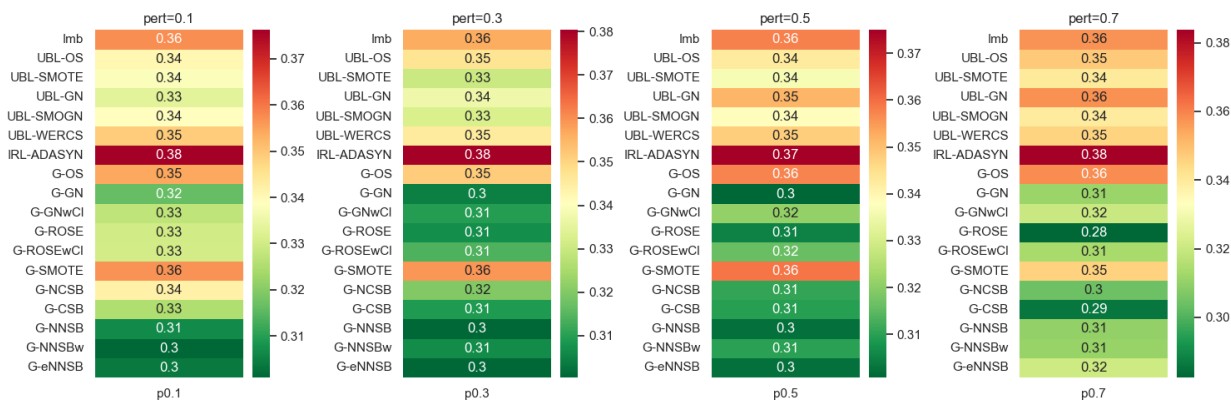

(b) Heatmap of weighted RMSE mean by train samples and for different noise parameter $k$

We observe that regardless of the level of noise, GOLIATH gives better results than the benchmark.

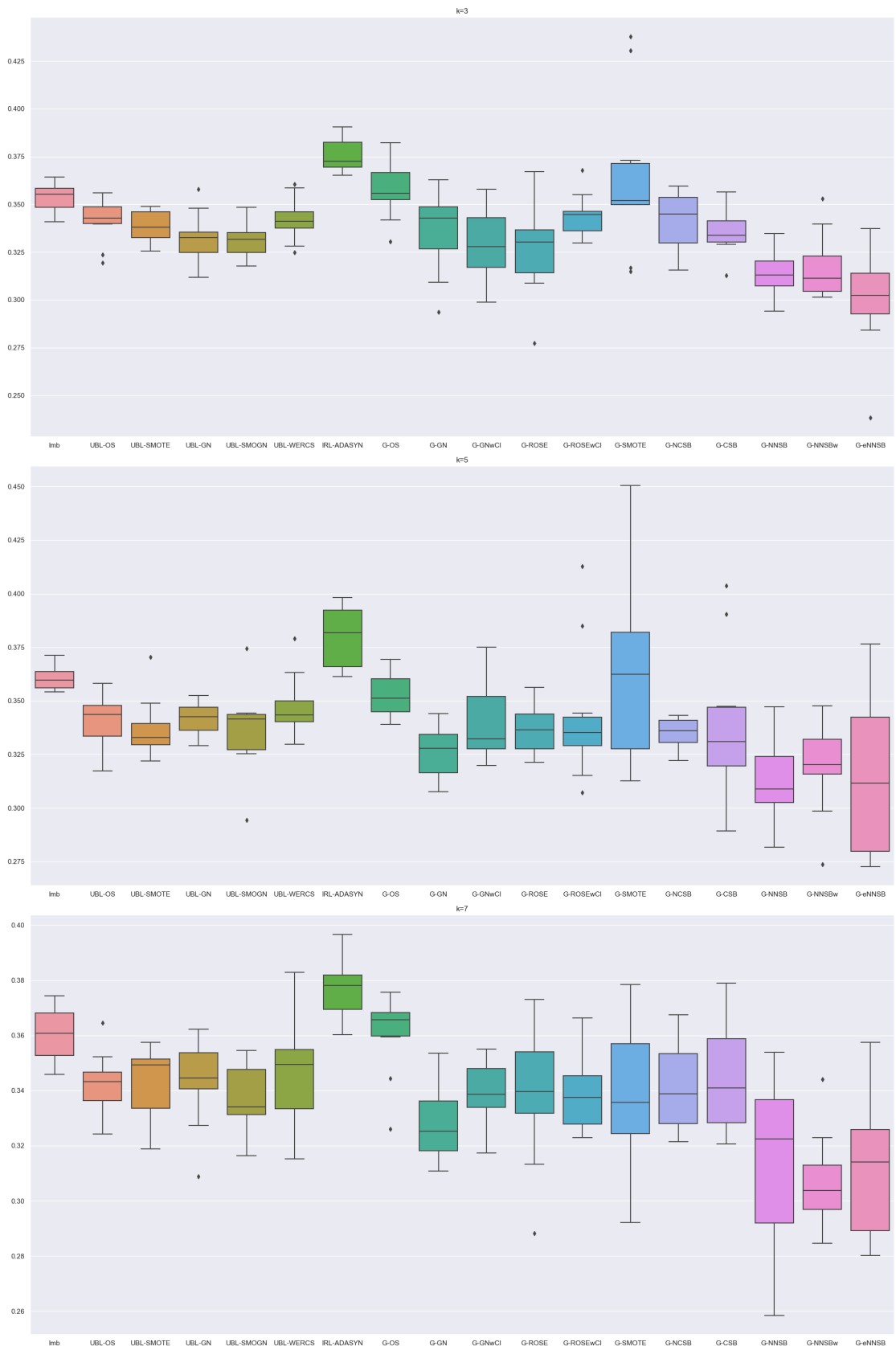

Figure 42: Boxplots of weighted RMSE by train samples and for different $k$ parameter

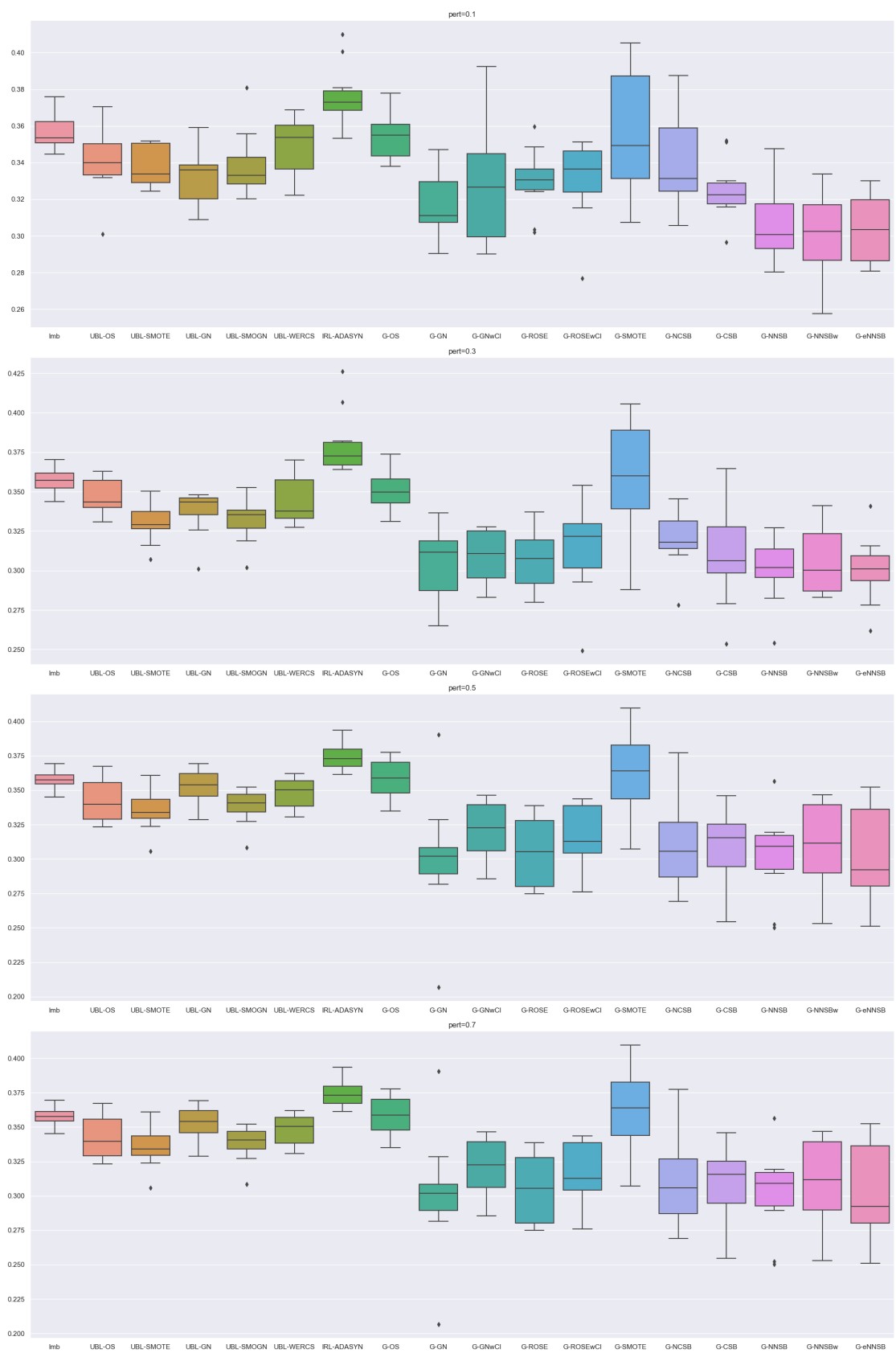

Figure 43: Boxplots of weighted RMSE by train samples and for different *pert* parameter