# OpenReview forum: "Generalized Oversampling for Learning from Imbalanced datasets and Associated Theory: Application in Regression"
_TMLR — Accepted by TMLR_

### Review · Reviewer_1Gej · 2024-03-25

**Summary Of Contributions:**

This paper presents GOLIATH, a kernel-based generalization of several oversampling methods commonly used in the literature, with the goal of applying it to regression problems, which is less common in the oversampling literature. Starting with a very general form of data synthesis, the authors reframe SMOTE and other approaches in this form. Having shown this, the authors then develop more general versions of these approaches. For regression tasks, the paper proposes several kernels based on each feature's idiosyncrasies. Experiments are provided on 5 datasets.

**Audience:**

Yes

**Broader Impact Concerns:**

None.

**Claims And Evidence:**

No

**Requested Changes:**

* Some minor editorial fixes (see Strengths & Weaknesses above)
* More motivation is necessary to justify the inverse KDE rule used in the paper.

**Strengths And Weaknesses:**

## Strengths

* Broadly, the paper is well-presented. Aside from some notational confusion (see below), it is easy to follow.
* Relevant alternatives to parts of the approach (such as k-nn-based KDE) are attempted.
* Many of the experimental decisions are based on precedent set by prior work.

## Weaknesses

- In Section 1, "Very few works have addressed the problem of imbalanced regression although many important real-world applications in different fields such as economy, meteorology, or insurance"--needs refactoring.
- In Section 2, the authors claim, "However, the techniques proposed in this context rely on deep learning, which is effective for images but less so for tabular data", although that seems debated in literature: [1, 4, 5] seem to disagree, while [2-3] agree, though it should be noted that [3] specifically excluded datasets of high dimensionality.
- Is Section 4 supposed to be Section 3.1?
- Equation 1 is a bit unclear: since $\tilde{x}$ represents all the observations, it is unknown at this point whether the kernel is over a matrix (of $n$ observations with $p$ dimensions), or if there should be another summation.
- Continuing from the previous point, this paper would benefit from a Notation section. For example, in Section 4.1, it is unclear what some terms such as $x_{ij}(\ell)$ mean. My working assumption is that $x_{ij}$ is the $j$th component of a data point, given that the $i$th data point was randomly chosen, and that $x_{ij}(\ell)$ is the $j$th component of the $\ell$th nearest neighbor, given that the $i$th data point was chosen.
- Some of the links to the appendix seem incorrect. For example, for the evaluation protocol mentioned in Section 7, the paper points to "Appendix 6", which likely refers to Appendix D.1.
- In the protocol for drawing training/test sets, it's not clear to me why the different weighting is necessary (and experimentally valid). Throughout the paper and the appendix, the authors refer to the use of the inverse of the KDE (or its squared) to draw extreme values, and the motivation for doing this is unclear.

## References

[1] Gorishniy, Y., Rubachev, I., Khrulkov, V., & Babenko, A. (2021). Revisiting deep learning models for tabular data. *Advances in Neural Information Processing Systems*, *34*, 18932-18943.

[2] Borisov, V., Leemann, T., Seßler, K.,  Haug, J., Pawelczyk, M., & Kasneci, G. (2022). Deep neural networks  and tabular data: A survey. *IEEE Transactions on Neural Networks and Learning Systems*.

[3] Grinsztajn, L., Oyallon, E., &  Varoquaux, G. (2022). Why do tree-based models still outperform deep  learning on typical tabular data?. *Advances in Neural Information Processing Systems*, *35*, 507-520.

[4] Katzir, L., Elidan, G., & El-Yaniv, R. (2020, October). Net-dnf: Effective deep modeling of tabular data. In *International conference on learning representations*.

[5] Shavitt, I., & Segal, E. (2018). Regularization learning networks: deep learning for tabular datasets. *Advances in Neural Information Processing Systems*, *31*.

---

### Review · Reviewer_toEQ · 2024-03-29

**Summary Of Contributions:**

Author introduce a novel method for data augmentation, coined GOLIATH algorithm, that is particularly well-suited for imbalanced regression. The usefulness of the proposed methods is evaluated mainly empirically by numerical experiments.

**Audience:**

Yes

**Claims And Evidence:**

No

**Requested Changes:**

see above

**Strengths And Weaknesses:**

* Authors tackle a somewhat under-explored topic, i.e., imbalanced regression problems.

My main concern is the poor quality of presentation and use of language. First of all, the contributions are not explained precisely enough in Section 1. It seems that the main contribution is the GOLIATH algorithm. However, it is not clear what the benefits of this algorithm are, e.g., in terms of computational and statistical aspects. Moreover the GOLIATH algorithm needs to be discussed in much more detail and should also be formulated in more self-contained fashion.

Some statements are quite vague and should be made more precise (e.g., "..measuring imbalance in regression is challenging because the data is continuous")

Section 2 is quite short and maybe integrated in Section 1. Section 3 seems to be empty.

Each Section should start with an introduction that provides an overview of its contents and how the section build on/serves other sections.

The authors should also try to obtain theoretical performance guarantees for some simple probabilistic model for the datasets (similar in spirit to the analysis of clustered federated learning in [Ref1])

It might be useful if authors could try to put their work into context of graph-based learning methods. Indeed, the kernel function in (1) could be interpreted as a variant of an adjacency matrix for a graph.

[Ref1] Y. SarcheshmehPour, Y. Tian, L. Zhang and A. Jung, "Clustered Federated Learning via Generalized Total Variation Minimization," in IEEE Transactions on Signal Processing, vol. 71, pp. 4240-4256, 2023, doi: 10.1109/TSP.2023.3322848.

---

> ### Author Response · Authors · 2024-04-05
> **We sincerely appreciate your careful review, understanding, and constructive feedback. We have edited the submission with the updated paper based on your feedback., and we would like to provide some clarifications below.**
>
> **Weaknesses**
> - We apologize for the difficulty in understanding the notations. We have simplified the writing to improve readability. We have also better presented the problem as well as the contributions of our article. We added a “*Notations and Problem Setting*” subsection to formally define the Imbalanced Regression. GOLIATH cannot be considered simply as another algorithm. It's rather a unique formulation that encompasses the two main families of generators (see Branco et al., 2016) and associates it with a statistical expression rather than an algorithmic one.
> - We added a “*Notations and Problem Setting*” subsection with the following sentences: "*In regression, the continuous and infinite nature of $Y$ introduces two main challenges: i) It is not easy to define "rare" values and differentiate them from "frequent" values} - unlike in classification where this is directly given by the classes; ii) Therefore, measuring imbalance is not immediate and is complicated to assess - unlike in classification where it suffices to compare the classes. GOLIATH provides a solution by using the inverse of the kernel density to determine the weights associated with $Y$; [iii) Unlike classification, where the labels of $Y$ remain unchanged during synthetic data creation, in regression it is necessary to generate new and relevant values for the target variable.]*"
> - We are truly sorry for this typo: "4 General Formulation of GOLIATH" is a subsection of 3, hence (3.1). We expanded Section 2.
> - It's a relevant point, we followed this suggestion, and it indeed contributes to making the reading smoother. Thank you for this advice.
> - We only discuss convergence results in remarks 1 for classical kernels and 4 and for the non-classical kernels. Indeed, we indicated that the perturbation-based generators we propose exhibit convergence properties in univariate settings. However, additional theoretical properties would require further studies beyond the scope of this paper.
> - As suggested, we added the reference [Ref1] in the paper.  We mention that "*kernel functions can be linked to a wide range of statistical work, such as graph-based learning methods ([ref1])*"
>
>
> Again, Thank you for your careful review and constructive comments, which allow us to improve the paper. If you have any other concerns, please feel free to let us know.

---

> > ### Comment · Reviewer_toEQ · 2024-04-07
> >
> > I appreciate the effort of the authors put into answering my concerns. I leave the final decision to the AE.

---

### Review · Reviewer_GhSw · 2024-03-31

**Summary Of Contributions:**

Class imbalance is a common problem in supervised learning. One way to alleviate class imbalance is data augmentation, by adding synthetic data points from the minority class.

This paper presents a generalization of well-known data augmentation techniques SMOTE (Synthetic Minority Oversampling Technique), ROSE (Randomly Oversampling Examples), and GN (Gaussian noise) in the context of imbalanced regression (i.e. regression in the presence of rare extreme values). The idea is that all of these augmentation approaches involve: picking some “seed” point, identifying its nearest neighbors, and sampling from some distribution which is effectively a (convex) combination of kernel densities from the neighbors. The proposed algorithm GOLIATH (Generalized Oversampling for Learning from Imbalanced Datasets and Associated Theory) optimizes over various choices of kernel and the weighting and achieves better performance in practice. A loose theoretical explanation is provided: GOLIATH finds a sort of middle ground between “interpolation techniques” like SMOTE and “perturbation techniques” like ROSE and GN.

More specifically, GOLIATH operates as follows:
- Sample a seed s by looking at the inverse of the kernel value. The smaller this kernel value is, the farther the point is from its neighbors, the more of an isolated observation it is. So we are more likely to oversample these outlier points.
  - Notably, this method avoids discretizing the support of the target variable, which might be the more “obvious” approach to approaching imbalanced regression.
- Then generate the corresponding target y*_s by (1) using a random forest regressor to obtain a distribution on the residuals and (2) draw a random residual, scale it by a Gaussian, and add this to the y_s.

**Audience:**

Yes

**Claims And Evidence:**

No

**Requested Changes:**

Address the weaknesses that are discussed in the previous section.

**Strengths And Weaknesses:**

The paper presents a good idea, but the execution and presentation of this idea has serious problems. The paper is not in a publishable state, but with the requisite edits it could be.

Strengths:
- In terms of theoretical contributions: the paper provides a general framework for thinking about minority oversampling techniques. It shows how popular methods like SMOTE, GN, and ROSE specialize this more general kernel-based framework. The connection with SMOTE is perhaps the least obvious and made precise in Section 4.1.
  - NOTE for Section 4.1: Notation (regarding the seed S) is inconsistent when rewriting SMOTE in terms of GOLIATH; sometimes S is the index and sometimes it is the point
- Section 4.3 communicates effectively the different drawbacks between SMOTE and ROSE, and positions GOLIATH as something which, through generalization of the kernel, strikes a compromise.
- Figures 2 and 3 illustrate that GOLIATH-based methods for the kernel sampling technique work better than SMOTE and other existing methods for the presented cases.

Weaknesses:
- The overall paper is surprisingly poorly written. As a few examples: What is an imbalanced dataset in the context of regression is not even formally defined, description of interpolation approaches (Section 4.1) is unclear - what is “1” function (usually it is an indicator function but is it used differently, Section 5.3.1 suddenly starts discussing clustering?
- Section 3 is missing. All that is there is the title, “A New Kernel-Based Oversampling Formulation.”
- Lack of systematic theory. The paper title says “Associated Theory”, but there are hardly any theoretical results in the paper. Consistency of the proposed technique is handwaived by stating that it “can be considered as smooth bootstrap methods”.
- Section 5.3.1: The GOLIATH algorithm description is hard to follow. The font is small. The connection to the cartography next to it is unclear. Abbreviations like NCSB and NNSB and eNNSB should have been established in section 5, and it should have been made clearer that GOLIATH chooses one of these as a kernel setting. In fact, choosing between kernel settings seems to be the only thing communicated by this algorithm description: this might be better communicated in an itemized list or plain text. Furthermore, all of these subroutines called in the algorithm description ought to be made more precise somewhere in the text.
- Lack of detailed/systematic experimental analysis. There is practically no attempt made to understand how the proposed technique perform and scales with key parameters of practical dataset (eg number of training samples, dimensionality, level of ‘imbalance’, relationship between covariates and response, noise levels, etc).
- Lack of Runtime Analysis: Extending from nearest neighbors to support of the whole dataset potentially could be costly (See Section 5, subtitle “Extended Nearest Neighbors Bootstrap”). Same with training random forest regressor. Runtime of algorithms is only mentioned in Remark 2 and a footnote at the bottom of page 8.
- Figures 2 and 3 should do a better job of distinguishing between GOLIATH and previous methods. Also, is there any argument in the paper given for why we choose these particular datasets and whether they have any salient differences in size or geometry?

---

> ### Author Response · Authors · 2024-04-05
> **We sincerely appreciate your careful review, understanding, and constructive feedback. We have edited the submission with the updated paper based on your feedback., and we would like to provide some clarifications below.**
>
> **Weaknesses**
> - We apologize for the difficulty in understanding the notations. We have simplified the writing to improve readability. We have also better presented the problem as well as the contributions of our article. We have added a “*Notations and Problem Setting*” subsection to formally define the Imbalanced Regression. To ensure clarity and precision, we have introduced a clarification regarding the "1" function, specifying it as the indicator function. Additionally, we have revised the notations to improve consistency and readability throughout the document. In Section 5.3.1 we proposed to incorporate the Gaussian Mixture Models in Goliath as a pre-analysis. As said in Remark 5: “*It is also possible to use a clustering (Gaussian Mixture Model) in GOLIATH in order to apply a generation by cluster*”. This is a first step towards making better use of Goliath in each cluster.
>
> - We are truly sorry for this typo: "4 General Formulation of GOLIATH" is a subsection of 3, hence (3.1).
>
> - Indeed, the paper provides an explicit formulation for generating synthetic data, and our main contribution is to introduce new generators and methodologies to address imbalanced regression. It is therefore more a practical contribution than a theoretical one. However we discuss convergence results in Remarks 1 for classical kernels and in Remark 4 and for the non-classical kernels. For interpolation-based approaches, such as SMOTE, we note that very few  theoretical results have been provided to date in the literature, despite being the most well-known algorithm in Imbalanced Learning, with over 29,000 citations and +6,000 since 2023. The only theoretical properties are proposed by Elreedy et al. (2023), and more recently in Sakho et al (2024) as cited in the paper. The form of GOLIATH offers possible extensions of such results but this is beyond the scope of our article.
>
> - We moved the algorithm to the appendix and provided a detailed explanation.
>
> - This is an interesting point, but GOLIATH cannot be considered simply as another algorithm. It's rather a general formulation that encompasses the two main families of generators (see Branco et al., 2016 in the paper) and associates it with a statistical expression rather than an algorithmic one. Although such an analysis could be beneficial, its addition could significantly weigh down the paper, which is already quite rich in content since we compare for particular generators the potential of GOLIATH through 5 datasets already used in Branco et al. (2019).
>
> - We added the following remark in Appendix B.2 : “*Remark on the computation time: Both the UBL package and the GOLIATH algorithm are fast enough to generate a new sample: between 3 and 5 seconds for a dataset with about 500 rows. Note that, with the Non-Classical Smoothed Bootstrap, the estimation of the bandwidth parameter for a non-Gaussian distribution could take several minutes due to the package used, especially for a Binomial one.*”
>
> - Thank you for the advice. We updated the format of both figures to better distinguish GOLIATH from its competitors. We indicated in 7.2 : “*We test our approach on several real data set from a repository provided as a benchmark for imbalanced regression problems and presented in Branco et al. (2019) (descriptions in Appendix D).*”.
>
>
> Thank you once again for your careful review and constructive comments, which allow us to improve the paper. If you have any other concerns, please feel free to let us know.

---

### Comment · Reviewer_1Gej · 2024-04-13
**On the revised version**

Having read the revised portions of the submission (in blue font), I have the following comments:

* In Section 2, it contextually appears that $\hat{f}$ represents the KDE, but this is only defined later in Section 5.
* In Section 3.1, is the third point also an aside (since it is within square brackets)?
* In Section 3.1, I'm not sure what value the Yang et al. definition of imbalanced regression adds--as is, the quote defines imbalanced regression as regression over imbalanced data.
* Following this, you define it as, " learning from continuous distributions that include rare values (including extremes) that are precisely relevant to the studied phenomenon", without specifying what "rare" means yet. Also, what does "precisely relevant" mean?

---

> ### Author Response · Authors · 2024-04-14
> **Thank you for your careful review. The paper has been updated with your relevant comments**
>
> - It is indeed necessary to define $\widehat{f}$ at this location. So, we have added "*$(1-\alpha \widehat{f}(y)$ with $\widehat{f}(y)$ the estimated density function of the target variable $Y$)*.
>
> - In fact, this is a potential challenge in the case where synthetic data is generated, so it is not systematic (for example, with a weighting of the loss function of an algorithm). To be clearer, we have modified this part as follows: "*If the proposed solution to the imbalanced regression problem relies on synthetic data generation, then it introduces a **third challenge**. Indeed, unlike classification, where the labels of $Y$ remain unchanged when creating synthetic data, in regression, it is necessary to generate new relevant values for the target variable.*"
>
> - We mentioned this definition because it is supposed to be formal, and this paper introduced the concept of 'deep imbalanced regression'. We agree that this definition does not add much value, but this is more to show that there is not really a formal definition of the problem. We leave this reference to show the lack of consensus on this point. For example, Tian (2023) suggests another definition: "*The problem caused by the unbalanced distribution of the target value in the regression task is called the unbalanced regression problem. The unbalanced regression problem is shown in the following three aspects: 1) The distribution of the target value of the sample is unbalanced; 2) Samples in some ranges of target value are missing; 3) Samples in different intervals of the target value are divided into different importance levels.*”
>
> - In reality, there is not a formal definition of imbalanced regression, nor a precise definition of what constitutes a rare or relevant value. This lack of formality is not surprising because the imbalance problem is more complex than just the imbalance alone. This complexity also applies to imbalanced classification. Imbalanced Learning difficulties are not solely due to imbalance ratios; they also depend on data representativeness and the complexity of the modeled phenomenon. This point was emphasized by Krawczyk (2016), who noted that imbalance ratio alone isn't the sole source of learning challenges. Even with high disproportion, well-represented classes from non-overlapping distributions can achieve good classification rates using standard classifiers. Performance degradation can also stem from difficult examples, particularly within the minority class. This observation was echoed by Johnson (2019). Japkowicz (2000) investigated class imbalance effects by generating artificial datasets with varying complexity, training set sizes, and imbalance degrees. The findings revealed that sensitivity to imbalance increases with problem complexity, while linearly separable and non-complex classes are less affected by imbalance levels.
> Similar to binary classification, the imbalance in regression is not solely due to the imbalance itself. Defining the problem in imbalanced regression can be more challenging than in classification due to two key factors mentioned earlier.
> In our definition, the concept of the rarity of observations is important and is tied to the sample size. These rare values may not necessarily be observed at the extremes of distributions but also within certain sub-parts of the support, which may be empty, indicating missing data. It is this notion of rarity that contributes to the imbalance as these values lack sufficient influence in learning. Finally, to echo the definition of the UBL approach: these rare values may be more important. However, this is not always the case: sometimes the user wants a model that performs well for all values without necessarily being better for certain ones. Nonetheless, to rebalance the algorithms, more importance must be given to these rare values in learning.
>
>
> Tian (2023) Tian, H., Tian, C., Li, K., & Jia, W. (2023). Unbalanced regression sample generation algorithm based on confrontation. Information Sciences, 642, 119157.
>
> Krawczyk (2016) Krawczyk, B. (2016). Learning from imbalanced data: open challenges and future directions. Progress in Artificial Intelligence, 5(4), 221-232.
>
> Johnson (2019) Johnson, J. M., & Khoshgoftaar, T. M. (2019). Survey on deep learning with class imbalance. Journal of Big Data, 6(1), 1-54.
>
> Japkowicz (2000) Japkowicz, N. (2000, June). The class imbalance problem: Significance and strategies. In Proc. of the Int’l Conf. on artificial intelligence (Vol. 56, pp. 111-117).

---

> > ### Comment · Reviewer_1Gej · 2024-04-14
> >
> > Thank you for the detailed comments.
> >
> > * Yes, in my experience with classification, the imbalance alone is not the sole cause for poor performance; we found that neural networks that generate decision boundaries with low margin can also be a cause, so I agree with your statement here.
> > * Especially because this paper tackles imbalanced regression and samples at the extreme values, it is quite important that your paper define rarity clearly and precisely. From your comment, it seems your ideal target distribution would be uniform:
> >
> > > These rare values may not necessarily be observed at the extremes of distributions but also within certain sub-parts of the support, which may be empty, indicating missing data. It is this notion of rarity that contributes to the imbalance as these values lack sufficient influence in learning.
> >
> > Is my understanding correct?

---

### Decision · Action_Editor_azto · 2024-05-13

**Recommendation:** Accept with minor revision

**Comment:**

The paper studies regression under data imbalance, and presents and evaluates a novel algorithm for this problem. The authors position their algorithm among other oversampling-based methods via a general framework of kernel density estimation. This also allows the authors to motivate the choice of inverse KDE weights for their algorithm. Empirical results evaluate the proposed method on a benchmark of datasets with imbalance. The paper contained minor issues with grammar, terminology and notation, all of which seem to have been addressed. As requested by the reviewers, the authors also included discussions on computation time, presented a better motivation of the algorithm, and clarified the contributions.

The authors yet need to address the concerns raised by one of the reviewers on providing a more detailed/systematic experimental analysis. Concretely, for the final version of the paper, this can be achieved by adding empirical results on how the proposed technique performs and scales with key parameters of the datasets: number of training samples, dimensionality, level of imbalance (potentially through KDE weights), relationship between covariates and response, noise levels. These results are critical in order to ensure that the proposed method is properly evaluated, with an accurate assessment of strengths and weaknesses.

**Audience:**

I believe the problem being studied (regression under imbalanced data) is of interest to TMLR audience.

**Claims And Evidence:**

The claims made in the submission appear to be mostly supported by accurate, convincing, and clear evidence. The authors provide evidence in the form of experimental results on a benchmark of imbalanced regression problems. These experiments demonstrate the effectiveness of their proposed GOLIATH algorithm in imbalanced regression tasks. The authors also present how several oversampling-based algorithms for learning on imbalanced data relate to kernel density estimation. Working together with the reviewers, the authors also provide a definition that captures the imbalanced regression problem they are trying to solve (which has been in general lacking in the literature). This definition makes the problem statement and the contributions more accurate.

---

> ### Author Response · Authors · 2024-05-22
>
> We are very pleased to have received positive feedback and would like to sincerely thank you for the time and effort you dedicated to reviewing our paper. Your detailed and constructive comments have been incredibly valuable in improving the quality of our work. We especially appreciate your suggestions regarding adding detail to contributions, problem settings, definitions, and references, which have truly enriched and clarified our study.
>
> For the final version, we propose to conduct the following sensitivity analyses, which will help better evaluate our contributions:
>
> - Analysis of the impact of sample size: number of training samples
> - Analysis of the impact of weighting: with $\omega$ and $\alpha$ parameters and the level of imbalance
> - Analysis of the impact of noise: parameter $k$ for interpolation approaches and $pert$ for perturbation approaches
>
> Thank you once again, and we are available for any further clarification.